# SPriFed-OMP: A Differentially Private Federated Learning Algorithm for Sparse Basis Recovery

**Ajinkya K Mulay**                                                              *mulay@purdue.edu*
*Elmore Family School of Electrical and Computer Engineering*
*Purdue University*

**Xiaojun Lin\***                                                               *linx@ecn.purdue.edu*
*Elmore Family School of Electrical and Computer Engineering*
*Purdue University*

**Reviewed on OpenReview:** *https: // openreview. net/ forum? id=Dsavre6gjN)*

## Abstract

Sparse basis recovery is a classical and important statistical learning problem when the number of model dimensions $p$ is much larger than the number of samples $n$. However, there has been little work that studies sparse basis recovery in the Federated Learning (FL) setting, where the client data's differential privacy (DP) must also be simultaneously protected. In particular, the performance guarantees of existing DP-FL algorithms (such as DP-SGD) will degrade significantly when $p \gg n$, and thus, they will fail to learn the true underlying sparse model accurately. In this work, we develop a new differentially private sparse basis recovery algorithm for the FL setting, called *SPriFed-OMP*. *SPriFed-OMP* converts OMP (Orthogonal Matching Pursuit) to the FL setting. Further, it combines SMPC (secure multi-party computation) and DP to ensure that only a small amount of noise needs to be added in order to achieve differential privacy. As a result, *SPriFed-OMP* can efficiently recover the true sparse basis for a linear model with only $n = \mathcal{O}(\sqrt{p})$ samples. We further present an enhanced version of our approach, *SPriFed-OMP-GRAD* based on gradient privatization, that improves the performance of *SPriFed-OMP*. Our theoretical analysis and empirical results demonstrate that both *SPriFed-OMP* and *SPriFed-OMP-GRAD* terminate in a small number of steps, and they significantly outperform the previous state-of-the-art DP-FL solutions in terms of the accuracy-privacy trade-off.

## 1 Introduction

For many statistical learning applications, such as genomics and economics (Liang & Kelemen, 2008; Clarke et al., 2008; Belloni et al., 2014; Fan et al., 2011), it is essential to deal with situations where the number of samples ($n$) is significantly lower than the number of model parameters ($p$). Without additional constraints, such problems are *ill-determined* as there will be many models that can fit the same set of training samples. To address this issue, a popular approach is to assume some sparsity conditions on the desired model. Along this line, there have been significant advances in compressed sensing techniques, such as LASSO (Zhao & Yu, 2006), (Meinshausen, 2007) Orthogonal Matching Pursuit (OMP) (Tropp et al., 2007), (Needell & Tropp, 2009) Forward-Backward Algorithm (FoBA) (Zhang, 2011) and Least Angles Regression (LARS) (Efron et al., 2004), which leverage sparsity assumptions on the data to extract the true sparse basis of the underlying model. Once the sparse basis is identified, the sparse model can be estimated using standard learning techniques, as the problem is no longer ill-determined.

However, the above methods usually assume that the training datasets are already hosted on a server and thus neglect the privacy of the users who generate the training data. Recently, Federated Learning (FL)

---

\*Part of this work was completed when the second author moved to The Chinese University of Hong Kong.

(McMahan et al., 2017) has been proposed as a standard approach to protect user privacy, which keeps user data at the clients during training. Specifically, the clients update the models on their data and only upload the new gradients to the parameter server. The server updates the model parameters based on the aggregated global gradient and then sends the new model parameters back to the clients, furthering the training process. Such iterations are repeated until the global model converges, with no need to upload the user data to the server. However, note that FL is insufficient for privacy protection because the client gradients transmitted to the server may still leak information (Huang et al., 2021), (Geiping et al., 2020). Therefore, in the literature, Differential Privacy (DP) (Dwork et al., 2014) has also been introduced into FL. DP privatizes FL by adding noise to client gradients before uploading them to the parameter server. By adding appropriate noises, individual client gradients are hidden from the server and any external adversary while still enabling the learning of the global model. A notable example is DP-SGD, which can be applied to any SGD-based training algorithms and arbitrary loss functions (e.g., regularized objective functions in LASSO).

Unfortunately, this DP-FL pipeline is a poor fit for training sparse models when the number of training samples ($n$) is significantly below the number of model parameters ($p$). For ease of discussion, in this paper, we assume that each client contains a single training sample. Therefore, in our analysis, the number of clients equals the number of samples. [1] Note that, intuitively, the gradient uploaded by each client has $p$ elements, one for each model parameter. For DP to achieve the desirable privacy guarantee, we usually need to add noise to the gradients with variance proportional to the model dimensions (Dwork et al., 2014). Thus, when $p \gg n$, the noise required in the DP-FL setting can completely overwhelm the signal and thus prevent the recovery of the correct sparse basis. We refer to this problem as the *curse-of-dimensionality* for DP-FL.

In particular, for DP-SGD (Abadi et al., 2016), even when it is applied to Lipschitz loss functions, the empirical risk is of the order $\mathcal{O}(\frac{p}{n})$ (Theorem 2.4 in Bassily et al. (2014)). When $p \gg n$, even though one can apply DP-SGD to a sparse-solution-seeking objective, such as LASSO, it will not produce accurate answers. Other DP-FL algorithms, such as objective perturbation (Kifer et al., 2012), have similar issues. For example, the empirical loss of the objective perturbation mechanism in Kifer et al. (2012) is of the order $\mathcal{O}(\frac{p^2}{n})$ (see Theorems 4.1-4.2 and 5 in Kifer et al. (2012)), which will also fail to produce accurate sparse models when $p \gg n$. Thus, developing DP-FL algorithms that can attain provable sparse recovery under high-dimensional settings remains an open question. We note that if we do not consider the FL setting, there are DP solutions in the literature for sparse recovery under high dimension (Thakurta & Smith, 2013; Talwar et al., 2015). However, these algorithms assume full data access at the server and thus fail to work in the FL setting (where data are kept at the client for privacy).

In this work, we develop a new sparse-recovery algorithm specifically for the DP-FL setting, which, under suitable conditions, can ensure DP for data at the clients and recover the true sparse basis even when $p \gg n$. Specifically, our new algorithm called *SPriFed-OMP* (Sparse Private Federated OMP), is based on Orthogonal Matching Pursuit (OMP) (Tropp et al., 2007). At each iteration, OMP picks one basis with the highest correlation with the target, subtracts the contribution of this basis from the target, and continues to search for the next basis with the highest correlation with the residual until a given number $s$ of sparse basis is identified. However, the standard OMP (Tropp et al., 2007) is not designed for the FL setting, nor does it ensure DP. We augment OMP with a noisy SMPC (secure multi-party computation) algorithm (Bonawitz et al., 2017; Kairouz et al., 2021) so that only a smaller amount of noise is added to the aggregate correlation, which ensures DP. In some steps, e.g., to compare the correlation with the target across all bases, this noise is still in order $O(\sqrt{p})$. However, we only need to perform such steps $s$ times, which is usually much smaller than the number of iterations for DP-SGD to converge. As a result, the overall amount of noise will be much smaller. Our careful analysis shows that, as long as $n = \mathcal{O}(\sqrt{p})$, our algorithm will be able to recover the true sparse basis with high probability under the similar assumption of Restricted Isometry Property (RIP) as in standard OMP. While this general idea is quite intuitive, the detailed design of *SPriFed-OMP* also matters a lot. Indeed, we present two versions of *SPriFed-OMP*. In the first version, we privatize the individual correlations (feature and feature-residual correlations) to compute the total feature-residual correlation in each step. This version introduces our first enhancement that specifically adds lower order noise to the selected basis elements. Lower order noise enables lower sample size requirement for sparse basis recovery

---

[1]Our analysis can be easily extended to multiple samples per client by the group privacy notion discussed in Section 2.

while also incurring lower test error as we will see in Section 7. Next, we identify a simplified second version of *SPriFed-OMP* that introduces our second enhancement based on gradient privatization. We will see in the experimental section (Section 7) that adding noise to the gradient is much more advantageous under clipping than adding noise to correlations. Furthermore, for both methods, we can quantify the estimation error (Theorem 9) and the empirical risk (Theorem 10) to be on the order of $\mathcal{O}\left(\sqrt{\frac{s \log(s)}{n}}\right)$ and $\mathcal{O}\left(\frac{s \log(s)}{n}\right)$, respectively which are in the same order as the traditional non-private Ordinary Least Squares (OLS) estimate that already assumes knowledge of the correct sparse basis.

## 1.1 Related Work

In the literature, we have two major categories for private sparse learning: (1) The sparse learning methods that only work in the central DP setting (*i.e.,* where the data can be sent to the server non-privately and only the server privatizes the output) and, (2) the learning methods that are either directly DP-FL compatible or can be adapted to be suitable for the DP-FL setting. We note that not all of these methods can provide basis recovery guarantees (some of them only guarantees empirical risks). Below, we discuss the related work in each category.

**Central DP compatible sparse learning methods:** The recent sparse methods in Asi et al. (2021); Bassily et al. (2021) have empirical risk that holds for $n = \mathcal{O}(\sqrt{p})$ samples under the RSC constraint. However, both of these methods require centralized data access due to their unique sampling requirements and certain non-private mechanisms, and thus, they are not suitable for the DP-FL domain.

**DP-FL compatible sparse learning methods:** As we discussed earlier, DP-SGD is not designed for the $p \gg n$ setting or exploiting sparsity. Amongst the DP-FL sparse learning methods, the work of Kifer et al. (2012) is most closely related to ours. Although the method proposed in Kifer et al. (2012) is not geared toward the DP-FL setting, the *Samp-Agg* algorithm can be adapted to the DP-FL setting. However, support recovery in Kifer et al. (2012) requires that each client's data matrix individually satisfies the Restricted Strong Convexity (RSC) (Zhang, 2011) assumption. In contrast, the RIP assumption in our proposed solution is considerably weaker, as it only needs to hold over the entire training set across all clients. Thus, our solution can apply to a larger set of scenarios. There are two other related work that aim at the DP-FL and $p \gg n$ setting. Mangold et al. (2023) recently proposed a mechanism that has an empirical risk that holds for $n = \mathcal{O}(\sqrt{p})$ samples under the RSC constraint. We note that Mangold et al. (2023) is only studied for the centralized DP setting. Still, the algorithm can be appropriately adjusted for the DP-FL setting with the Gaussian noise [2]. However, their analysis lacks a sparse basis recovery guarantee, and thus it is unclear whether their methods can find the underlying true basis. Furthermore, their empirical risk values cannot avoid the polynomial dependency on $p$. Wang & Xu (2019) also considers the problem of estimating sparse models under the $p \gg n$ scenario and where noise is directly added to the client communication to the server to preserve DP. They provide two algorithmic results. The first result applies to the $p \gg n$ case, but it studies the privacy scenarios where only the response vector $\boldsymbol{y}$ must be private. The measurement matrix $\boldsymbol{X}$ has no privacy protection. When the matrix $\boldsymbol{X}$ must also be protected, their second result still requires $n = O(p)$ (ignoring the logarithmic terms). Thus, our proposed algorithms provide better privacy guarantees with fewer samples. In a similar sense, the private hard-thresholding methods proposed in Wang & Gu (2019); Hu et al. (2022) either do not recover the sparse models or require $n = O(p)$ samples for convergence.

**Noisy OMP:** Finally, our analysis of the noisy version of OMP is also related to (Chen & Caramanis, 2012), which studies how adding noise to the measurement matrix $\boldsymbol{X}$ will impact the accuracy of OMP. However, their analysis does not apply to the DP or the DP-FL cases as it does not privatize the response vector $\boldsymbol{y}$. Furthermore, they directly add noise to the measurement matrix $\boldsymbol{X}$. As a result, the noise level needed for ensuring DP for the matrix $\boldsymbol{X}$ would have had to be at least $\sqrt{pn}$, which is higher than our solution's. Specifically, our solution adds noise to the correlation between each basis and the target. Further, by employing *Noisy-SMPC* (Algorithm 1), the level of noise added is only of the order $\sqrt{ps}$. Thus, a new analysis is needed to study our improved DP and accuracy guarantees.

---

[2] We provide a modified/enhanced DP-FL version of Algorithm 1 in Mangold et al. (2023) for comparison in the form of Algorithm 5.

The rest of the paper is structured as follows. Section 2 presents the system model and provides a brief overview of differential privacy and the goal of compressed sensing. In Section 3, we provide the details of our proposed algorithm *SPriFed-OMP* and its other variant *SPriFed-OMP-GRAD*. Sections 4 and 5 provide a thorough utility-privacy analysis of our proposed implementation. Section 6 provides an intuitive sketch of the core result. Section 7 provides empirical results. Then, we conclude. Complete proofs and supporting results are omitted due to the page limits.

## 2 System Model

We assume that the ground truth model is $\boldsymbol{y} = \boldsymbol{x}\boldsymbol{\alpha}_* + \boldsymbol{\epsilon}$, where $\boldsymbol{\alpha}_* \in \mathbb{R}^p$ is the underlying model parameter that we wish to recover, $\boldsymbol{x} \in \mathbb{R}^p$ (a row vector) and $\boldsymbol{y} \in \mathbb{R}$ are the input-output pair, and $\boldsymbol{\epsilon} \sim \mathcal{N}(0, \sigma_\epsilon^2) \in \mathbb{R}$ is an additive error. We assume that this ground-truth model is sparse so that $\boldsymbol{\alpha}_*$ has at most $s$ non-zero elements, *i.e.,* $||\boldsymbol{\alpha}_*||_0 \leqslant s$. We now consider $n$ input-output training pairs represented by $(\boldsymbol{x_i}, y_i), i \in \{1, ..., n\}$ where each pair belongs to a single distinct client. [3] We stack the row vectors $\boldsymbol{x_i}$ vertically together to form an $n \times p$ matrix $\boldsymbol{X}$. Similarly, we stack $y_i$ into an $n \times 1$ vector $\boldsymbol{y}$. We emphasize that the server does not have the entire data, so we need to study a federated learning setting (McMahan et al., 2017).

Any DP-FL algorithm iterates over $n$ distributed clients over steps $t = 1, ..., T$. For each step $t$, the $i^{th}$ client transmits message $m_{i,t}$, computed as a function of the client dataset and any information it receives from the server plus possibly additional randomness. At the end of the $T$-th iteration, the server aims to recover $\boldsymbol{\alpha}_*$ (likely with some error). Let us use $\mathcal{M}$ to denote such a distributed and randomized mechanism. We aim to ensure Differential Privacy (DP) for the clients, and hence we present the Differential Privacy (DP) definition for $\mathcal{M}$ below.

**Definition 2.1.** *[**Approximate Distributed Differential Privacy**] For any two neighboring distributed datasets $\boldsymbol{X}, \boldsymbol{X}' \in \mathbb{R}^{n \times p}$ differing in one of their clients and the corresponding data-pairs (*i.e.*, they only differ in the data-pair for one client $k \in \{1, ..., n\}$). Let $\mathcal{M}(\boldsymbol{X})$ and $\mathcal{M}(\boldsymbol{X}')$ be the outputs when $\mathcal{M}$ operates on $\boldsymbol{X}$ and $\boldsymbol{X}'$ respectively. We say that the randomized and distributed mechanism $\mathcal{M}$ is $(\epsilon, \delta)$ differentially private if, for any set $\mathcal{F} \in \mathbb{R}^p$,*

$$Pr[\mathcal{M}(\boldsymbol{X}) \in \mathcal{F}] \leqslant e^\epsilon \cdot Pr[\mathcal{M}(\boldsymbol{X}') \in \mathcal{F}] + \delta.$$

Next, we introduce essential assumptions relevant to our system model. Recall that the goal of the server is to estimate the sparse model parameter $\boldsymbol{\alpha}_*$ without access to the client data. Even without the DP requirement, recovering the true sparse model typically requires additional assumptions on the data matrix (Zhang, 2009; Candes & Tao, 2005). Below, in assumption 1 we describe the Restricted Isometry Property (RIP), which is usually required for OMP (Candes & Tao, 2005), which our proposed algorithm is based on.

**Assumption 1.** *Restricted Isometry Property (RIP): A measurement matrix $\boldsymbol{X} \in \mathbb{R}^{n \times p}$ is said to satisfy RIP of order-K if there exists a constant $\zeta \in [0, 1]$ such that,*

$$(1 - \zeta)||\boldsymbol{v}||_2^2 \leqslant ||\boldsymbol{X}\boldsymbol{v}||_2^2 \leqslant (1 + \zeta)||\boldsymbol{v}||_2^2$$

*for all vectors $\boldsymbol{v} \in \mathbb{R}^p, ||\boldsymbol{v}||_0 \leqslant K$. Note that $\zeta$ is referred to as the isometric constant. Furthermore, the Restricted Isometry Constant (RIC) is defined as the infimum of all possible $\zeta$ values that satisfy RIP of order-K for a given measurement matrix $\boldsymbol{X} \in \mathbb{R}^{n \times p}$. In other words, assuming that the RIP of order $K$ is satisfied, the corresponding RIC is given by,*

$$\zeta_K = \inf_\zeta \Big\{ \zeta | (1 - \zeta)||\boldsymbol{v}||_2^2 \leqslant ||\boldsymbol{X}\boldsymbol{v}||_2^2 \leqslant (1 + \zeta)||\boldsymbol{v}||_2^2$$

$$\text{for all } \boldsymbol{v} \in \mathbb{R}^p, ||\boldsymbol{v}||_0 \leqslant K \Big\}.$$

---

[3] This assumption can be easily relaxed to allow more than one training pair per client by performing the privacy analysis with group privacy. As long as the samples of each client are considerably lower than $n$, the group privacy cost will be small. If a client holds a large number of samples, we could consider limiting the maximum number of samples per client in each iteration of our algorithm to upper bound our privacy costs.

We acknowledge that RIP is a more restrictive condition than other related conditions in the compressed-sensing literature, such as the Restricted Strong Convexity from Jalali et al. (2011) and Zhang (2009) or the Positive Cone Condition from Efron et al. (2004). However, even under this RIP assumption, no solution exists in the literature to attain DP in an FL setting and achieve exact support recovery with the number of samples much smaller than the model dimension. Thus, our contribution under the RIP assumption[4] still represents a significant contribution. We will leave the study of other assumptions for future work.

We also make the following assumption, typical in the differential privacy literature (Dwork et al., 2014; Wang & Xu, 2019).

**Assumption 2.** *[Bounded data matrix and response] We assume that the elements in the measurement matrix $\boldsymbol{X}$ and its response $\boldsymbol{y}$ are bounded by scalars $X_M$ and $y_M$ respectively.*

The boundedness of matrix or vector elements can be easily achieved by clipping the values to particular bounds. We can also easily maintain the original vector variance by re-scaling after clipping. Alternatively, if the underlying data distribution is light-tailed, we could leverage concentration bounds combined with the union bound to obtain a realistic bound on the data values.

As discussed in section 1, the standard DP approaches of adding noise are highly ineffective for recovering sparse models when $p \gg n$. Thus, our paper aims to develop a mechanism that is both DP-FL and can recover the exact support with several samples much smaller than the total model dimension. Specifically, let $\hat{\boldsymbol{\alpha}}$ denote the estimated model parameter of our proposed DP-FL algorithm. Recall that $\boldsymbol{\alpha}_*$ is the ground-truth model parameter with sparsity $s$. We wish to (1) Ensure that the support of $\hat{\boldsymbol{\alpha}}$ matches the true support with high probability, (2) Quantify the empirical risk $\Delta R \triangleq \frac{1}{n} \sum_{i=1}^{n} \left( (\boldsymbol{x_i}\hat{\boldsymbol{\alpha}} - y_i)^2 - (\boldsymbol{x_i}\boldsymbol{\alpha}_* - y_i)^2 \right)$ and estimation error $\Delta\alpha \triangleq ||\hat{\boldsymbol{\alpha}} - \boldsymbol{\alpha}_*||_2$ such that both are small and do not blow up with large $p$.

## 3   The *SPriFed-OMP* Algorithm

Our primary goal in this work is to recover a sparse model given a high number of features and a few samples. Note that if we can guarantee exact sparse basis recovery, we only need to add noise to the output model with variance proportional to the model sparsity, which is a much easier goal to accomplish. Thus, below, we will first focus on the goal to identify the true sparse basis. The analysis for recovering $\alpha_*$ with a small estimation error will then be presented later (see Theorems 9 and 10).

**Non-Private OMP:** Our proposed algorithm is based on OMP, a popular algorithm for exact sparse recovery without DP considerations. Below, we first describe the standard version of OMP. OMP iteratively selects a single new feature in each step. The feature with the absolute maximum correlation to the current residual is picked during each step. The residual is set to the model response $y$ in the initial step. At each subsequent step, the residual is updated by removing the newly picked feature's contribution from the response. OMP continues iterating until a model of a predetermined dimension/sparsity $s$ is selected.

**Challenge to make OMP differentially private:** Below, we explain the challenge to make OMP differentially private in the FL setting. First, when OMP computes the correlation of each basis with the residual and picks the basis with the highest correlation, it must be able to do so without direct access to the data on all clients. Second, when OMP subtracts the contribution of a new basis from the current residual, it needs the covariance of the existing basis. This must also be done without direct access to the data on all clients.

Both of these challenges can be overcome if we can differentially privately compute the correlation between two columns that are spread across all clients. Further, the total number of such computations must be carefully controlled to limit the amount of noise added. This idea leads to our first version of *SPriFed-OMP*, which focuses on computing the correlations in a DP-FL manner. The server maintains the estimated model parameter $\hat{\alpha}$ and is never released to the clients. (In contrast, the second version of *SPriFed-OMP*, which will be presented later, will release the privatized model parameter to the clients.)

---

[4]Note that we do not specifically assume that the clients are homogeneous. Even if the clients are heterogeneous, as long as the RIP assumption holds, our analytical results in the paper will hold.

**Differentially-Privately and Distributed Computation of Correlation :** Specifically, consider two columns of data $z_i^a$ and $z_i^b$ spread across all clients $i \in \{1, ..., n\}$. We now develop a mechanism so that the server can compute the sum $\sum_{i=1}^{n} z_i^a z_i^b$. For privacy, though, each client cannot disclose $z_i^a z_i^b$ directly. We can add noise directly to the client, which will lead to a noise level that is too high. In contrast, below, we use an approach that requires lower noise.

***NoisySMPC*:** We use the *NoisySMPC* mechanism (Algorithm 1) that combines SMPC (Bonawitz et al., 2017) with a much lower level of noise to ensure DP. This algorithm, which is distributed and satisfies DP, forms a core component of our proposed *SPriFed-OMP* algorithm. To achieve DDP (distributed and differentially private), *NoisySMPC* adds two levels of randomness. First, *NoisySMPC* modifies each client's contribution as $\tilde{f}(\boldsymbol{x_i}, y_i) = f(\boldsymbol{x_i}, y_i) + \eta_i$ where $\boldsymbol{x_i}, y_i$ is the $i^{th}$ client's data-response pair and $f$ computes a statistic such as the correlation between the client's data and response. The per-client noise $\eta_i$ is added so that the total noise $\eta_{sum} = \sum_{i=1}^{n} \eta_i$ at the server is sufficient to differentially privatize the sum of client outputs. Second, *NoisySMPC* adds $f(z_i^a, z_i^b)$ across all clients through SMPC, which further protects the privacy of individual clients. This underlying SMPC mechanism allows the clients to sum their contributions $\tilde{f}$ without disclosing any individual values. Examples of such SMPC mechanisms can be pair-wise client key sharing as described in (Bonawitz et al., 2017) or the distributed discrete Gaussian (Kairouz et al., 2021). We refer the reader to the above literature for further details regarding SMPC and related mechanisms. *NoisySMPC* adds significantly lower variance noise to the sum (reduced by a factor of $n$), compared to privatizing each client's output individually, as shown in line 11 of the Algorithm 1.

**Private OMP:** We are now ready to present the complete *SPriFed-OMP* algorithm in the DP-FL setting. Algorithm 3 contains the pseudocode. Recall that, for each step in OMP, we compute, for all features, the correlation of the feature column and the residual. For the first step, the residual is set to the model response, and thus, the computation can be denoted by $\boldsymbol{X^T y}$. The lines $4 - 6$ perform this computation privately using the *NoisySMPC* Algorithm 1. This noisy correlation selects the feature with the highest absolute correlation as part of the predicted basis (line 8). We then subtract the correlation contributed by the newly contributed feature $l^*$ from the previous residual to obtain the new residual. Mathematically, we represent this correlation contribution as $\boldsymbol{\beta_{l*}}(\boldsymbol{\beta_{l*,l*}})^{-1}\boldsymbol{\gamma_{l*}} = \boldsymbol{X^T X_{l*}}(\boldsymbol{X_{l*}^T X_{l*}})^{-1}\boldsymbol{X_{l*}^T y}$ where $\boldsymbol{\beta_{l*}} = \boldsymbol{X^T X_{l*}}, \boldsymbol{\beta_{l*,l*}} = (\boldsymbol{X_{l*}^T X_{l*}}), \boldsymbol{\gamma_{l*}} = \boldsymbol{X_{l*}^T y}$. Note that this computation computes the correlation between the new column $l^*$ and other columns. These correlations can be computed via the *NoisySMPC* mechanism. We represent these combined computations on lines $10 - 18$. Note that after a feature is chosen, its associated correlation computation must only be done twice via *NoisySMPC*. We first privatize correlations over all un-selected features (order $\sqrt{p-l}$ where $l$ are the number of features already chosen). From amongst these correlations, we choose the correlation with the highest value. Once the highest correlated feature is identified, we add noise to this correlation with a much lower noise value (of order $\int$ where $s$ is the sparsity of the model). Thus, we can significantly reduce the noise impact by privatizing the highest correlation twice.

*Remark:* We notice that, to privatize the $p$-dimensional correlations (lines *5* and *11*), we cannot reduce the variance of DP noise below $\mathcal{O}(p)$ (see the Gaussian mechanism from Dwork et al. (2014)). Thus, each time we compute the $p$-dimensional correlations, we are required to add noise of magnitude $\mathcal{O}(\sqrt{p})$. However, we only need to do so finitely many times (*i.e.,* $s$ times), thus requiring a finite privacy budget. Further, thanks to *NoisySMPC*, we can get away with adding a significantly smaller amount of noise (lines *14-17*).

Further, we note that for our method, we need to compute a maximum of $p(s+1) + 2s^2$ private correlations. In *DP-SGD*, we instead need to compute $pT$ private correlations, where the number of iterations $T$ is significantly higher than $s$ and often around the order of $n$. Furthermore, in our proposed algorithm, we require order-$p$ variance noise only for recovering the basis while order-$s$ variance noise is used to compute the model parameter on these bases. *DP-SGD* on the other hand requires order-$p$ variance noise throughout for all $T$ iterations. Thus, the noise required for our method is much lower in terms of the number of private correlation computations and the model parameter computation. Therefore, we expect to recover a model with significantly higher accuracy.

### *Enhancement 1: Adding lower noise to selected features*

While the above idea of privatize OMP may seem straightforward, there is a key step in Line *14* which greatly enhances the performance of SPriFed-OMP. Note that in lines *5* and *11* of Algorithm 3, we already

privatize the $p$ dimensional artifacts $\boldsymbol{X^T y}$ and $\boldsymbol{X^T X_{l*}}$ respectively with noise of variance $p\sigma_1^2$. Although we could have directly use such privatized values in estimating $\boldsymbol{\alpha}$, the resulting error will be high. Instead, in line *14*, we re-privatize the private model with much lower noise of variance $s\sigma_2^2$. We have found that this re-privatization step can significantly enhance the performance of our proposed algorithm, ensuring its sparse basis recovery success. Please see the numerical results in Section 7 for details. Intuitively, although we re-privatize the same features/artifacts twice, since, we add lower noise in the second time, the noise in $\tilde{\alpha}$ estimate (refer line *(15)* in Algorithm 3) is of order only depending on $s$. In contrast, if we do not utilize this modification, then the order of noise in both $\boldsymbol{\beta}$ and $\tilde{\boldsymbol{\alpha}}$ will be $\mathcal{O}(\sqrt{p})$. Since these estimates will be used in later steps to identify the next basis, without re-privatization the higher noise will disrupt the correlation computation, leading to stricter requirements on the sample size.

### 3.1 The SPriFed-OMP-GRAD Algorithm

We now present the second version of private OMP with our second enhancement in Algorithm 4 that further enhances the performance of Algorithm 3.

***Enhancement 2: Re-visiting SPriFed-OMP from the gradient perspective***

In line 15 of Algorithm 3, we compute the private correlation (essentially the gradient) at the $l^{th}$ step given by, $\boldsymbol{\gamma_0^-} - \boldsymbol{\beta\beta_{S_{\mathcal{A}}}\gamma_{S_{\mathcal{A}}}} = (\boldsymbol{\gamma_0^-})^{\boldsymbol{t}} + \boldsymbol{\eta_{\gamma_0}} - (\boldsymbol{\beta^t} + \boldsymbol{\eta_\beta})\tilde{\boldsymbol{\alpha}}$ where $(\boldsymbol{\gamma_0^-})^{\boldsymbol{t}}, \boldsymbol{\beta^t}$ represent the true (non-noisy correlations), $\boldsymbol{\eta_{\gamma_0}}, \boldsymbol{\eta_\beta}$ represent the corresponding noise values and $\tilde{\boldsymbol{\alpha}}(= \boldsymbol{\beta_{S_{\mathcal{A}}}\gamma_{S_{\mathcal{A}}}})$ represents the privatized linear model. We notice that since we are already privatizing the correlations required in the computation of $\tilde{\boldsymbol{\alpha}}$, we are naturally inclined also to privatize $(\boldsymbol{\gamma_0^-})^{\boldsymbol{t}}$ and $\boldsymbol{\beta^t}$ similarly. Thus, in our first approach of Algorithm 3, all clients collaboratively compute correlations, and then the server computes the final private model and the private gradient. However, our second approach improves this correlation computation by first computing the $L_2$ sensitivity (Definition A.2) of the gradient calculated using the private model (or mathematically, $(\boldsymbol{\gamma_0^-})^{\boldsymbol{t}} - \boldsymbol{\beta^t}\tilde{\boldsymbol{\alpha}}$) and then adding the corresponding DP noise to the entire correlation rather than separately privatizing each correlation. In this second approach, given the model, each client computes the gradient on its own device. Collaboratively, all clients and the server then compute the aggregated gradient. However, the server still handles the model computation with the noisy correlation method. That way, the server can share the private model with individual clients without sacrificing privacy. Later in the next section and the experimental section, we will also discuss a clipping-based version of the new algorithm, *SPriFed-OMP-GRAD* (Algorithm 4) that leverages clipping to provide better empirical performance.

We now present the second major enhancement in our proposed method in Algorithm 4. We note that the correlation computed in line *15* of Algorithm 3 is essentially the private gradient computed at the current model value. In Algorithm 3, we first separately privatize correlation values and then combine them to obtain the private gradient. On the other hand, in Algorithm 4, we let the server pass the private model to the clients, and let the clients directly compute the residues and the local gradients (*i.e.,* the gradient computed on line *4* in Algorithm 4). Then, we aggregate the local gradients in a private manner. We expect that Algorithm 4 will see a significant performance benefit due to the effect of clipping. Note that clipping can control the sensitivity of individual data items, and thus has a directly impact on the magnitude of DP noise needed. Since, gradients typically reduce in magnitude as training proceeds, we can expect to clip the gradients more aggressively without affecting performance. In contrast, the correlations in Algorithm 3 will stay at large values during the entire training, which is not amenable to clipping. We report experimental effects of clipping on gradients and correlation in section 7 to visualize their varying impact.

## 4 Privacy Analysis

This section includes the details for the privacy cost incurred by running Algorithms 3 and 4.

Since we need to account for the overall DP guarantee by composing multiple DP mechanisms, in this paper, we will use Gaussian Differential Privacy (GDP) (Dong et al., 2019) (a variant of DP) for composition. In the later part of this section, we will discuss the reason behind choosing GDP over other composition methods, e.g., those based on Renyi DP, in more detail. Before stating the Gaussian mechanism, we first formally define the $L_2$-sensitivity of a function.

---

**Algorithm 1** NOISY-SMPC

---

1: **Input Parameters: Vectors to be multiplied: $\boldsymbol{X_k}, \boldsymbol{X_j}$ (or $\boldsymbol{y}$) $\in \mathbb{R}^n$, Number of clients: $n$, Noise Variance: $\sigma_0^2$**
2: **Output: $\boldsymbol{X_k^T X_j} + \boldsymbol{\eta_*}; \boldsymbol{\eta_*} \sim \mathcal{N}(0, \sigma_0^2)$**
3: **procedure** NOISY-SMPC($\boldsymbol{X_k}, \boldsymbol{X_j}, n, \sigma_0$)
4:     Server broadcasts noise variance $\sigma_0^2$ to all clients
5:     **for** client $i \in [n]$ **do**
6:         $\eta_i \sim \mathcal{N}(0, \frac{\sigma_0^2}{n})$
7:         Compute $q_i \leftarrow \boldsymbol{X}_{k,i} \boldsymbol{X}_{j,i} + \eta_i$
8:     **end for**
9:     //   **Vanilla SMPC**
10:     Compute $q_{total} \leftarrow \sum_{i=1}^n q_i$ using SMPC (Bonawitz et al., 2017)
11:     //   The resulting private sum satisfies the following $\rightarrow q_{total} = \sum_{i=1}^n \boldsymbol{X}_{k,i}, \boldsymbol{X}_{j,i} + \eta_i = \boldsymbol{X_k^T X_j} + \sum_{i=1}^n \eta_i = \boldsymbol{X_k^T X_j} + \eta_*; \eta_* \sim \mathcal{N}(0, n \cdot \frac{\sigma_0^2}{n})(= \mathcal{N}(0, \sigma_0^2))$
12:     Server obtains and returns $q_{total}$ to all clients
13: **end procedure**

---

**Algorithm 2** PRIVATE-OLS

---

1: **Input Parameters: Data: $\boldsymbol{X} = \{\boldsymbol{X_1}, \boldsymbol{X_2}, ..., \boldsymbol{X_p}\} \in \mathbb{R}^{n \times p}$**, where $\boldsymbol{X_i}$ is the $i^{th}$ feature. **Response Variable: $y \in \mathbb{R}^n$, Feature Set: $S_\mathcal{A}$, New Feature to be privatized: $l^*$, Privacy Parameter: $\sigma_2$, True Model Support Cardinality: $s$.**
2: **Output: Private Model: $\tilde{\alpha} = (X_{S_\mathcal{A}}^T X_{S_\mathcal{A}} + \eta_{\beta_{S_\mathcal{A}}})^{-1}(X_{S_\mathcal{A}}^T y + \eta_2); \eta_{\beta_{S_\mathcal{A}}}$ and $\eta_{\gamma_{S_\mathcal{A}}}$ both have iid elements with distribution $\mathcal{N}(0, \sigma_2^2 s)$.**
3: **procedure** PRIVATE-OLS($\boldsymbol{X}, \boldsymbol{y}, S_\mathcal{A}, l^*, \sigma_2, s$)
4:     $\boldsymbol{\gamma_{S_\mathcal{A}}}[l] \leftarrow$ **NOISY-SMPC**($\boldsymbol{X}_{l^*}, y, n, \sigma_2\sqrt{s}$)    /* Note, that total private correlation over all previous rounds is $\boldsymbol{\gamma_{S_\mathcal{A}}} = \boldsymbol{X}_{S_\mathcal{A}}^T \boldsymbol{y} + \boldsymbol{\eta}_{\gamma_{S_\mathcal{A}}}; \boldsymbol{\eta}_{\gamma_{S_\mathcal{A}}} \in \mathbb{R}^{l \times 1}$ has iid elements from $\mathcal{N}(0, \sigma_2^2 s)$ */
5:     **for** $k = 0 : l$ **do**
6:         $\boldsymbol{\beta_{S_\mathcal{A}}}[l, k] \leftarrow$ **NOISY-SMPC**($\boldsymbol{X}_{l^*}, \boldsymbol{X_k}, n, \sigma_2\sqrt{s}$)
7:     **end for**   /* Note that, as $\boldsymbol{\beta_{S_\mathcal{A}}}$ is symmetric, we have $\boldsymbol{\beta_{S_\mathcal{A}}}[k, l] = \boldsymbol{\beta_{S_\mathcal{A}}}[l, k]$ and thus total private covariance is $\boldsymbol{\beta_{S_\mathcal{A}}} \triangleq \boldsymbol{X^T \widehat{S_\mathcal{A} X}}_{S_\mathcal{A}} + \boldsymbol{\eta}_{\beta_{S_\mathcal{A}}}; \boldsymbol{\eta}_{\beta_{S_\mathcal{A}}} \in \mathbb{R}^{s \times s}$ has iid elements from $\mathcal{N}(0, \sigma_2^2 s)$ */
8:     Return $(\boldsymbol{\beta_{S_\mathcal{A}}})^{-1}\boldsymbol{\gamma_{S_\mathcal{A}}}$
9: **end procedure**

---

**Definition 4.1** ($L_2$ sensitivity of a function). *Given a deterministic function $f$, consider all possible pairs of neighboring datasets $\boldsymbol{X}$ and $\boldsymbol{X}'$ in the domain of $f$ that differ on a single row. Then, we denote $l_2$-sensitivity of $f$ as*

$$\Delta_2 f \triangleq \max_{\boldsymbol{X}, \boldsymbol{X}' \in dom(f)} ||f(\boldsymbol{X}) - f(\boldsymbol{X}')||_2$$

Thus, $\Delta_2 f$ records the maximum possible change in $f$ due to a change in a single row. Often, each row belongs to a different client, and thus, $\Delta_2 f$ will record the maximum change due to a change in a client or due to the inclusion or removal of a client from the dataset. Below, we state the Gaussian mechanism from Dong et al. (2019) required to achieve the GDP guarantee.

**Lemma 1.** *[GDP mechanism (Theorem 2.7 from Dong et al. (2019))] Let $f : \mathbb{R}^{n \times p} \to \mathbb{R}^p$ be some function computed over the dataset $\boldsymbol{X} \in \mathbb{R}^{n \times p}$. Then, the randomized Gaussian mechanism $\mathcal{M}(\boldsymbol{X}) = f(\boldsymbol{X}) + \eta$ is $\mu - GDP$ where $\eta \sim \mathcal{N}(0, \frac{(\Delta_2 f)^2}{\mu^2}\mathbb{I}_p)$ and $\Delta_2 f$ is the $L_2$- sensitivity of $f$ (Definition A.2).*

**Reasons behind choosing GDP for privacy analysis:** We choose GDP because GDP gives the optimal privacy guarantees for compositions of Gaussian mechanisms. For example, GDP has been shown to produce lower $\epsilon$ values when composed over a finite (possibly small) number of Gaussian mechanism steps $T$ Dong

---

**Algorithm 3** SPriFed-OMP: Private Orthogonal Matching Pursuit

---

1: **Input Parameters: Data:** $X = \{X_1, X_2, ..., X_p\} \in \mathbb{R}^{n \times p}$, where $X_i$ is the $i^{th}$ feature. **Response Variable:** $y \in \mathbb{R}^n$, **Feature Set:** $\Omega \triangleq \{1, 2, ..., p\}$, **True Model Support Cardinality:** $s$, **GDP Privacy Parameters:** $\mu_p, \mu_s(\mu_p > \mu_s)$, **Noise standard deviation:** $\sigma_1 = \frac{1}{\mu_p}, \sigma_2 = \frac{1}{\mu_s}$.
2: **Output:** Predicted Support $S_{\mathcal{A}}$, Predicted sparse model $\hat{\alpha}$ over predicted support $S_{\mathcal{A}}$
3: **Initialize:** $S_{\mathcal{A}} = \varnothing, X_{S_{\mathcal{A}}} = \bar{0}, l = 0$
4: **for** $k = 0 : p - 1$ **do**    /* **Server and Clients privately compute data-response correlation** */
5:     $(\gamma_0^-)[k] \leftarrow$ **NOISY-SMPC**$(X_k, y, n, \sigma_1\sqrt{p})$
6: **end for**    /* *Note that the total private correlation overall k is* $\gamma_0^- \triangleq \widehat{X^T y} = X^T y + \eta_{\gamma}; \eta_{\gamma} \in \mathbb{R}^{p \times 1}$ *has iid elements from* $\mathcal{N}(0, \sigma_1^2 p)$ */
7: **while** $|S_{\mathcal{A}}| \leqslant s$ **do**
8:     $l^* \leftarrow \arg\max_{j \in \Omega / S_{\mathcal{A}}} |(\gamma_l^-)_j|$    /* **Server privately extracts new highest-correlated feature** */
9:     $S_{\mathcal{A}} \leftarrow S_{\mathcal{A}} \cup l^*$
10:     **for** $k = 0 : p - 1$ **do**    /* **Server and Clients privately compute data covariance for newly extracted feature** */
11:         $\beta[k, l] \leftarrow$ **NOISY-SMPC**$(X_k, X_{l^*}, n, \sigma_1\sqrt{p})$
12:     **end for**    /* *Note that, the total private covariance over all previous rounds is* $\beta = \widehat{X^T X_{S_{\mathcal{A}}}} \triangleq X^T X_{S_{\mathcal{A}}} + \eta_{\beta}; \eta_{\beta} \in \mathbb{R}^{p \times (l+1)}$ *has iid elements from* $\mathcal{N}(0, \sigma_1^2 p)$ */
13:     //    **Server privately updates residual with newly chosen feature**
14:     $\tilde{\alpha} \leftarrow$ **PRIVATE-OLS**$(X, y, S_{\mathcal{A}}, l^*, \sigma_2, s)$
15:     $\gamma_l^- \leftarrow \gamma_0^- - \beta\tilde{\alpha}$
16:     $l = l + 1$
17: **end while**
18: Return $S_{\mathcal{A}}, \hat{\alpha} \leftarrow (\beta_{S_{\mathcal{A}}})^{-1}\gamma_{S_{\mathcal{A}}}$    /* **Server privately publishes sparse basis and model overall extracted features** */

---

**Algorithm 4** SPriFed-OMP-GRAD: Private Orthogonal Matching Pursuit

---

1: **Input Parameters: Data:** $X = \{X_1, X_2, ..., X_p\} \in \mathbb{R}^{n \times p}$, where $X_i$ is the $i^{th}$ feature. **Response Variable:** $y \in \mathbb{R}^n$, **Feature Set:** $\Omega \triangleq \{1, 2, ..., p\}$, **True Model Support Cardinality:** $s$, **GDP Privacy Parameters:** $\mu_p, \mu_s(\mu_p > \mu_s)$, **Noise standard deviation:** $\sigma_1 = \frac{1}{\mu_p}, \sigma_2 = \frac{1}{\mu_s}$, **Privacy Constants:** $X_M = y_M = 1, B_C = 1 + 2X_M\kappa_{\varepsilon} + 2\sqrt{s}X_M^2(\frac{\|(\alpha_*)_{S_{\mathcal{A}}^c}\|_{\infty}}{1-\zeta_{s+1}} + \frac{\sqrt{1+\zeta_{s+1}}\kappa_{\varepsilon}}{1-\zeta_{s+1}})$ (refer Lemma 5 and Section 7 for further details about setting $B_C$ in practice).
2: **while** $|S_{\mathcal{A}}| \leqslant s$ **do**
3:     **for** $j = 0 : p$ **do**
4:         $\tilde{\gamma}_l[j] \leftarrow$ **NOISY-SMPC**$(X_j, y - X_{S_{\mathcal{A}}}\tilde{\alpha}, n, \sigma_1 B_C\sqrt{p})$
5:     **end for**
6:     $l^* \leftarrow \arg\max_{j \in \Omega / S_{\mathcal{A}}} |(\tilde{\gamma}_l)_j|$
7:     $S_{\mathcal{A}} \leftarrow S_{\mathcal{A}} \cup l^*$
8:     //    **Server privately updates model with newly chosen feature**
9:     $\tilde{\alpha} \leftarrow$ **PRIVATE-OLS**$(X, y, S_{\mathcal{A}}, l^*, \sigma_2, s)$
10:     $l = l + 1$
11: **end while**

---

et al. (2019). This small-$T$ regime is important for our work because our proposed algorithms aim to find the correct basis with only a small number of iterations (proportional to the sparsity level $s$). Readers can refer to Table 1 of Reference Liu et al. (2022) for an empirical comparison of various adaptive composition mechanisms where GDP performs the best (i.e., produces the smallest $\epsilon$ values across varying values of $\delta$) compared to other composition mechanisms including Renyi DP, Advanced composition and naive DP. For related results on the non-adaptive composition of Gaussian mechanisms see Theorem 8 in Balle & Wang (2018)). Finally, we note that a single value can demonstrate GDP guarantees, and GDP can easily and optimally be composed over multiple iterations via Corollary 3.3 in (Dong et al., 2019). Furthermore, we

can convert between a $\mu - GDP$ guarantee and the $(\varepsilon, \delta)$-DP guarantee losslessly via Corollary 2.13 in Dong et al. (2019). However, we only need to convert $\mu$-GDP to DP for our purpose. Below, we present both of these lemmas.

**Lemma 2.** *[Conversion from $\mu$-GDP to $(\varepsilon, \delta)$-DP] (Corollary 2.13 in Dong et al. (2019))* *A mechanism is $(\varepsilon, \delta(\varepsilon))$-DP for all $\varepsilon \geqslant 0$ and*

$$\delta(\varepsilon) = \Phi(-\frac{\varepsilon}{\mu} + \frac{\mu}{2}) - e^{\varepsilon}\Phi(-\frac{\varepsilon}{\mu} - \frac{\mu}{2}).$$

*if it is $\mu$-GDP. Here, $\Phi$ denotes the standard normal CDF.*

Further, multiple mechanisms with $\mu$-GDP into an overall GDP mechanism.

**Lemma 3.** *Composition of $\mu$-GDP (Corollary 3.3 in Dong et al. (2019))* *The composition $k$-GDP mechanisms, each of which is $\mu_i$-GDP, $i = 1, ..., k$ is $\sqrt{\sum_{i=1}^{k} \mu_i^2}$-GDP.*

In summary, leveraging GDP allows us to obtain better composition results, control the privacy leakage by a single parameter, and losslessly convert between GDP and $(\varepsilon, \delta)$-DP. For further details regarding GDP, we refer the reader to Dong et al. (2019).

For our proposed Algorithm 3, we notice that we only need to differentially privatize the terms $\boldsymbol{X_j^T y}$ and $\boldsymbol{X^T X_j}$ and their combinations. Thus, we use Lemma 4 to identify their $L_2$-sensitivity (Definition A.2).

**Lemma 4.** *Under Assumption 2, the $L_2$-sensitivity (Definition A.2) of the terms $\boldsymbol{X_j^T y}$ and $\boldsymbol{X^T X_j}$ for any $j \in S_{\mathcal{A}}$ as defined in Algorithm 3 are given by $2\sqrt{p}X_M y_M = \mathcal{O}(\sqrt{p})$ and $2\sqrt{p}X_M^2 = \mathcal{O}(\sqrt{p})$ respectively. Here, $\boldsymbol{X} \in \mathbb{R}^{n \times p}$ and $\boldsymbol{y} \in \mathbb{R}^n$.*

*Proof.* The proof is available in the Appendix section C. $\square$

**Lemma 5.** *Consider a design matrix $X \in \mathbb{R}^{n \times p}$ satisfying RIP of order $s + 1$ and RIC $\zeta_{s+1}$. Now given the correlation/gradient $C_j$ (as computed in Line 7 of algorithm 4) for any $j \in [p]$, we show that the upper bound on the $L_2$ sensitivity of the expression is upper bounded by the following values,*

$$\Delta_2(C_j) \leqslant 1 + 2X_M \kappa_{\varepsilon} + 2\sqrt{s}X_M^2 \left(\frac{\|(\alpha_*)_{S_{\mathcal{A}}^c}\|_{\infty}}{1 - \zeta_{s+1}} + \frac{\sqrt{1 + \zeta_{s+1}}\kappa_{\varepsilon}}{1 - \zeta_{s+1}}\right) \triangleq B_C$$

*when Assumptions 1 and 2 hold. Here, $\alpha_*$ is the underlying ground-truth model, $\kappa_{\varepsilon} = \sqrt{2\log(2n/p_b)}\sigma_{\varepsilon}$ and $\sigma_{\varepsilon}$ is the additive system error's standard deviation. We also need to assume that $\zeta_{s+1} < \frac{1}{\sqrt{s}}$ and $n > \frac{6s^2 C_1 X_M^5 y_M \sqrt{2\log(2s/p_b)}}{\mu_s^2(1 - \zeta_{s+1})^2}$. The result holds with probability $1 - 3p_b$ where $p_b$ is a small positive probability value.*

*Proof.* The proof is provided in the Appendix Section F. $\square$

**Theorem 6.** *Algorithms 3 and 4 satisfies $\mu$-GDP with $\mu = \sqrt{s \cdot \mu_p^2 + 2s \cdot \mu_s^2}$, where $\mu_p, \mu_s$ are privacy constants as inputs to Algorithm 3.*

*Proof.* The proof follows simply by Lemma 3. In our proposed algorithm, we use the $\mu_p$-GDP (lines *5* and *11* in Algorithm 3 and line *4* in Algorithm 4) mechanism $s$ times and the $\mu_s$-GDP mechanism $2s$ times (lines *4* and *6* in algorithm 2). Thus, the result follows. Finally, using Lemma 2, we can obtain the corresponding DP guarantee of Algorithm 3. $\square$

## 5 Accuracy of Private Orthogonal Matching Pursuit

This section will present our main result on the accuracy of sparse basis recovery. Theorem 7 states that the proposed algorithm *SPriFed-OMP* (Algorithm 3) terminates in finite steps and can recover the true sparse basis and the corresponding model *w.h.p.* as $n \to \infty$ and when $n = \mathcal{O}(\sqrt{p})$.

Before we state our main results, we define the relevant constants. Let $\zeta_{s+1}$ denote the RIC of $\boldsymbol{X}$ of $(s+1)$-order. Let $\nu$ be a small positive constant, close to 1, that satisfies the inequality $\frac{1}{1-\zeta_{s+1}} \leqslant (1 + \nu\zeta_{s+1})$. Next, we define $\kappa_\epsilon = \sigma_\varepsilon\sqrt{2\log(\frac{2n}{p_b})}$, $\kappa_1 = \kappa_{M_2}\sqrt{\log(s/p_b)}/\mu_s$ and $\kappa_2 = \kappa_{M_2}\sqrt{\log(p/p_b)}/\mu_p$. Recall that $\sigma_\varepsilon$ is the standard deviation of the additive error in the system model presented in Section 2. Here, $p_b$ is a small positive probability, eventually affecting the with-high-probability statement of Theorem 7. Further define the constant $\kappa_{M_2}$ as

$$\kappa_{M_2} = 1 + \frac{C_1 s^{3/2}}{n\mu_s(1 - \zeta_{s+1})},$$

where the constant $C_1$ (which depends on $p_b$) is chosen such that the following probability statement holds: For a square matrix $N$ with dimensions $s \times s$, where the elements follow an i.i.d. distribution of $\mathcal{N}(0, 1)$, its largest eigenvalue is no greater than $C_1\sqrt{s}$ with probability $1 - p_b$, *i.e.*,

$$\Pr\left(||N||_2 > C_1\sqrt{s}\right) \leqslant p_b$$

By these choices (which will be used in our proof in the technical report), each constant bound the corresponding random variable with probability $1 - p_b$. Finally, we denote the true basis of the underlying model by $S_*$.

**Theorem 7.** *[Sparse recovery of SPriFed-OMP] Under the provided system model (Section 2) and Assumptions 1 and 2, suppose that the following conditions holds:*

1. *Sample size:* $n \geqslant \max\{4\sqrt{p}s\kappa_1, 16s^{5/2}\kappa_2, \frac{4\sqrt{\log(p-s)}\sqrt{p}}{\mu_p\|\alpha_{S_*}\|_2}\} = \mathcal{O}(\max\{\frac{s\sqrt{p\log p}}{\mu_s\mu_p}, \frac{s^{5/2}}{\mu_s}\})$

2. *RIC of* $\boldsymbol{X}$ *of order-$s+1$:* $\zeta_{s+1} \leqslant \frac{1}{4(1+\sqrt{s})}$

3. $\kappa_\epsilon$ *:* $\kappa_\epsilon \leqslant \frac{min_{j \in S_*}|\boldsymbol{\alpha_j}|}{16(\sqrt{s}+1)(1+\nu\zeta_{s+1})}$

*Then, the SPriFed-OMP algorithm will correctly recover the true basis with probability $1 - 40(s + 1) \cdot p_b$.*

*Remark (Intuition behind the assumptions for successful recovery in Theorem 7)*: If we ignore the $\log p$ term, condition 1 only requires $n = \mathcal{O}(\sqrt{p})$ for sparse recovery. To the best of our knowledge, this is the first result in the literature that both attains DP in an FL setting and attains sparse recovery when $n$ is much smaller than $p$. We note that condition 1 is stricter than typical OMP results in the non-private settings, where $n = \mathcal{O}(\log p)$ is sufficient. We believe the main reason for this difference is that we must simultaneously ensure DP in the FL setting. As we discussed earlier when we present Algorithm 3, even in the first step for computing the data-response correlation, $\mathcal{O}(\sqrt{p})$ noise seems unavoidable (remark under Section 3). Thus, it seems difficult to suppress the noise when $n$ is smaller than $\sqrt{p}$.

Condition 2 is related to the strength of the RIP assumption. The RIC is an important attribute of the RIP. The smaller the RIC, the closer $\boldsymbol{X}$ is to a unitary matrix, and thus, the corresponding condition 2 becomes more demanding. Our condition 2 provides a simple upper bound for the RIC. As expected, we observe that, with the increasing value of $s$, our RIC bound reduces, implying that sparse recovery gets more challenging as the model cardinality increases. This intuition is in line with that in the non-private OMP literature.

Condition 3 on the additive error in the training data is also intuitive. If the minimum true-model column norms are large enough (*i.e.,* the signals due to true features are sufficiently strong), then we expect that the impact of the additive error in the system will be minuscule. This is reflected by the term $min_{j \in S_*}|\boldsymbol{\alpha_j}|$ in the numerator. We also expect that model recovery gets more complex with rising model cardinality (*i.e.,* rising $s$). Thus, the impact of the error in the training data will rise as reflected by the $\sqrt{s} + 1$ term in the denominator. Similar to Theorem 8, we can state the recovery guarantee of *SPriFed-OMP-GRAD* below.

**Theorem 8.** *[Sparse Recovery of SPriFed-OMP-Grad] Under the provided system model (Section 2) and Assumptions 1 and 2 suppose that the following conditions hold,*

1. *Sample Size:* $n \geqslant \max\left\{\frac{4(\sqrt{s}+1)(1+\sqrt{s}(\kappa_\alpha+1))\sqrt{p}\kappa'_p}{\mu_p\|\alpha_{S^c_\mathcal{A}}\|_2}, \frac{4\sqrt{\log(p-s)}\sqrt{p}}{\mu_p\|\alpha_{S_*}\|_2}\right\} = \mathcal{O}(\frac{s\sqrt{p\log p}}{\mu_p})$

2. *RIC of X of order $s+1$:* $\zeta_{s+1} \leqslant \frac{1}{\sqrt{s+1}}$

3. $\kappa_\varepsilon < \frac{\min_{j\in S_*}|\alpha_j|}{6(1+\nu\zeta_{s+1})^{3/2}\left(\sqrt{s+1}+\frac{C_1\sqrt{s}\kappa'_s\sqrt{1+\zeta_{s+1}}}{n\mu_s}\right)}$

*Then the SPriFed-OMP-GRAD algorithm will correctly recover the true basis with probability $1 - 8(s+1)p_b$. Note that here, $\kappa_\alpha = \frac{\sqrt{1+\zeta_{s+1}}(\sqrt{1+\zeta_{s+1}}\|\alpha_*\|+\kappa_\varepsilon)}{1-\zeta_{s+1}}, \kappa'_p = \sqrt{2\log(\frac{2p}{p_b})}, \kappa'_s = \sqrt{2\log(\frac{2s}{p_b})}$ and $B_C = 1 + 2X_M\kappa_\varepsilon + 2\sqrt{s}X_M^2(\frac{\|(\alpha_*)_{S^c_\mathcal{A}}\|_\infty}{1-\zeta_{s+1}} + \frac{\sqrt{1+\zeta_{s+1}}\kappa_\varepsilon}{1-\zeta_{s+1}})$ and $p_b$ is a small positive probability value similar to Theorem 7.*

*Remark (The utility-privacy trade-off)* Although we do not explicitly mention privacy requirements in Theorems 7 and 8, the results here directly depend on the privacy parameters such as $\mu_s$ and $\mu_p$. Thus, we can combine them with Theorem 6 to obtain the utility-privacy trade-off. Specifically, suppose that we wish to re-write the conditions stated in Theorems 7 and 8 in terms of $(\epsilon, \delta)$-DP guarantees. We first pick $\mu$, which can be easily converted to $(\epsilon, \delta)$ DP guarantees according to Lemma 2. Then, by assuming that $\mu_s = \mu_p \cdot m_s$, where $m_s$ is a known constant factor, we obtain from Theorem 6 that $\mu = \mu_p\sqrt{s(1 + 2 \cdot m_s^2)}$. We can thus solve $\mu_p$ and $\mu_s$ as:

$$\mu_p = \mu/\sqrt{s(1 + 2 \cdot m_s^2)}$$
$$\mu_s = \mu_p \cdot m_s.$$

These values can then be directly plugged into Theorems 7 and 8 to obtain the accuracy guarantees.

*Remark (Benefits of Algorithm 4 over Algorithm 3*: Readers can see that the conditions of Theorem 8 are on the same order as that of Theorem 7. However, this is because we assume the DP noise magnitudes are the same in both cases. In practice and as we discussed earlier, due to the decreasing magnitude of the gradients, Algorithm 4 can benefit from more aggressive clipping and lower DP noise magnitude. We will present numerical results to demonstrate the benefits of these enhancement in Section 7.

Based on our main Theorems 7 and 8, next, we provide results on the parameter estimation error and empirical risk. Both of these results are optimal as they are in the same order as several previous results for both the parameter estimation error (Meinshausen & Yu, 2009; Wasserman & Roeder, 2009) and the empirical risk analysis (Kifer et al., 2012; Bassily et al., 2014) even under the assumption that they have already known the correct basis before-hand. In particular, these error bounds depend on $s$ but are independent of the dimension $p$ since we have already identified the correct sparse basis as in Theorems 7 and 8.

**Theorem 9.** *[**Estimation Error of SPriFed-OMP and SPriFed-OMP-Grad**] Under the same system model and assumptions as Theorem 7, with a high probability of $1 - p_b$, the estimation error satisfies,*

$$\Delta\boldsymbol{\alpha} \triangleq \|\hat{\boldsymbol{\alpha}} - \boldsymbol{\alpha}_{S_*}\|_2$$

$$\leqslant \frac{(1-\zeta_{s+1})\sqrt{2s\log(2s/p_b)}\sigma_\varepsilon X_M}{\sqrt{n}} + \frac{s\sqrt{2\log(\frac{2s}{p_b})}}{n\mu_s(1-\zeta_{s+1})}$$

$$+ \frac{s^{3/2}\kappa_M\sqrt{2\log(\frac{2s^2}{p_b})}}{\mu_s n(1-\zeta_{s+1})^2}\Big((1+\zeta_{s+1})\|\alpha_{S_*}\|_2$$

$$+ \sqrt{1+\zeta_{s+1}}\kappa_\epsilon + \frac{\sqrt{s}\sqrt{2\log(\frac{2s}{p_b})}}{\mu_s n}\Big)$$

$$= \mathcal{O}\Big(\max\left\{\sqrt{\frac{s\log(s)}{n}}, \frac{s^{3/2}\sqrt{\log(s)}}{n\mu_s}\right\}\Big)$$

*where $\hat{\boldsymbol{\alpha}}$ is the predicted model from the SPriFed-OMP algorithm, $\boldsymbol{\alpha}_{\boldsymbol{S}_*}$ is the true ground-truth model, $\zeta_{s+1}$ is the RIC constant for the matrix $\boldsymbol{X}$ and $\mu_s$ is as defined in Algorithm 3. Our result holds with a high probability of $1 - 4p_b$.*

**Proof:** The proof is omitted due to page limits.

**Theorem 10.** *[**Risk Analysis of SPriFed-OMP**] Under the same system model and assumptions as Theorem 7, with probability $1 - 4p_b$ the risk $\Delta R$ from algorithm SPriFed-OMP is bounded by,*

$$
\begin{aligned}
\Delta R &\triangleq R(\hat{\boldsymbol{\alpha}}; \boldsymbol{X}, \boldsymbol{y}, n) \\
&= \frac{1}{n} \sum_{i=1}^{n} \left( (\boldsymbol{x_i}\hat{\boldsymbol{\alpha}} - y_i)^2 - (\boldsymbol{x_i}\boldsymbol{\alpha_{S_*}} - y_i)^2 \right) \\
&\leqslant 2\left( \frac{(1 + \zeta_{s+1})s\sqrt{2\log(\frac{2s}{p_b})}}{\mu_s n(1 - \zeta_{s+1})} \right)^2 \\
&\quad + 2\left( \frac{(1 + \zeta_{s+1})\sqrt{2ns\log(2s/p_b)}\sigma_\varepsilon X_M}{n(1 - \zeta_{s+1})} \right)^2 \\
&\quad + 2\left( \frac{s^{3/2}\kappa_M\sqrt{2\log(\frac{2s^2}{p_b})}}{\mu_s n(1 - \zeta_{s+1})^2} \left( (1 + \zeta_{s+1})\|\alpha_{S_*}\|_2 \right. \right. \\
&\quad \left. \left. + \sqrt{1 + \zeta_{s+1}}\kappa_\epsilon + \frac{\sqrt{s}\sqrt{2\log(\frac{2s}{p_b})}}{\mu_s n} \right) \right)^2 \\
&= \mathcal{O}\left( \max\left\{ \frac{s\log(s)}{n}, \frac{s^3\log(s)}{n^2\mu_s^2} \right\} \right)
\end{aligned}
$$

*where $\hat{\boldsymbol{\alpha}}$ is the predicted model from the SPriFed-OMP algorithm, $\boldsymbol{\alpha}_{\boldsymbol{S}_*}$ is the true ground-truth model, $\zeta_{s+1}$ is the RIC constant for the matrix $\boldsymbol{X}$ and $\mu_s$ is as defined is defined in Algorithm 3. $\boldsymbol{X}, \boldsymbol{y}$ are the data matrix and the label vector while $n$ is the number of clients.*

**Proof:** The proof is omitted due to page limits.

*Remark (privacy parameters in Theorems 9 and 10* Results in Theorems 9 and 10 only depend on the $\mu_s$ parameter and not $\mu_p$ privacy parameter as the privatized model in these results are outputted based on the PRIVATE-OLS routine which only adds noise with a privacy parameter of $\mu_s$ (as the predicted bases are already known).

## 6 Proof Sketch of Theorems 7 and 8

This section identifies the key steps required to prove Theorems 7 and 8. From Algorithms 3 and 4, we notice that in the very first round, the server picks the feature with a maximum absolute data-response correlation over the entire feature set. To ensure that we only pick features from the true feature set, we thus require expression 1 below to hold.

$$
\min_{j \in S_*} |\boldsymbol{X_j^T}\boldsymbol{y} + \boldsymbol{\eta_{\gamma j}}| \geqslant \max_{j \notin S_*} |\boldsymbol{X_j^T}\boldsymbol{y} + \boldsymbol{\eta_{\gamma j}}|. \tag{1}
$$

In other words, for correct basis recovery to occur in the first round, the minimum absolute feature correlation over the true set should exceed the maximum absolute feature correlation over all the features not in this true set. The strategy for proving 1 is to bound the randomness-inducing terms (*i.e.,* noise terms) with high probability. This high-probability event will also dictate the relationship between the sample size, the restricted isometry constant, and the rest of the system constants. Using the restricted isometry property over the dataset contributed by all clients, we can then show that the above inequality holds. The remaining rounds are handled similarly over the updated residuals. The complete proof is available in the Appendix sections E and F.

*Remark:* Here, we re-iterate that both algorithms' analyses have subtle but critical differences. In Algorithm 3, the noise is added to column correlations (that are collaboratively computed by the clients), and then the private model and the aggregate correlation between columns and the residual (*i.e.,* the gradient) are both computed by the server. In Algorithm 4, the clients first compute their individual gradients and then collaboratively find the average private gradient. The server computes the private model similarly to in Algorithm 3 but then shares this model with the individual clients. Here, the clients then recompute their gradients for the next step. Algorithm 3 adds the noise to the correlations of the gradient, which are then multiplied. On the other hand, in Algorithm 4, the bulk of the noise value is added outside of the correlations, directly to the gradient.

Although our analysis idea is similar to several OMP proofs in the literature (such as Chen & Caramanis (2012)), we have to redo the analysis because the way with which we add noise to the system (to ensure DP) is completely different. Notably, Chen & Caramanis (2012) adds noise to the measurement matrix $\boldsymbol{X}$, which has several issues. First, it does not even protect the privacy of the response $y$. Second, the noise variance is of the order-$np$, which is too high and likely leads to poor estimation accuracy. Indeed, the estimation error bound (Corollary 8) in Chen & Caramanis (2012) is given by $||\boldsymbol{\alpha}_* - \hat{\boldsymbol{\alpha}}||_2 = \mathcal{O}(\frac{(\sigma_w + \sigma_w^2)||\boldsymbol{\alpha}_*||_2 \sqrt{s \log p}}{n})$, where $\sigma_w$ is the standard deviation of the noise matrix $\boldsymbol{W^T W}$ such that $\boldsymbol{W}$ is the additive noise added to the original matrix $\boldsymbol{X}$. From their analysis, the term $\sigma_w + \sigma_w^2 = \mathcal{O}(p^2)$ when $\sigma_w$ is set to the DP-compliant noise value. The above error bound, therefore, will be huge when $p \gg n$, negating its usefulness in the high-dimensional regime. The experimental results in Chen & Caramanis (2012) assume small values of $\sigma_w$, not at the level of $\mathcal{O}(p)$ needed for achieving DP. Third, Chen & Caramanis (2012) does not provide a condition on the RIC. They assume that the matrix elements are subgaussian i.i.d.; thus, their model implicitly determines the RIC constant in their results. In contrast, we allow an arbitrary design matrix $\boldsymbol{X}$ (irrespective of the matrix's data distribution) that satisfies an RIP condition. Thus, it is important to quantify the condition of RIC to guarantee sparse recovery.

# 7 Empirical Results

In this section, we compare our proposed algorithm *SPriFed-OMP* with a version of the *DP-SGD* algorithm (Abadi et al., 2016) that uses $L_1$ regularization (Zhao & Yu, 2006). The first set of experiments is based on a synthetic data set so that we can freely vary the sample size $n$, the model dimension $p$, and the model sparsity $s$. Given $p$ and $n$, we first generate $\boldsymbol{X}$ with *i.i.d.* elements, each of which is from $\mathcal{N}(0,1)$. Given the model sparsity $s$, we then generate the non-zero coefficients of the model parameters $\boldsymbol{\alpha}_*$, each of which is from the distribution $\mathcal{N}(2,1)$. We then add additive error with mean zero and standard deviation $\sigma_\epsilon = 0.001$ to generate the response $y$. We then clip (to 1 or $-1$) every element in $\boldsymbol{X}$ and $\boldsymbol{y}$ with magnitude greater than 1 and then re-scale the whole matrix/vector appropriately to maintain their unit variance as required by the RIP condition 1.

## 7.1 Warm-up Experiments: Intuition behind our proposed enhancements

Before moving on to the main experimental results, we revisit the enhancements noted in Section 3. Note that both experiments below use the synthetic data setup above.

**Importance of Enhancement 1: Adding lower noise to selected features**

Here, we test the sparse basis recovery capabilities of Algorithm 3 on a synthetic toy dataset. We compare Algorithm 3 with another modified version, where Enhancement 1 is removed, i.e., we directly use the artifacts privatized by the higher order noise in lines *5* and *11* to estimate the private model. Thus, the private model will now have noise of order $\mathcal{O}(\sqrt{p})$ instead of $\mathcal{O}(\sqrt{s})$. In Table 1, we observe worse test MSE and poor sparse basis recovery performance when enhancement 1 is removed. Thus, we conclude that we need to re-privatize, *i.e.,* add lower-order noise to the correlations/gradients computed on the chosen feature set to obtain competitive sparse basis recovery in the DP-FL case.

**Importance of Enhancement 2: Privatizing the gradient**

| Samples | Test MSE | | Number of samples recovered | |
|---|---|---|---|---|
| | *SPriFed-OMP* (low noise) | *SPriFed-OMP* (high noise) | *SPriFed-OMP* (low noise) | *SPriFed-OMP* (high noise) |
| $n = 2000$ | 0.83 | 335.84 | 1 | 0 |
| $n = 4000$ | 0.52 | 6.45 | 2.67 | 1.67 |
| $n = 8000$ | 0.35 | 0.77 | 7.67 | 5.67 |

Table 1: **Enhancement 1 Performance Improvement:** Comparison of Test MSE and basis recovery performance for when Algorithm 3 runs with and without enhancement 1. The experiments are run over 3 randomized trials over the synthetic setup (above) with $p = 10000$, $s = 10$, and a privacy guarantee of $(5.74, 10^{-4})$ DP.

In this experiment, we look closely at the two kinds of artifacts released by Algorithms 3 and 4, *i.e.* the correlations and gradients respectively. We measure the change in the artifacts' value over iterations by measuring and plotting the total $l_2$ norm values for each artifact. To do so, we run a non-private version of Algorithm 3 and plot the changing correlation norms and gradient $l_2$ norms over multiple iterations. Here, in each iteration we use the correlation value with respect to the current newly chosen feature $j$ (referred in Figure 1's legend). We observe from Figure 1 that the gradient norm values decrease over time while most of the correlation norms are constant. As expected from the way OMP is designed, only the value of the correlation norm $\|\boldsymbol{X_j^T y}\|_2$ decreases over iterations since the newly chosen features have lower correlation values than previous features. Thus, we conclude that we can employ more aggressive clipping in Algorithm 4 than in Algorithm 3. That is, for gradients, a lower clipping bound should work well, especially toward the final rounds, while such lower clipping bounds will significantly affect the correlations irrespective of the iteration.

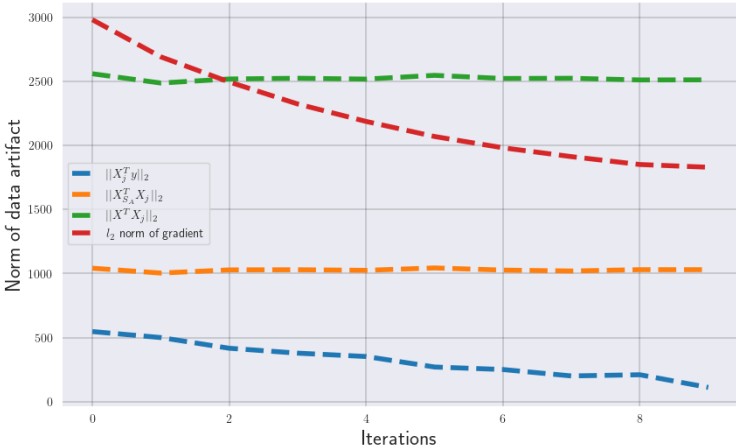

Figure 1: **Enhancement 2 Performance Intuition:** Change in size of artifacts (correlations and gradient) measured by their norm over multiple iterations.

## 7.2 Performance Comparison: Synthetic Data Sets

In the rest of the section, we will compare *SPriFed-OMP* with various baselines. We first discuss the experimental setup and the baselines. In *SPriFed-OMP* (Algorithm 3), we set the parameter $\mu_p$ to be either 0.4 (for $s = 10$) or 0.543 (for $s = 5$) and set the parameter $\mu_s$ as 0.02. The cumulative privacy cost $(\epsilon, \delta)$ for all the experiments is set to at most $(5.34, 10^{-4})$ (for DP) or equivalently 1.32 (GDP). Note that *SPriFed-OMP* does not require hyperparameter tuning besides choosing the privacy parameters $\mu_s$ and $\mu_p$. Since $p \gg s$, we can easily tolerate $\mu_s$ much smaller than $\mu_p$. Thus, the $\mu_p$ parameter will primarily dominate the privacy cost.

The first baseline *DP-SGD* is implemented to minimize the $L_1$-regularized mean square error. After the algorithm is terminated, we pick the top-$s$ elements and leverage those to compute the final model and the test MSE. One requirement for DP-SGD is that we also need to choose the hyper-parameters, which are the

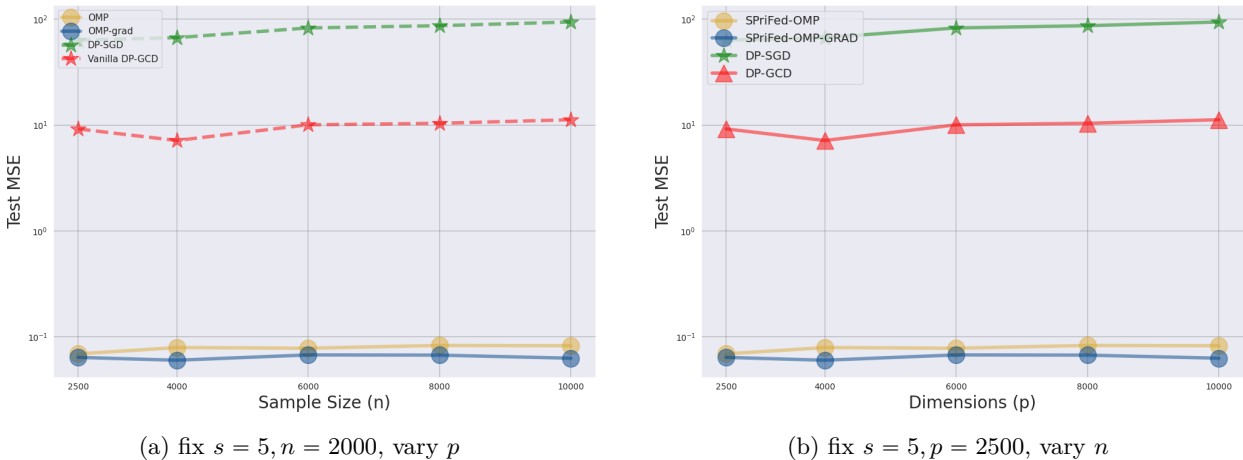

(a) fix $s = 5, n = 2000$, vary $p$          (b) fix $s = 5, p = 2500$, vary $n$

Figure 2: Test MSE is shown for both *SPriFed-OMP* and *DP-SGD* for privacy parameters $(4.94, 10^{-4})$. Figure (a) fixes the sample size and varies the model dimensions; Figure (b) fixes the model dimensions and varies the sample size. Measurements averaged over 3 randomized simulation runs.

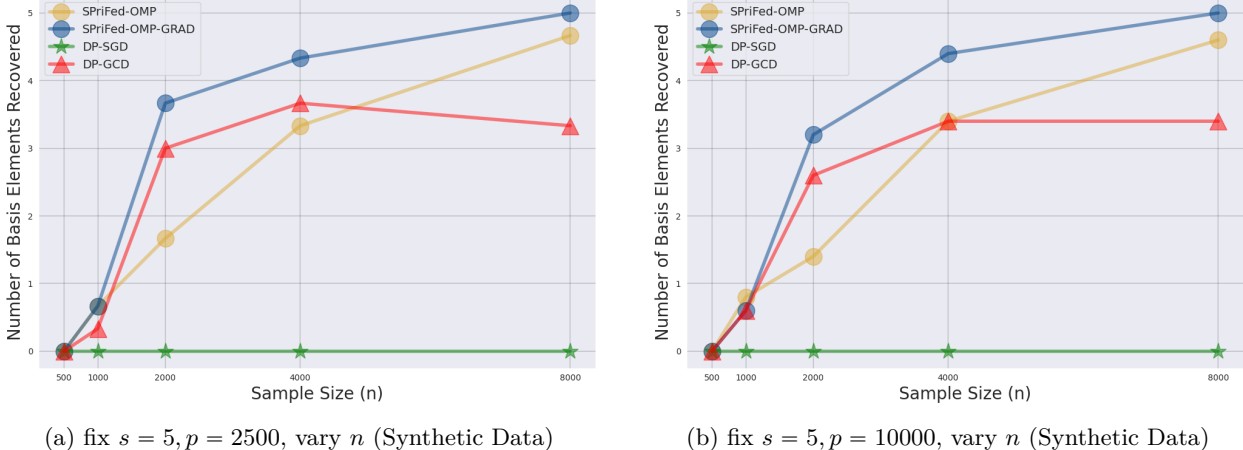

(a) fix $s = 5, p = 2500$, vary $n$ (Synthetic Data)          (b) fix $s = 5, p = 10000$, vary $n$ (Synthetic Data)

Figure 3: The number of basis correctly recovered by *SPriFed-OMP* for privacy parameters $(4.94, 10^{-4})$. We choose $s = 5$. Figure (a) demonstrates basis recovery for model sparsity $p = 2500$ over varying sample sizes. Figure (b) demonstrates basis recovery for model sparsity $p = 10000$ over varying sample sizes. Measurements averaged over 3 randomized simulation runs.

coefficient for the $L_1$-regularization term and the learning rate. Although hyper-parameter tuning may incur an additional privacy cost for DP-SGD, here, to benefit DP-SGD, we perform the hyper-parameter tuning non-privately using the Optuna library(Akiba et al., 2019). To maintain a standardized notion of privacy definition, we also study the composed privacy cost of DP-SGD using GDP. In particular, for achieving GDP in DP-SGD, we set the privacy parameter $\mu_{DP-SGD}$ as either 0.4 (when $s = 10$) or 0.543 (when $s = 5$). $\mu_{DP-SGD}$ matches the $\mu_p$ parameter from *SPriFed-OMP*, which is also being used to privatize $p$-dimensional vectors. DP-SGD stops training when the overall privacy budget matches that of *SPriFed-OMP*.[5]

The second baseline DP-GCD is based on a modified DP-FL compatible version of the algorithm presented in Mangold et al. (2023). Algorithm 1 in Mangold et al. (2023) primarily follows from the Gradient Coordinate Descent (GCD) algorithm. Like DP-SGD, although DP-GCD does not provide performance guarantees on sparse basis recovery, it remains a natural algorithm when used along with $l_1$-regularization objectives.

---

[5]Although $\mu$-GDP alone provides an intuitive privacy definition, for completeness, we also convert the final $\mu$-GDP privacy parameter to the approximate DP parameters using the AutoDP Python library (Wang, 2023).

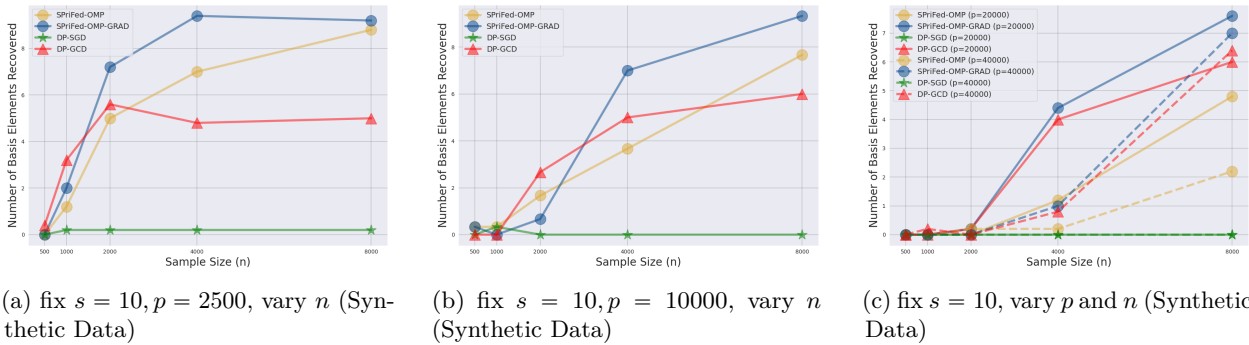

(a) fix $s = 10, p = 2500$, vary $n$ (Synthetic Data)

(b) fix $s = 10, p = 10000$, vary $n$ (Synthetic Data)

(c) fix $s = 10$, vary $p$ and $n$ (Synthetic Data)

Figure 4: The number of basis correctly recovered by *SPriFed-OMP* for privacy parameters $(5.34, 10^{-4})$. We choose $s = 10$, and figures (a) and (b) have an additive error with standard deviation $\sigma_\varepsilon = 0.001$. Figure (a) demonstrates basis recovery for model sparsity $p = 2500$ over varying sample sizes. Figure (b) demonstrates basis recovery for model sparsity $p = 10000$ over varying sample sizes. Figure (c) varies $p = 20000$ and $p = 40000$ while increasing the system model's additive error's standard deviation to $\sigma_\varepsilon = 0.1$. Measurements averaged over 3 randomized simulation runs.

However, DP-GCD in Mangold et al. (2023) considered DP only in the server setting but not the DP-FL setting as in this paper. Thus, we make an effort to compare with it by first converting it to a DP-FL compatible version. We first present our converted DP-GCD algorithm (Algorithm 5). Here, in each iteration, the maximum gradient is picked privately in lines *5-8*, and the model takes a step in only the direction of this chosen gradient element in lines *10-11*. Such gradient-based optimization steps are repeated until the model converges or the privacy budget is exhausted.

In our modified version presented in Algorithm 5, we also include two enhancements to the DP-GCD algorithm from Mangold et al. (2023). First, we convert DP-GCD to be DP-FL compatible by leveraging the NOISY-SMPC mechanism while identifying the feature with the maximum gradient (*lines 5-8*). We further ensure that much lower noise is added while following the gradient-based approach in DP-GCD by separately re-privatizing (in *line 10*) the feature gradient chosen via the maximum gradient mechanism. Thus, in summary, our proposed modification of DP-GCD (Algorithm 5) significantly enhances the private version in Mangold et al. (2023) by making it DP-FL compatible while also ensuring that it has traits suitable for sparse basis recovery. Similar to DP-SGD, we again perform hyper-parameter tuning non-privately using the Optuna library (Akiba et al., 2019). For both DP-SGD and DP-GCD, we do not include the cost of tuning these hyperparameters in the privacy cost, and hence, the results we report below are more optimistic for these two algorithms. Below we discuss our empirical findings.

**Fix $n$, Change $p$:** In Fig. 2(a), we fix $s = 5$ and $n = 2000$ and vary $p \in \{2500, 4000, 6000, 8000, 10000\}$. With rising $p$, we observe that the test MSE loss increases much more dramatically for DP-SGD (note the logarithmic y-axis) than *SPriFed-OMP*. In fact, *SPriFed-OMP*'s performance is almost constant as $p$ varies since its empirical risk (Theorem 10) is mostly dependent on $s$. Further, although not shown in the figure, *SPriFed-OMP* extracts most of the basis elements ($\geq 3$) even when $p = 10000$. Finally, even though the sample size can be quite small in comparison to the model dimension, we obtain highly accurate models for *SPriFed-OMP* with test MSE close to zero, which is in line with our theory (see Theorems 9 and 10). Similarly, even though DP-GCD performs better than DP-SGD in terms of test MSE (thanks to the lower noise that our enhanced DP-GCD algorithm adds), its test MSE is still much worse than SPriFed-OMP. We will discuss this limitation of DP-SGD in the latter part of this section.

**Fix $p$, Change $n$:** In Fig. 2(b), we set $s = 5$ and $p = 2500$ and vary $n \in \{400, 800, 1200, 1600, 2000\}$. We observe that when $p \gg n$, the performance of all algorithms is impacted. However, increasing the sample size improves performance more dramatically for *SPriFed-OMP* and *SPriFed-OMP-GRAD* (again, note the logarithmic scale on the y-axis). Notably, *SPriFed-OMP* and *SPriFed-OMP-GRAD* significantly outperform DP-SGD and DP-GCD for all choices of $n$. We specifically note that the test loss for both versions of

---

**Algorithm 5** Enhanced DP-GCD (DP-FL)

---

1: **Input Parameters: Data:** $\boldsymbol{X} = \{\boldsymbol{X_1}, \boldsymbol{X_2}, ..., \boldsymbol{X_p}\} \in \mathbb{R}^{n \times p}$ where $X_i$ is the $i^{th}$ feature, **Response:** $\boldsymbol{y} \in \mathbb{R}^n$, **Learning Step Size:** $\gamma$, **Total Steps:** $T$, **Privacy Parameters:** $\sigma_1 = \frac{1}{\mu_p}$, **Initialize Model:** $\tilde{\alpha} = \bar{0}$, **Initialize Basis:** $S = \varnothing$, **Gradient Bound:** $C$ (true bound over all gradients or bound achieved via gradient clipping).

2: **Output:** Predicted Sparse Model $\tilde{\alpha}$, Predicted Sparse Basis §

3: **procedure** DP-GCD($\boldsymbol{X}, \boldsymbol{y}, n, T$)

4:     **for** $t = 1 : T$ **do**

5:         **for** $j = 0 : p$ **do**

6:             $g_j \leftarrow$ **NOISY-SMPC**$(X_j, y - X_{S_\mathcal{A}}\tilde{\alpha}, n, \sigma_1 C \sqrt{p})$

7:         **end for**

8:         $j^* \leftarrow \arg\max_j |g_j|$

9:         $S \leftarrow S \cup \{j^*\}$

10:         $g'_{j*} \leftarrow$ **NOISY-SMPC**$(X_{j*}, y - X_{S_\mathcal{A}}\tilde{\alpha}, n, \sigma_1 C)$

11:         $\tilde{\alpha}[j^*] = \tilde{\alpha}[j^*] - \gamma g'_{j*}$

12:     **end for**

13:     Return $\tilde{\alpha}, S$

14: **end procedure**

---

*SPriFed-OMP* gets very close to zero (as $n$ increases), which matches our theoretical results (Theorems 9 and 10).

**Fix $s$, Change $p$ and $n$:** In Figs. 3 and 4, we report the number of non-zero basis in $\alpha_*$ that are correctly recovered by the proposed algorithms. We show two plots, Fig. 3 for $s = 5$ and Fig. 4 for $s = 10$. The sample sizes are varied over $\{500, 1000, 2000, 4000, 8000\}$ and the model dimensions are varied over $\{2500, 100000\}$ for both $s = 5$ and $s = 10$. Additionally, we also vary p over $\{20000, 400000\}$ for the $s = 10$ case. First, we can see for correctly recovering majority of the sparse basis, the value of $n$ increases much slower than $p$. Let us examine more closely the cases when we vary $s$ as either 5 or 10. We first look at the case when $s = 5$. For $p = 2500$ by setting $n = 2000$, we manage to recover 3+ out of 5 basis elements with *SPriFed-OMP-Grad*. It turns out that even when $p$ is increased to 10000, using $n = 2000$ is still sufficient to recover 3 out of 5 basis. When $s = 10$, we consider the number of samples required to recover 7+ out of 10 basis elements. For $p = 2500$, $n = 2000$ suffices. But when $p$ is increased 4 times to 10000, we only need $n$ to be doubled to 4000. Finally, when $p$ is either 20000 or 40000, we only need $n = 8000$ samples to recover the same number of basis elements. In summary, we find that, as the model dimension and the model sparsity increase, the sample size required for sparse recovery increases as well, which is consistent with our main results, Theorems 7 and 8. More interestingly, the value of $n$ increases much slower than that of $p$. This aligns with our theoretical result that $n$ only needs to be order-$\sqrt{p}$.

Second, as seen while comparing Figures 4*(a-b)* with Figure 4*(c)*, increasing the additive error variability affects *SPriFed-OMP* the most, while both the gradient-based methods *SPriFed-OMP-Grad* and DP-GCD are minimally affected. Note that the results shown in Figures 4*(a-b)* have system error standard deviation of 0.001. Here, we can see that the performance of *SPriFed-OMP* is quite comparable to both *SPriFed-OMP-GRAD* and *DP-GCD*. *SPriFed-OMP* even manages to outperform *DP-GCD* for the higher sample size range. However, Figure 4*(c)* uses a standard deviation of 0.1. Here, the performance of *SPriFed-OMP* dips significantly in comparison to both *SPriFed-OMP-GRAD* and *DP-GCD*. Thus, error variability affects *SPriFed-OMP* more than the other methods. These results are in line with our theory (refer Theorems 7 and 8) that shows that *SPriFed-OMP* has stricter system error requirements compared to *SPriFed-OMP-GRAD*.

Let us now discuss the sparse basis recovery performance of the two baselines. Note that both DP-GCD and DP-SGD converge to a private model by leveraging gradient-based optimization. However, DP-GCD adds much lower noise than DP-SGD since, at each round, it only updates a single dimension rather than all dimensions as done by DP-SGD. This explains why we see a better performance for DP-GCD than DP-SGD. However, DP-GCD still adds significant noise to the model compared to both *SPriFed-OMP*

and *SPriFed-OMP-GRAD* due to the following two reasons. First, due to its gradient-based optimization, DP-GCD may take more than one step to optimize any given direction fully. In contrast, for each new basis, SPriFed-OMP only collects new correlation values (or gradients) once. Thus, depending on the number of steps taken, the amount of noise added by DP-GCD per dimension is often much higher. For instance, if the same dimension is picked $t$ times, then the noise added will be of $t$ orders higher for DP-GCD than the *SPriFed-OMP* class of methods. The remark in this section covers further details. Second, unlike *SPriFed-OMP* and *SPriFed-OMP-GRAD*, DP-GCD does not compute the exact true value of the model (using the OLS formula) in each round rather than choosing to add noise to the previously predicted model continually and thus the noise added in previous rounds might affect the predicted model significantly. In contrast, private OMP estimates the model parameters $\alpha$ directly using the OLS formula (Algorithm 2) based on the re-privatized correlation values in Lines *(14))* and *(9)* of Algorithms 3 and 4 respectively, which have much lower noise. Thus, this noise accumulation effect is greatly reduced.

*Remark (Significant difference of Test MSE between DP-SGD, DP-GCD and both flavors of SPriFed-OMP):* From Figures 2(a) and 2(b), we observe that the test MSE of *SPriFed-OMP* and DP-SGD are of different orders (note the logarithmic scale on the y-axis). This difference is because, in *SPriFed-OMP*, we only introduce noise with a variance of order-$s$ to the final model. In contrast, DP-SGD adds noise of order-$p$ variance to all its model dimensions. Therefore, even though DP-SGD selects a subset of these dimensions for the final model, the considerable noise significantly impacts the resulting model coefficients. As a result, the test MSE for DP-SGD is much worse than the one we obtain from *SPriFed-OMP*. Once more, we note that due to such a high noise variance, DP-SGD cannot recover any of the true basis in our experiments. For the case of DP-GCD, the noise added to the model is of order $s$ as well. However, DP-GCD follows a gradient-based iterative approach where noise is added to the gradient. Then, this noise is passed to the model based on the feature selected in that step, the learning rate, and the number of total steps taken. Unlike DP-GCD, the *SPriFed-OMP* flavors add noise to the exact value of the true model based on the currently selected features. Thus, overall, *SPriFed-OMP* only adds noise via the privacy mechanism. On the other hand, DP-GCD adds a similar noise via the privacy mechanism while also including a bias term based on the difference between the DP-GCD model and the optimal linear model for the chosen feature set.

*Remark (Run-time trends of algorithms):* To understand the general trend of computational time required by each algorithm, we consider the sample case when $n = 2000, s = 20, p = 20000$. Here we provide the CPU time as measured by the time library in Python. We note that SPriFed-OMP takes 20.17 seconds, SPriFed-OMP-Grad takes 26.38 seconds, DP-SGD takes 11.99 seconds, and DP-GCD takes 9.56 seconds. As expected, the OMP algorithms take longer due to their separate computations for the sparse model in each step. In particular, SPriFed-OMP-Grad takes the longest since it needs to compute the gradient separately for each sample, sum them up in a DP-FL fashion (like DP-SGD and DP-SGD), and compute the sparse models separately. Finally, DP-SGD takes a bit longer than DP-GCD since, in each step, it operates over all possible features, while DP-GCD only runs on the feature with the maximum gradient. Note that we expect when the number of clients is fewer, the gradient computations might be speedier. However, we will consider optimizing these algorithms and exploring a lengthier ablation study as part of future work.

## 7.3 Performance Comparison: More Realistic Data Sets

In the second set of experiments, we test our proposed algorithms on the synthetic MADELON dataset from the NIPS 2003 challenge (Guyon et al., 2004), a popular dataset used in the sparse recovery literature. The original MADELON dataset has 20 true features, but only 5 of these are true raw features, and the remaining 15 are linear combinations of these raw features. Its total model dimension is $p = 500$, with $n = 2000$ training samples. However, since we are only interested in the $p > n$ case, we randomly sub-sample 300 samples in each trial. We run 5 randomized trials and report the average results. Here, we set $\mu_p = 0.45, \mu_s = 0.09$ for *SPriFed-OMP* and $\mu_{DP-SGD} = 0.45$ for DP-SGD. The corresponding DP guarantee is $(8.38, 10^{-3})$, and the GDP guarantee is 1.72. For *SPriFed-OMP*, we clip the elements of $\boldsymbol{X}$ to 0.12 and elements of $\boldsymbol{y}$ to 0.36. We clip the gradient elements to 1. As expected, DP-SGD performs the worst with a high Test MSE of *105.73*. DP-GCD performs reasonably well with Test MSE of *1.23*. Finally, *SPriFed-OMP* has a Test MSE of *1.55* while *SPriFed-OMP-GRAD* handily outperforms DP-GCD with a Test MSE of only *0.27*. We attribute the relatively better performance of DP-GCD for this data set to two reasons, (1) proper hyperparameter

tuning of DP-GCD; thus, we expect that the performance of DP-GCD would be worse if the privacy cost of hyperparameter tuning must also be accounted for; and (2) due to the synthetic nature of MADELON, DP-GCD might perform well even when it does not pick the exact best features (since MADELON has 15 features that are linear combinations of the true 5 features).

In the third set of experiments, we show results on two real high-dimensional datasets from the repository presented in Drysdale (2022). Here, we consider the high-dimensional datasets chop and gse1992. The chop dataset contains 414 total samples with 3836 features, and gse1992 contains 124 samples with 15537 features. Similar to MADELON, we sub-sample the dimensions of chop to be 2000 and that of gse1992 to be 500 in each trial. The rest of the privacy setup is identical to MADELON. We present the results for these datasets in Table 2. We note that both *SPriFed-OMP* and *SPriFed-OMP-GRAD* outperforms all other algorithms over both datasets with *SPriFed-OMP-GRAD* providing the best utility. As previously noted in the remark above, we attribute the success of both versions of *SPriFed-OMP* to their ability to add significantly lower order noise to the true model value while leveraging the true value of the model using the ordinary least squares formula. Note that DP-GCD also adds lower-order noise to the model. However, unlike both flavors of *SPriFed-OMP*, DP-GCD does not add noise to the exact value of the true model, instead relying on a gradient-based approach that iteratively adds noise to the gradient. The gradient-based noise is then passed to the model value based on the feature selected and the learning rate.

| Algorithm | chop | gse1992 |
|---|---|---|
| *SPriFed-OMP* | 0.101 | 1.90 |
| *SPriFed-OMP-GRAD* | 0.154 | 1.15 |
| DP-GCD | 18.79 | 525.60 |
| DP-SGD | 546.76 | 988.72 |

Table 2: **Test MSE Performance:** We observe that without model/basis selection, both DP-SGD and DP-GCD are required to add substantial noise in the models, leading to much higher Test MSE than both *SPriFed-OMP* and *SPriFed-OMP-GRAD*. However, both *SPriFed-OMP* and *SPriFed-OMP-GRAD* enjoy close to optimal test MSE.

### 7.4    Ablation study of privacy-utility trade-off

To understand the impact of the privacy parameters on the algorithm's performance, we next study the test error and the number of bases recovered for both SPriFed-OMP and SPriFed-OMP-Grad, as we vary the net epsilon value. We first consider the synthetic datasets of Section 7.2 for the case of $n = 2000, s = 5$, and $p = 10000$. Both algorithms are run for 3 random seeds. The value of $\mu_p$ is varied over $[0.1, 0.2, 0.3, 0.4, 0.5, 0.8, 1]$, which in turn affects the final $\epsilon$ value. Note that $\mu_s = 0.09$ is constant since it does not greatly affect the net $\epsilon$ value. In Fig. 5(a) and (b), we plot both algorithms' test MSE and the basis recovery capabilities vs $\epsilon$. We can observe that, as $\epsilon$ increases, the performance of both algorithms improves (i.e., the test MSE decreases, and the number of correctly recovered bases increases). Both algorithms generally reach constant performance for any  *epsilon* larger than 5 (which is around the value chosen in our results reported in Section 7.3).

We then perform a similar study of privacy-utility trade-off on the realistic chop dataset. We consider the following setup where $\mu_p$ is again varied over $[0.1, 0.2, 0.3, 0.4, 0.5, 0.8, 1]$ to adjust the net privacy budget. We run both algorithms over 7 random seeds and set the clipping bound for elements of $X, y$ to be 0.5. $\mu_s = 0.15$. Since we do not know the underlying basis for the chop dataset, we simply report the Test MSE error in Fig. 5c. We can see that as $\epsilon$ increases, the error generally tends to lower. However, we observe that the error dips the most from $\epsilon \in [6, 12]$. Thus, we pick $\epsilon \approx 8$, which gives a reasonable privacy-utility trade-off for a $\epsilon < 10$.

*Remark (Sub-sampling with DP-FL):* We recognize that under a DP-FL setting, sub-sampling is beneficial in reducing per-round communication costs and amplifying privacy. However, the benefit of sub-sampling in the $n \ll p$ regime that we focus on in this paper has not been thoroughly studied, which could be an interesting direction for future work.

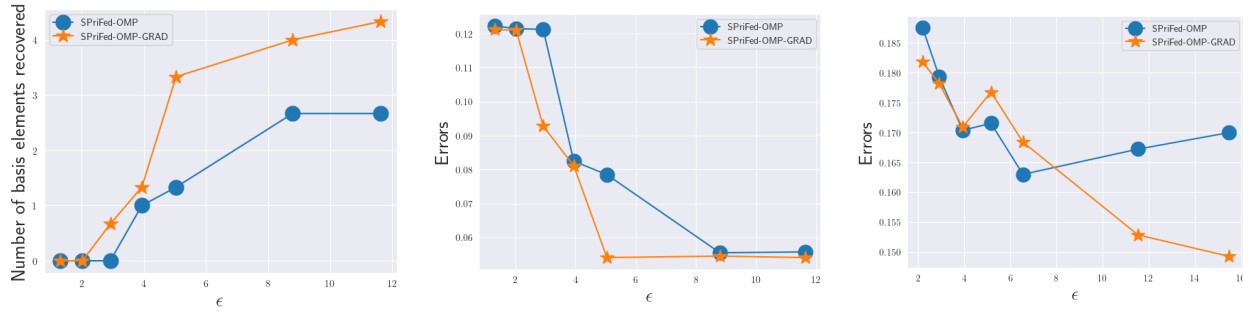

(a) Basis Recovery-privacy trade-off (Synthetic Data)

(b) Error-privacy trade-off (Synthetic Data)

(c) Error-privacy trade-off (Chop Dataset Drysdale (2022))

Figure 5: Ablation study for finding privacy-utility trade-off. We look at the impact of choosing varying privacy parameters on the accuracy of the proposed algorithms. Figures (a) and (b) study the basis recovery, and Test MSE values varied over changing values of the overall privacy budget $\epsilon$, respectively, for the synthetic data with $n = 2000, s = 5, p = 10000$. Figure (c) provides the Test MSE of the chop dataset again varied over the net privacy budget $\epsilon$. $\epsilon$ is implicitly modified by adjusting the privacy parameter $\mu_p$.

## 8 Conclusion

In this work, we propose two new private sparse basis recovery algorithms called *SPriFed-OMP* and *SPriFed-OMP-GRAD* for the DP-FL setting based on the Orthogonal Matching Pursuit (OMP) algorithm. Specifically, we prove analytically that both algorithms can recover the sparse basis in a finite number of steps. Further, we bound the privacy cost while leveraging the obtained low-dimensional model to obtain dimension-free model estimation error and empirical risk. Here, *SPriFed-OMP* is our first attempt at leveraging OMP for sparse basis recovery in the DP-FL setting. In contrast, *SPriFed-OMP-GRAD* improves upon *SPriFed-OMP* (both analytically and empirically) by requiring a simpler DP-FL mechanism, fewer samples, and a less complicated analysis. To the best of our knowledge, these are the first DP-FL algorithms that successfully manage sparse recovery under the RIP assumption while only requiring $n = \mathcal{O}(\sqrt{p})$ samples. Our experimental results on both synthetic and real datasets strongly corroborate our theoretical analysis. For future work, we will explore DP-FL algorithms that can perform sparse recovery even without the RIP assumption.

## 9 Acknowledgement

This work has been partially supported by NSF grants CNS-2113893 and CNS-2225950.

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

# A    Differential Privacy

Differential privacy was introduced to privatize changes caused by individual clients (Dwork et al., 2014). For a given function, differential privacy obscures the maximum possible change in the function's output due to the inclusion or removal of a single client's data. Here, obscuring is allowed by introducing noise and, thus, a form of deniability in the function's output. Deniability allows an individual client to deny its presence in the dataset and disallows an adversary from linking the client or its data to a non-privatized record. Differential privacy is significantly stronger than previous anonymization methods as it works even in the worst-case scenario when the entire dataset is publicly released except for a single client's data. Here, we look at the precise mathematical definition of differential privacy, simple techniques to introduce privacy, and its additional properties.

**Definition A.1.** *[**Approximate Differential Privacy**] For any two neighboring datasets $X, X' \subseteq \mathcal{S}$ differing over a single sample, we say that the randomized mechanism $\mathcal{M} : \mathcal{S} \to \mathcal{R}$ is $(\epsilon, \delta)$ differentially private if,*

$$Pr[\mathcal{M}(X) \in \mathcal{R}] \leqslant e^{\epsilon} \cdot Pr[\mathcal{M}(X^{'}) \in \mathcal{R}] + \delta$$

**Definition A.2** ($L_2$ sensitivity of a function)**.** *Given a deterministic function $f$, consider all possible pairs of conformable neighboring datasets in the domain of $f$ differing on a single row. Then we denote $f$'s $L_2$ sensitivity as*

$$\Delta_2 f := \max_{x,x \in dom(f), ||x} ||f(x) - f(x')||_2$$

*Thus, $\Delta_2 f$ records the maximum possible change in $f$ due to a change in a single row. Often each row belongs to a different client and thus, $\Delta_2 f$ will record the maximum change due to a change in a client or the inclusion or removal of a client from the dataset.*

We can achieve differential privacy by the Laplacian, Exponential, and Gaussian mechanisms (Dwork et al., 2014). Here, we use the Gaussian mechanism as it performs better for multi-step (or composed) privacy algorithms. In particular, we leverage the Gaussian Differential Privacy (GDP) composition method for optimally tracking the privacy loss over multiple steps (Dong et al., 2019). Although, other composition methods such as advanced composition (Kairouz et al., 2015), Renyi DP (Mironov, 2017), and concentrated DP (Bun & Steinke, 2016) exist, we chose GDP for its superior composition performance for Gaussian mechanisms. For completeness, we state the connection between the Gaussian mechanism and the GDP privacy constant.

**Lemma 11.** *[GDP Gaussian mechanism (Theorem 2.7 from (Dong et al., 2019))] Let statistic $f(\cdot)$ be computed over the dataset $\boldsymbol{X}$. Then, the randomized Gaussian mechanism $\mathcal{M}(\boldsymbol{X}) = f(\boldsymbol{X}) + \eta$ where $\eta \sim \mathcal{N}(0, \frac{\Delta_2 f^2}{\mu^2})$ is $\mu - GDP$.*

**Lemma 12.** *[Restated Theorem 2.7 from Dong et al. (2019)] Let statistic $f(\cdot)$ be computed over the dataset $\boldsymbol{X}$. Then, the randomized Gaussian mechanism $\mathcal{M}(\boldsymbol{X}) = f(\boldsymbol{X}) + \eta$ where $\eta \sim \mathcal{N}(0, \frac{\Delta_2 f^2}{\mu^2})$ is $\mu - GDP$.*

An important property of DP, known as post-processing maintains that no amount of processing on a differentially private system will cause the new output to leak any more information than the unprocessed input. This is expressed clearly in Lemma 13.

**Lemma 13** (Differential Privacy Post-Processing)**.** *Consider a randomized mechanism $M$ that is $(\varepsilon, \delta)$ differentially private. Now, let $M_{post}$ be another randomized mechanism that can be applied to the output of $M$. Then, we state that the output of the combination of $M_{post}(M(\cdot))$ function group has at least differential privacy guarantee of $(\varepsilon, \delta)$ or higher.*

## B  Compressed Sensing Theory

Let $\Omega = \{1, 2, ..., p\}$ be the overall feature index set and the support $S \subseteq \Omega$ of $\beta^*$ be defined such that, $S = \{i | i \in \Omega, \beta_i^* \neq 0\}$. By definition, $|S| \leqslant s$ (where $s$ is the sparsity index of the dataset). For a vector $v \in \mathbb{R}^p$, let us define $v_W$ as the restriction of $v$ to the indices in $W$ where $W \subseteq \Omega$. Similarly, for a matrix $Q \in \mathbb{R}^{n \times p}$ let us define the sub-matrix $Q_W \in \mathbb{R}^{n \times |W|}$ s.t. it only contains columns indexed by $W(\subseteq \Omega)$. Our primary assumption for the data matrix follows the Restricted Isometry Property (RIP) as detailed in Assumption 1.

Intuitively, assumption 1 for $X$, a matrix satisfying RIP of order $K$ states that the eigenvalues for all $K$-dimensional (or smaller) sub-matrices derived from $X$, lie in a tight interval around 1. Interestingly, the assumption holds for several popular sub-Gaussian random matrices. Construction of such random RIP matrices is detailed in Theorem 5.65 from Vershynin (2010). We note that although we use the random matrix construction as a reference for our primary analysis, we only require a matrix (even deterministic) normalized by $\sqrt{n}$ that satisfies the RIP condition and and it should contain (probably) bounded values. Further details are omitted due to page limits.

**Lemma 14.** *[Construction of Random Matrices satisfying RIP] (Theorem 5.65 from Vershynin (2010)) Let $A \in \mathbb{R}^{n \times p}$ be a subgaussian random matrix with independent and isotropic rows. Then the normalized matrix $\frac{A}{\sqrt{n}}$ satisfies the following for every sparsity level $1 \leqslant k \leqslant p$ and every number $\delta \in (0, 1)$,*

$$if\,n \geqslant \frac{Ck \log(\frac{ep}{k})}{\delta^2} \rightarrow \delta_k(\frac{A}{\sqrt{n}}) \leqslant \delta$$

*with a probability at least $1 - 2e^{-c\delta^2 n}$. Here, $C = C_K, c = c_K > 0$ depend only on the subgaussian norm $K = \max_i ||A_i||_{\Psi_2}$ of the rows of $A$. Here, $k$ is the RIP order, and $\delta_k(\cdot)$ is the RIC. The exact construction is demonstrated below.*

One instance of such a random matrix satisfying RIP, $X \in \mathbb{R}^{n \times p}$, can be constructed with the following steps,

1. For a given row $v$ in $X$, independently sample the row's elements from a sub-Gaussian distribution s.t. each row is isotropic. $v \in \mathbb{R}^p$ is an isotropic vector if $E[vv^T] = \mathbb{I}$.

2. The standard normal distribution satisfies the isotropic property. Therefore, assume $X_{ij} \sim \mathcal{N}(0, 1)$, for all $i \in [n], j \in [p]$. We note that alternative distributions as detailed in Vershynin (2010) also satisfy the isotropy property.

3. Normalize $X \rightarrow \frac{X}{\sqrt{n}}$.

Given an RIP satisfying matrix $X \in \mathbb{R}^{n \times p}$ satisfying RIP of order $s$ with an RIC of constant $\zeta_s$, we note some useful properties (Lemmas 15-18) derived from the RIP condition ((Candes & Tao, 2005)).

Consider the set of vectors $\mathcal{W} := \{w \in \mathbb{R}^p \mid ||w||_0 \leqslant s\}$. The support for vectors in $\mathcal{W}$ is $\mathcal{S}$. For a vector $w \in \mathcal{W}$ and by RIP and using $||X_S w_S||_2 = ||Xw||_2$,

$$
(1 - \zeta_s)||w||_2^2 \leqslant ||X_S w_S||_2^2 \leqslant (1 + \zeta_s)||w||_2^2
$$
$$
\rightarrow (1 - \zeta_s)||w||_2^2 \leqslant ||Xw||_2^2 \leqslant (1 + \zeta_s)||w||_2^2
$$
$$
\rightarrow (1 - \zeta_s) \leqslant \frac{w^T X^T X w}{w^T w} \leqslant (1 + \zeta_s) \tag{2}
$$

Thus, the following bound immediately follows,

$$
(1 - \zeta_s) \leqslant \sigma_{min}(X_S^T X_S), \sigma_{\max}(X_S^T X_S) \leqslant (1 + \zeta_s)
$$

**Lemma 15.** *[**Monotonocity of the Isometric Constant**] If a measurement matrix $X \in \mathbb{R}^{n \times p}$ satisfies RIP of orders $K_1$ and $K_2$ s.t. $K_1 \leqslant K_2$ then their corresponding RICs follow the same order* i.e., $\zeta_{K_1} \leqslant \zeta_{K_2}$.

**Lemma 16.** *[**Direct RIP Consequences**] Let $S \subseteq \Omega$ be the support of the true model $\beta^*$. If $\zeta_s < 1; s = |S|$ then we can show that for the measurement matrix $X$ satisfying RIP with order $s$ and for any $w \in \mathbb{R}^s$,*

$$
(1 - \zeta_s)||w||_2 \leqslant ||X_S^T X_S w||_2 \leqslant (1 + \zeta_s)||w||_2
$$
$$
\frac{1}{1 + \zeta_s}||w||_2 \leqslant ||(X_S^T X_S)^{-1} w||_2 \leqslant \frac{1}{1 - \zeta_s}||w||_2
$$

**Lemma 17.** *[RIP: Near Orthogonality] Let $S_1, S_2 \subseteq \Omega s.t., S_1 \cap S_2 = \varnothing$. If $\zeta_{|S_1|+|S_2|} < 1$ and $S_1 \cup S_2 = S$ then for the measurement matrix $X$ satisfying RIP of order $s$ and for $u \in \mathbb{R}^{|S_2|}$ we have*

$$
||X_{S_1}^T X_{S_2} u||_2 \leqslant \zeta_{|S_1|+|S_2|}||u||_2
$$

**Lemma 18.** *Let $\mathcal{S} \subseteq \Omega$ be the support of $\beta^*$ (our ground truth model parameter). If $\zeta_s < 1; s = |\mathcal{S}|$ then we can show for the measurement matrix $X$ satisfying RIP with order $s$ and for any $w \in \mathbb{R}^n$.*

$$
||X_{\mathcal{S}}^T w||_2 \leqslant \sqrt{1 + \zeta_s}||w||_2
$$

## C    Useful Results for Privacy Analysis

**Proof of Lemma 4** Consider two neighboring input-output pairs $X, y$ and $X', y'$ such that the pairs differ in a single sample at row $k$. We approach this proof by first considering the sensitivity of the $j^{th}$ column $(X_j)$ and then computing the combined sensitivity of the $p$ columns.

$$
\Delta_2(X_j y_j) = \max_{k \in [n]} ||(X_j')^T y' - X_j^T y||_2
$$
$$
\leqslant 2 \max_{k \in [n]} ||X_{jk} y_k||_2
$$
$$
\leqslant 2 \max_{k \in [n]} X_{jk} y_j \leqslant X_M y_M
$$
$$
\rightarrow \Delta_2(X^T y) \leqslant 2\sqrt{p} X_M y_M = \mathcal{O}(\sqrt{p})
$$

Similarly, let us consider the sensitivity of $X^T X_{S_{\mathcal{A}}}$,

$$
\Delta_2(X^T X_j) = \max_{k \in [n]} ||(X')^T X_i' - X^T X_i||_2
$$
$$
\leqslant 2 \max_{k \in [n]} (||[X_{lk} X_{jk}]_{l \in [p]}||_2)
$$
$$
\leqslant 2\sqrt{p} X_M^2
$$
$$
\rightarrow \Delta_2(X^T X_{S_{\mathcal{A}}}) \leqslant 2 X_M^2 \sqrt{ps} = \mathcal{O}(\sqrt{ps})
$$

# D Useful Results for optimality of SPriFed-OMP

**Lemma 19.** *Consider the term $M = [\mathbb{I} + NA^{-1}]^{-1}v$ where $N := \frac{\eta_{\beta_{S_\mathcal{A},S_\mathcal{A}}}}{n}, A := (\frac{X_{S_\mathcal{A}}^T X_{S_\mathcal{A}}}{n})^{-1}$, $v$ is any conformable vector and $n$ is the sample size. The remaining terms have been borrowed from Algorithm 3. Note that each element in $\eta_{\beta_{S_\mathcal{A},S_\mathcal{A}}}$ has a distribution of $\mathcal{N}(0, p/\mu_p^2)$. We show that*

$$||M||_2 \leqslant \kappa_M ||v||_2; \kappa_M = \frac{1}{1 + \frac{\varepsilon_\eta \sqrt{s}\kappa_s}{n(1-\zeta_{s+1})}}$$

*Here, $\varepsilon_\eta$ is chosen as per the lower bound of the smallest singular value of the random matrix $X_{S_\mathcal{A}}$ (by Theorem 1.2 from Rudelson & Vershynin (2008)), $n = \sqrt{ps}\kappa_n$ from assumptions in Theorem 7 and so we have that each row in $\frac{\eta_{\beta_{S_\mathcal{A},S_\mathcal{A}}}}{n}$ has the distribution $\sim \frac{1}{(n/\sqrt{s}\kappa_s)}\mathcal{N}(0,1)$.*

**Proof:**

$$
\begin{aligned}
||M||_2 &= ||[\mathbb{I} + NA^{-1}]^{-1}v||_2 \\
&\overset{(1)}{\leqslant} \frac{||v||_2}{\sigma_{\min}[\mathbb{I} + NA^{-1}]} \\
&\overset{(2)}{=} \frac{||v||_2}{1 + \sigma_{\min}[NA^{-1}]} \\
&\overset{(3)}{\leqslant} \frac{||v||_2}{1 + \sigma_{\min}[N]\sigma_{\min}[A^{-1}]} \\
&\overset{(4)}{\leqslant} \frac{||v||_2}{1 + \frac{s\varepsilon_\eta \kappa_s}{n\sqrt{s}}(1/\sigma_{\max}[A])} \\
&\overset{(5)}{\leqslant} \frac{||v||_2}{1 + \frac{\sqrt{s}\varepsilon_\eta \kappa_s}{n(1+\zeta_{s+1})}}
\end{aligned}
$$

Here *(1)* follows from Lemma 30 and since $\sigma_{\max}[K^{-1}] = \frac{1}{\sigma_{\min}[K]}$. *(2)* is true since $||k\mathbb{I} + K||_2 = k + ||K||_2$ where $\mathbb{I}$ is the identity matrix and $K$ is some conformable matrix. Furthermore, with high probability, the minimum eigenvalue of the random Gaussian matrix is lower bounded by a positive value. This statement is true by Theorem *1.2* from Rudelson & Vershynin (2008) since here each row of $\eta_{\beta_{S_\mathcal{A},S_\mathcal{A}}}$ has the distribution $\sim s\kappa_s\mathcal{N}(0,\mathbb{I}_s)$. Thus the random Gaussian matrix is positive definite *w.h.p.* Therefore, we have statement *(3)* since $\sigma_{\min}[AB] \geqslant \sigma_{\min}[A] \cdot \sigma_{\min}[B]$ where $A, B$ are conformable Hermitian matrices. *(4)* holds subsequently from Theorem 1.2 from Rudelson & Vershynin (2008) by considering the random matrix $n \cdot N$ with entries from the distribution $s\kappa_s\mathcal{N}(0,\mathbb{I}_{s^2})$. *(5)* is based on the bounds of the eigenvalues of a RIP matrix $K$ with RIC $\zeta_{s+1}$ and order $s + 1$.

**Lemma 20.** *Consider the term $M_2 = \mathbb{I} - (\mathbb{I} + NA^{-1})^{-1}NA^{-1}$. We show that $||M_2||_2$ is upper bounded by $\kappa_{M_2}$ where $N = \frac{\eta_{\beta_{S_\mathcal{A},S_\mathcal{A}}}}{n}$ and $A = \frac{X_{S_\mathcal{A}}^T X_{S_\mathcal{A}}}{n}$ (as terms represented in Algorithm 3). Here, $\kappa_{M_2} = 1 - \frac{\kappa_{M_1}\varepsilon_\eta \sqrt{s}\kappa_s}{(1+\zeta_{s+1})n}$ and $\kappa_{M_1} = \frac{||v||_2}{1+\frac{C_1 s^{3/2}\kappa_s}{n(1-\zeta_{s+1})}} = \kappa_{M_1}||v||_2$. Furthermore, $||M_1|| = ||(\mathbb{I} + NA^{-1})v||_2$ is upper bounded by $\kappa_{M_1}$. Here, $\varepsilon_\eta, C_1$ depend (polynomially) on the sub-Gaussian moment bounds of $\mathcal{N}(0,1)$ (Rudelson & Vershynin, 2008).*

**Proof:**

$$
\begin{aligned}
||M_2||_2 &= ||\mathbb{I} - (\mathbb{I} + NA^{-1})^{-1}NA^{-1}||_2 \\
&\overset{(1)}{=} 1 - ||(\mathbb{I} + NA^{-1})^{-1}NA^{-1}||_2 \\
&\overset{(2)}{\leqslant} 1 - \kappa_{M_1}||NA^{-1}||_2 \\
&\overset{(3)}{\leqslant} 1 - \frac{\kappa_{M_1}\varepsilon_\eta s\kappa_s}{n\sqrt{s}}||A^{-1}||_2
\end{aligned}
$$

$$\overset{(4)}{\leqslant} 1 - \frac{\kappa_{M_1}\varepsilon_\eta\sqrt{s}\kappa_s}{n(1+\zeta_{s+1})}$$

Here statement (1) follows by $||k\mathbb{I} - K||_2 = k - ||K||_2$ where $\mathbb{I}$ is the identity matrix and $K$ is a conformable matrix. (2) is based on identifying a lower bound for $M_1$ *(below)*. *(3)* employs theorem 1.2 from Rudelson & Vershynin (2008) where the rows in $n \cdot N$ have distribution of $s\kappa_s\mathcal{N}(0, \mathbb{I}_s)$. *(4)* uses the bound on the eigen-values of the *RIP* matrix $A$. To compute the lower bound for $||N||$ we consider some conformable vector $v$,

$$||M_1||_2 = ||(\mathbb{I} + NA^{-1})^{-1}v||_2$$

$$\overset{(1)}{\geqslant} \sigma_{\min}((\mathbb{I} + NA^{-1})^{-1})||v||_2$$

$$\overset{(2)}{\geqslant} \frac{||v||_2}{\sigma_{\max}(\mathbb{I} + NA^{-1})}$$

$$\overset{(3)}{=} \frac{||v||_2}{1 + \sigma_{\max}(NA^{-1})}$$

$$\overset{(4)}{\geqslant} \frac{||v||_2}{1 + \sigma_{\max}(N)\sigma_{\max}(A^{-1}))}$$

$$\overset{(5)}{\geqslant} \frac{||v||_2}{1 + \frac{C_1\sqrt{s}s\kappa_s}{(n)(1-\zeta_{s+1})}}$$

$$\overset{(6)}{=} \frac{||v||_2}{1 + \frac{C_1 s^{3/2}\kappa_s}{n(1-\zeta_{s+1})}} = \kappa_{M_1}||v||_2$$

*(1)-(4)* follow from simple matrix manipulations, *(5)* is based on Theorem 2.4 from (Rudelson & Vershynin, 2008) for matrix $n \cdot N$ with values distributed as $s\kappa_s\mathcal{N}(0,1)$ and *(6)* is based on bounds on eigenvalues of matrix $A^{-1}$.

**Lemma 21.** *We show that the terms 2-norm of the terms $L_1 = \frac{X_{S_\mathcal{A}}^T(X_{S_\mathcal{A}^c}\alpha_{S_*}^c + \epsilon)}{n}$ and $L_2 = \frac{X_{S_\mathcal{A}}^T y}{n}$ required in our analysis are upper-bounded as expressed below. Essentially, $L_1$ considers the value of y after multiplying it by the projection matrix that removes certain columns of $X_{S_\mathcal{A}}$ (due to orthogonality between these columns and the projection matrix).*

$$||L_1||_2 \leqslant \zeta_{s+1}||\alpha_{S_*}^c||_2 + \sqrt{1+\zeta_{s+1}}\kappa_\epsilon$$

$$||L_2||_2 \leqslant \sqrt{1+\zeta_{s+1}} \cdot \left(||\alpha_{S_*}||_2 + \sqrt{1+\zeta_{s+1}}\kappa_\epsilon\right)$$

**Proof:** Consider the analysis for $L_1$,

$$||L_1||_2 = ||\frac{X_{S_\mathcal{A}}^T(X_{S_\mathcal{A}^c}\alpha_{S_*}^c + \epsilon)}{n}||_2$$

$$\overset{(1)}{\leqslant} ||\frac{X_{S_\mathcal{A}}^T X_{S_\mathcal{A}^c}\alpha_{S_*}^c}{n}||_2 + ||\frac{X_{S_\mathcal{A}}^T e}{n}||_2$$

$$\overset{(2)}{\leqslant} (\zeta_{s+1})||\alpha_{S_*}^c||_2 + \sqrt{1+\zeta_{s+1}}\kappa_\epsilon$$

(1) follows by the triangle inequality and the RIP definition for matrices $X_{S_\mathcal{A}}^T/\sqrt{n}$, $X_{S_*}/\sqrt{n}$ respectively. By the $L_2$-$L_\infty$ inequality we further have $||x||_2 \leqslant \sqrt{d}||x||_\infty$ where $x \in \mathbb{R}^d$. Clearly, with Lemma 36 we then have $||e||_2 \leqslant \sqrt{n}\kappa_\epsilon$ *w.h.p.*. where $\kappa_\epsilon = \sigma_\epsilon\sqrt{2\log 2/p_0}$ where $e_i \sim \mathcal{N}(0, \sigma_\epsilon^2)$ and $p_0$ is determined by our probability requirements. Now, for $L_2$,

$$||L_2||_2 = ||\frac{X_{S_\mathcal{A}}^T y}{n}||_2$$

$$= ||\frac{X_{S_{\mathcal{A}}}^T(X_{S_*}\alpha_{S_*} + \epsilon)}{n}||_2$$

$$\overset{(1)}{\leqslant} ||\frac{X_{S_{\mathcal{A}}}^T X_{S_*}\alpha_{S_*}}{n} + \frac{X_{S_{\mathcal{A}}}^T e}{n}||_2$$

$$\overset{(2)}{\leqslant} (1+\zeta_{s+1})||\alpha_{S_*}||_2 + \sqrt{1+\zeta_{s+1}}\kappa_\epsilon$$

(1) follows since $y = X_{S_*}\alpha_{S_*} + \epsilon$. (2) follows by RIP definition and Lemma 18 for the first term and Lemmas 36,18 for the error term.

## D.1  Bounding the noise terms in SPriFed-OMP

**Lemma 22.** *[**Bound small inverse term**] We show that for a small value $0 < t < 1$, $\frac{1}{1-t} \leqslant 1 + \nu t$ if $\nu = \frac{1}{1-B_t}$ where $B_t$ is an upper bound of $t$. Furthermore, $\nu \in (1, 2)$.*

**Proof:** Suppose, we assume that $\frac{1}{1-t} \leqslant 1 + \nu t$ holds. Then, we can write the inequality simply,

$$0 \leqslant (\nu - 1)t - \nu t^2$$

$$\rightarrow t \leqslant \frac{\nu - 1}{\nu}$$

Setting, $B_t = \frac{\nu-1}{\nu}$, we obtain $\nu = \frac{1}{1-B_t}$. We note, that for the second statement to hold, we require that $\nu - 1 > 0$ and for $\nu \geqslant 2$ the statement holds trivially.

### D.1.1  Note on the difference between the basis columns and the non-basis columns

We present the analysis for noise terms when only a single feature $j$ is picked each time (Lemmas 23-27). However, we note that the same result holds when we pick the set $S_{\mathcal{A}}^c$ instead of $j$ alone. This is true since the total cardinality of the first matrix (either for the noise or signal matrix) is less than $s + 1$. Therefore, results from RIP and other singular value bounds apply to this case as well. This is particularly important for true basis columns. However, for the non-basis columns, this is not useful as the total cardinality of the non-basis set and the SPriFed-OMP selected feature set will easily exceed $s$ (closer to $p$).

**Lemma 23.** *The noise terms*

$$||\eta_1^j|| = ||X_j^T X_{S_{\mathcal{A}}}(X_{S_{\mathcal{A}}}^T X_{S_{\mathcal{A}}})^{-1}$$
$$\cdot [\mathbb{I} + \eta_{\beta_{S_{\mathcal{A}},S_{\mathcal{A}}}}(X_{S_{\mathcal{A}}}^T X_{S_{\mathcal{A}}})^{-1}]^{-1}; j \in S_*$$

$$||(\eta_1^j)'|| = ||X_j^T X_{S_{\mathcal{A}}}(X_{S_{\mathcal{A}}}^T X_{S_{\mathcal{A}}})^{-1}$$
$$\cdot [\mathbb{I} + \eta_{\beta_{S_{\mathcal{A}},S_{\mathcal{A}}}}(X_{S_{\mathcal{A}}}^T X_{S_{\mathcal{A}}})^{-1}]^{-1}; j \notin S_*$$

*(Theorem 7) are upper-bounded such that,*

$$||(\eta_1^j)'||_2 \leqslant s^2\zeta_{s+1}\kappa_s\sqrt{2\log(\frac{s^2}{p_b})}(1+\nu\zeta_{s+1})^2\kappa_M$$
$$\cdot (\sqrt{1+\zeta_{s+1}}||\alpha_{S_*}||_2 + \kappa_\epsilon)$$

$$||\eta_1^j||_2 \leqslant s^2\kappa_s\sqrt{2\log(\frac{s^2}{p_b})}(1+\nu\zeta_{s+1})^3\kappa_M$$
$$\cdot (\sqrt{1+\zeta_{s+1}}||\alpha_{S_*}||_2 + \kappa_\epsilon)$$

*Here, terms are borrowed from Theorem 7, Lemma 19) and $p_b$ is base failure probability defined in Lemma 36.*

**Proof:** We provide the proof for the $||(\eta_1^j)'||_2$ term. However, proving the inequality for the $||\eta_1^j||_2$ is straightforward.

$$||(\eta_1^j)'||_2 = ||X_j^T X_{S_{\mathcal{A}}}(X_{S_{\mathcal{A}}}^T X_{S_{\mathcal{A}}})^{-1}[\mathbb{I} + \eta_{\beta_{S_{\mathcal{A}},S_{\mathcal{A}}}}(X_{S_{\mathcal{A}}}^T X_{S_{\mathcal{A}}})^{-1}]^{-1}$$

$$\cdot \eta_{\beta_{S_\mathcal{A}},S_\mathcal{A}}(X_{S_\mathcal{A}}^T X_{S_\mathcal{A}})^{-1}X_{S_\mathcal{A}}^T y||_2$$

$$= n||\frac{X_{S_\mathcal{A}^c}^T X_{S_\mathcal{A}}}{n}(\frac{X_{S_\mathcal{A}}^T X_{S_\mathcal{A}}}{n})^{-1}$$

$$\cdot [\mathbb{I} + \frac{\eta_{\beta_{S_\mathcal{A}},S_\mathcal{A}}}{n}(\frac{X_{S_\mathcal{A}}^T X_{S_\mathcal{A}}}{n})^{-1}]^{-1}$$

$$\cdot \frac{\eta_{\beta_{S_\mathcal{A}},S_\mathcal{A}}}{n}(\frac{X_{S_\mathcal{A}}^T X_{S_\mathcal{A}}}{n})^{-1}\frac{X_{S_\mathcal{A}}^T y}{n}||_2$$

$$\overset{(1)}{\leqslant} n\frac{\zeta_{s+1}}{(1-\zeta_{s+1})^2}||M||_2||\frac{\eta_{\beta_{S_\mathcal{A}},S_\mathcal{A}}}{n}||_2 L_1$$

$$\overset{(2)}{\leqslant} n\zeta_{s+1}||\frac{\eta_{\beta_{S_\mathcal{A}},S_\mathcal{A}}}{n}||_2(1+\nu\zeta_{s+1})^2\kappa_M$$

$$\cdot(\sqrt{1+\zeta_{s+1}}||\alpha_{S_*}||_2 + \kappa_\epsilon)$$

$$\overset{(3)}{\leqslant} s^2\zeta_{s+1}\kappa_s\sqrt{2\log(\frac{s^2}{p_b})}(1+\nu\zeta_{s+1})^2\kappa_M$$

$$\cdot(\sqrt{1+\zeta_{s+1}}||\alpha_{S_*}||_2 + \kappa_\epsilon)$$

Here, *(1)* follows from Lemma 17, for matrices $\frac{X_j^T}{\sqrt{n}}$ and $\frac{X_{S_\mathcal{A}}}{\sqrt{n}}$ and bounded eigenvalues for $||(\frac{X_{S_\mathcal{A}}^T X_{S_\mathcal{A}}}{n})^{-1}||_2$. The middle and last terms are handled by Lemmas 20 and 21 respectively. In *(2)*, we replace the bounds for the terms $M, L_1$ using Lemmas 20 and 21 respectively. We replace the denominator by using the bound in Lemma 22. Finally, *(3)* follows from simply noticing that each element in $\eta_{\beta_{S_\mathcal{A}},S_\mathcal{A}}$ has distribution $\mathcal{N}(0, s^2\kappa_s^2)$, $L_2 - L_\infty$ inequality and lemma 36. The bound for the noise term holds with probability close to 1 of the form $p_b = 1 - k_1 e^{-k_2} \approx 1$ where $k_1, k_2$ are positive constants.

**Lemma 24.** *The noise terms*

$$||\eta_{31}^j||_2 = n||\frac{X_j^T X_{S_\mathcal{A}}}{n}(\frac{X_{S_\mathcal{A}}^T X_{S_\mathcal{A}}}{n})^{-1}\left(\mathbb{I} - [\mathbb{I} + \frac{\eta_{\beta_{S_\mathcal{A}},S_\mathcal{A}}}{n}(\frac{X_{S_\mathcal{A}}^T X_{S_\mathcal{A}}}{n})^{-1}]^{-1}\right.$$

$$\left.\cdot\frac{\eta_{\beta_{S_\mathcal{A}},S_\mathcal{A}}}{n}(\frac{X_{S_\mathcal{A}}^T X_{S_\mathcal{A}}}{n})^{-1}\right)\frac{\eta_{\gamma_{S_\mathcal{A}}}}{n}||_2; j \in S_*$$

$$||(\eta_{31}^j)'||_2 = n||\frac{X_j^T X_{S_\mathcal{A}}}{n}(\frac{X_{S_\mathcal{A}}^T X_{S_\mathcal{A}}}{n})^{-1}\left(\mathbb{I} - [\mathbb{I} + \frac{\eta_{\beta_{S_\mathcal{A}},S_\mathcal{A}}}{n}\right.$$

$$\left.\cdot(\frac{X_{S_\mathcal{A}}^T X_{S_\mathcal{A}}}{n})^{-1}]^{-1}\right.$$

$$\left.\cdot\frac{\eta_{\beta_{S_\mathcal{A}},S_\mathcal{A}}}{n}(\frac{X_{S_\mathcal{A}}^T X_{S_\mathcal{A}}}{n})^{-1}\right)\frac{\eta_{\gamma_{S_\mathcal{A}}}}{n}||_2; j \notin S_*$$

*is shown to be upper bounded such that,*

$$||(\eta_{31}^j)'||_2 \leqslant s\kappa_s\zeta_{s+1}\sqrt{2\log(\frac{2s}{p_b})}(1+\nu\zeta_{s+1})\kappa_{M_2}$$

$$||\eta_{31}^j||_2 \leqslant s\kappa_s\sqrt{2\log(\frac{2s}{p_b})}(1+\nu\zeta_{s+1})^2\kappa_{M_2}$$

*The terms follow the notation in Algorithm 3, Theorem 7, Lemmas 22, 20. $p_b$ is the base failure probability defined in Lemma 36.*

**Proof:** We first demonstrate the proof for the case when $j \notin S_*$. The proof for the $j \in S_*$ case follows simply by replacing the first inequality involving the term $X_j^T X_{S_\mathcal{A}}$.

$$||(\eta_{31}^j)'||_2 = n||\frac{X_j^T X_{S_\mathcal{A}}}{n}(\frac{X_{S_\mathcal{A}}^T X_{S_\mathcal{A}}}{n})^{-1}\left(\mathbb{I} - [\mathbb{I} + \frac{\eta_{\beta_{S_\mathcal{A}},S_\mathcal{A}}}{n}\right.$$

$$\cdot \left(\frac{X_{S_{\mathcal{A}}}^T X_{S_{\mathcal{A}}}}{n}\right)^{-1}]^{-1}$$

$$\cdot \frac{\eta_{\beta_{S_{\mathcal{A}},S_{\mathcal{A}}}}}{n}\left(\frac{X_{S_{\mathcal{A}}}^T X_{S_{\mathcal{A}}}}{n}\right)^{-1}\bigg)\frac{\eta_{\gamma_{S_{\mathcal{A}}}}}{n}||_2$$

$$\overset{(1)}{\leqslant} \frac{n\zeta_{s+1}}{(1-\zeta_{s+1})}||M_2||_2||\frac{\eta_{\gamma_{S_{\mathcal{A}}}}}{n}||_2$$

$$\overset{(2)}{\leqslant} \frac{n(\zeta_{s+1})\kappa_{M_2}}{(1-\zeta_{s+1})}||\frac{\eta_{\gamma_{S_{\mathcal{A}}}}}{n}||_2$$

$$\overset{(3)}{\leqslant} \frac{s\kappa_s\sqrt{2\log(\frac{2s}{p_b})}\zeta_{s+1}\kappa_{M_2}}{(1-\zeta_{s+1})}$$

$$\overset{(4)}{\leqslant} s\kappa_s\zeta_{s+1}\sqrt{2\log(\frac{2s}{p_b})}(1+\nu\zeta_{s+1})\kappa_{M_2}$$

Here, *(1)* follows from RIP properties in Lemmas 16-17 (since $j \notin S_{\mathcal{A}}$) and $M_2$ is defined in Lemma 20. *(2)* follows from the bound in Lemma 20. *(3)* follows from Lemma 36, $L_2$-$L_\infty$ inequality and the Gaussian Mechanism for differential privacy. Here, $(\eta_{\gamma_{S_{\mathcal{A}}}})^j \sim \mathcal{N}(0, \kappa_s^2 s)$. And $\kappa_s$ is defined in Algorithm 3. *(4)* is a direct application of Lemma 22.

**Lemma 25.** *The noise terms* $||\eta_{32}^j||_2, ||(\eta_{32}^j)'||_2$ *which both have the form* $n||\frac{\eta_{\beta_{j},S_{\mathcal{A}}}}{n}\left(\frac{X_{S_{\mathcal{A}}}^T X_{S_{\mathcal{A}}}}{n}\right)^{-1}\bigg(\mathbb{I} - [\mathbb{I} + $

$\frac{\eta_{\beta_{S_{\mathcal{A}},S_{\mathcal{A}}}}}{n}\left(\frac{X_{S_{\mathcal{A}}}^T X_{S_{\mathcal{A}}}}{n}\right)^{-1}]^{-1}\frac{\eta_{\beta_{S_{\mathcal{A}},S_{\mathcal{A}}}}}{n}\left(\frac{X_{S_{\mathcal{A}}}^T X_{S_{\mathcal{A}}}}{n}\right)^{-1}\bigg)\frac{X_{S_{\mathcal{A}}}^T y}{n}||_2$ *for sets* $j \in S_*$ *and* $j \notin S_*$ *respectively are both upper bounded by* $\sqrt{ps}\kappa_p\sqrt{2\log(\frac{2p}{p_b})}\kappa_{M_2}\left(\sqrt{1+\zeta_{s+1}}||\alpha_{S_*}||_2 + \kappa_\epsilon\right)(1+\nu\zeta_{s+1}))$. *The notation is borrowed from Algorithm 3 and Theorem 7, Lemmas 22, 20.* $p_b$ *is the base failure probability defined in Lemma 36.*

$$||\eta_{32}^j||_2 = n||\frac{\eta_{\beta_{j},S_{\mathcal{A}}}}{n}\left(\frac{X_{S_{\mathcal{A}}}^T X_{S_{\mathcal{A}}}}{n}\right)^{-1}\bigg(\mathbb{I} - [\mathbb{I} + \frac{\eta_{\beta_{S_{\mathcal{A}},S_{\mathcal{A}}}}}{n}$$

$$\cdot \left(\frac{X_{S_{\mathcal{A}}}^T X_{S_{\mathcal{A}}}}{n}\right)^{-1}]^{-1}$$

$$\cdot \frac{\eta_{\beta_{S_{\mathcal{A}},S_{\mathcal{A}}}}}{n}\left(\frac{X_{S_{\mathcal{A}}}^T X_{S_{\mathcal{A}}}}{n}\right)^{-1}\bigg)\frac{X_{S_{\mathcal{A}}}^T y}{n}||_2$$

$$\overset{(1)}{\leqslant} n||\frac{\eta_{\beta_{j},S_{\mathcal{A}}}}{n}||_2\frac{||M_2||_2||L_2||_2}{1-\zeta_{s+1}}$$

$$\overset{(2)}{\leqslant} n||\frac{\eta_{\beta_{j},S_{\mathcal{A}}}}{n}||_2\frac{\kappa_{M_2}\left(\sqrt{1+\zeta_{s+1}}||\alpha_{S_*}||_2 + \kappa_\epsilon\right)}{1-\zeta_{s+1}}$$

$$\overset{(3)}{\leqslant} n||\frac{\eta_{\beta_{j},S_{\mathcal{A}}}}{n}||_2\kappa_{M_2}\left(\sqrt{1+\zeta_{s+1}}||\alpha_{S_*}||_2 + \kappa_\epsilon\right)$$

$$\cdot (1+\nu\zeta_{s+1}))$$

$$\overset{(4)}{\leqslant} n\sqrt{s}||\frac{\eta_{\beta_{j},S_{\mathcal{A}}}}{n}||_\infty\kappa_{M_2}\left(\sqrt{1+\zeta_{s+1}}||\alpha_{S_*}||_2 + \kappa_\epsilon\right)$$

$$\cdot (1+\nu\zeta_{s+1}))$$

$$\overset{(5)}{\leqslant} \sqrt{ps}\kappa_p\sqrt{2\log(\frac{2p}{p_b})}\kappa_{M_2}\left(\sqrt{1+\zeta_{s+1}}||\alpha_{S_*}||_2 + \kappa_\epsilon\right)$$

$$\cdot (1+\nu\zeta_{s+1}))$$

Statement *(1)* follows by substituting for the middle and last term (from Lemmas 20, 21 resp.), separating the noise vector and bounding the singular values of $\frac{X_{S_{\mathcal{A}}}^T X_{S_{\mathcal{A}}}}{n})^{-1}$. *(2)* follows by bounds in Lemmas 20, 21.

*(3)* is a direct application of inverting the denominator term with Lemma 22. *(4)* uses the $L_2 - L_\infty$ inequality. The final statement uses Lemma 36, distribution of the noise matrix and bounds it for the appropriate set and failure probability as shown in Lemma 36.

**Lemma 26.** *The noise terms $||\eta_{33}^j||_2, ||(\eta_{33}^j)'||_2$ over sets $j \in S_*$ and $j \notin S_*$ respectively both have the form*

$$n||\frac{\eta_{\beta_{j,S_\mathcal{A}}}}{n}(\frac{X_{S_\mathcal{A}}^T X_{S_\mathcal{A}}}{n})^{-1}\Big(\mathbb{I} - [\mathbb{I} + \frac{\eta_{\beta_{S_\mathcal{A},S_\mathcal{A}}}}{n}(\frac{X_{S_\mathcal{A}}^T X_{S_\mathcal{A}}}{n})^{-1}]^{-1}$$

$$\cdot \frac{\eta_{\beta_{S_\mathcal{A},S_\mathcal{A}}}}{n}(\frac{X_{S_\mathcal{A}}^T X_{S_\mathcal{A}}}{n})^{-1}\Big)\frac{\eta_{\gamma_{S_\mathcal{A}}}}{n}||_2$$

*are both upper bounded by $2\kappa_{M_2}(1+\nu\zeta_{s+1})\frac{s\kappa_s\kappa_p}{\kappa_n}\sqrt{\log(\frac{2p}{p_b})\log(\frac{2s}{p_b})}$. The notation follows from Algorithm 3 and Theorem 7, Lemmas 22, 20. $p_b$ is the base failure probability defined in Lemma 36.*

$$||\eta_{33}^j||_2 = n||\frac{\eta_{\beta_{j,S_\mathcal{A}}}}{n}(\frac{X_{S_\mathcal{A}}^T X_{S_\mathcal{A}}}{n})^{-1}\Big(\mathbb{I} - [\mathbb{I} + \frac{\eta_{\beta_{S_\mathcal{A},S_\mathcal{A}}}}{n}(\frac{X_{S_\mathcal{A}}^T X_{S_\mathcal{A}}}{n})^{-1}]^{-1}\Big)$$

$$\cdot \frac{\eta_{\beta_{S_\mathcal{A},S_\mathcal{A}}}}{n}(\frac{X_{S_\mathcal{A}}^T X_{S_\mathcal{A}}}{n})^{-1}\Big)\frac{\eta_{\gamma_{S_\mathcal{A}}}}{n}||_2$$

$$\overset{(1)}{\leqslant} n\frac{||M_2||_2}{1-\zeta_{s+1}}||\frac{\eta_{\beta_{j,S_\mathcal{A}}}}{n}||_2||\frac{\eta_{\gamma_{S_\mathcal{A}}}}{n}||_2$$

$$\overset{(2)}{\leqslant} n\kappa_{M_2}(1+\nu\zeta_{s+1})\sqrt{s}||\frac{\eta_{\beta_{j,S_\mathcal{A}}}}{n}||_\infty\sqrt{s}||\frac{\eta_{\gamma_{S_\mathcal{A}}}}{n}||_\infty$$

$$\overset{(3)}{\leqslant} \kappa_{M_2}(1+\nu\zeta_{s+1})\frac{\sqrt{ps}s}{n}\kappa_s\kappa_p 2\sqrt{\log(\frac{2p}{p_b})\log(\frac{2s}{p_b})}$$

$$\overset{(4)}{=} 2\kappa_{M_2}(1+\nu\zeta_{s+1})\frac{s\kappa_s\kappa_p}{\kappa_n}\sqrt{\log(\frac{2p}{p_b})\log(\frac{2s}{p_b})}$$

Statement *(1)* follows from bounding the eigenvalues of $X_{S_\mathcal{A}}^T X_{S_\mathcal{A}} n)^{-1}$, substituting the value of the middle term as $M_2$ (from Lemma 20) and separating the noise 2-norms. *(2)* leverages the $L_2 - L_\infty$ inequality for the noise 2-norms, bounds $M_2$ with Lemma 20 and inverts the denominator with the simple Lemma 22. *(3)* follows from noting the variances of the noise matrices as being proportional to $p$ and $s$ and using the maximum concentration bound (Lemma 36) to pick an appropriate value. $\kappa_s$ and $\kappa_p$ are picked by the privacy mechanism in Algorithm 3. Depending on the set we are maximizing over for $j$ we can pick $d = s$ (basis columns) or $d = p - s$ (non-basis columns). $p_b$ is some appropriate base probability that is generally quite small. *(4)* follows by substituting $n = \sqrt{ps}\kappa_n$.

**Lemma 27.** *The noise term $||\frac{\eta_2^j}{n}||_2 = ||\frac{(\eta_{\gamma_0^-})_j}{n}||_2$ is upper bounded by $\frac{\sqrt{2\log(\frac{2p}{p_b})}\kappa_p}{\sqrt{s}\kappa_n}$. Given, $j \in T, d = |T|$ and $p_b$ is some base probability that is quite small.*

**Proof:**

$$||\frac{\eta_2^j}{n}||_2 = ||\frac{(\eta_{\gamma_0^-})_j}{n}||_2 \leqslant \frac{\sqrt{p}}{n}\kappa_p\sqrt{2\log(\frac{2p}{p_b})} = \frac{\sqrt{2\log(\frac{2p}{p_b})}\kappa_p}{\sqrt{s}\kappa_n}$$

where the last inequality holds because of Lemma 36 and since $(\eta_{\gamma_0^-})_j \sim \mathcal{N}(0, \kappa_p^2 p)$ where $\kappa_p$ is picked by the privacy mechanism in Algorithm 3.

# E    Proof of Main Theorem 7

We will use induction to prove our main result as noted in section 6.

**Step 1:** We show that we pick the correct basis in step 1 *with high probability (w.h.p.)*. Here, the correct basis implies that we choose the column index belonging to $\mathcal{S}_*$. First step from Algorithm 3 is written as

$j \leftarrow \mathrm{argmax}_{j \in \Omega} |(\gamma_0^-)_j|$. Our strategy is to find the maximum correlation's lower and upper bound under the assumptions $j \in S_*$ and $j \notin S_*$, respectively. Denote these correlations $C$ and $C'$ respectively and let their lower bound and upper bound be $C_*$ and $C'_*$ respectively. If $C'_* < C_*$ then the correct basis (from $S_*$) is chosen at each step. This statement is true since $C'_* < C_*$ implies that the upper bound of the maximum correlation of non-support columns is smaller than the lower bound of the maximum correlation support columns. Thus, we first attempt to find the values $C_*, C'_*$ so as to equate the inequality above.

Thus, the lower bound of the maximum correlation of support columns is given by,

$$
\begin{aligned}
C &= \max_{j \in S_*} |(\gamma_0^-)_j| \\
&= \max_{j \in S_*} |X_j^T y + \eta_{(\gamma_0^-)_j}| \\
&\geqslant \frac{||X_{S_*}^T y + \eta_{(\gamma_0^-)_j}||_2}{\sqrt{s}} \quad \text{(by } L_2, L_\infty \text{ ineq.)} \\
&\geqslant \frac{||X_{S_*}^T y||_2}{\sqrt{s}} - \max_{j \in S_*} |\eta_{(\gamma_0^-)_j}| \quad \text{(by reverse triangle inequality)}
\end{aligned}
$$

Suppose events $E_1, E_2, ..., E_s$ each occur w.p. $(1 - p_a)$. Event $E_k, k \in [s]$ indicates that $|\eta_{(\gamma_0^-)_k}| \leqslant B_a$ for some $B_a > 0$. This could be rewritten as $\Pr[E_k] = \Pr[|\eta_{(\gamma_0^-)_k}| \leqslant B_a] = 1 - p_a$. Invoking Lemma 32 and since $|\eta_{(\gamma_0^-)_k}| \sim \mathcal{N}(0, \sigma_0^2)$ we can write $p_a \triangleq 2e^{-\frac{B_a}{2\sigma_0^2}} \rightarrow B_a = \sigma_0 \sqrt{2\log(\frac{2}{p_a})} = \sqrt{2\log(\frac{2}{p_a})} \frac{\kappa_p}{\epsilon_{step}} \sqrt{p} = \kappa_a \cdot \sqrt{p}$ where $\kappa_a = \sqrt{2\log(\frac{2}{p_a})} \frac{\kappa_p}{\epsilon_{step}}$. To bound the maximum of the noise terms, we require that all events $E_k, k \in [s]$ hold simultaneously. By the union bound (Lemma 33), $\Pr[\max_{k \in S_*} |\eta_{(\gamma_0^-)_k}| > B_a] = \Pr[\cup_{k \in S_*} E_k^c] \leqslant \sum_{k \in S_*} \Pr[E_k^c] = s \cdot p_a$. Therefore, $\Pr[\max_{k \in S_*} |\eta_{(\gamma_0^-)_k}| \leqslant B_a] \geqslant 1 - s \cdot p_a$. And so w.p. $1 - s \cdot p_a$ we can say that,

$$
\begin{aligned}
C &\geqslant \frac{||X_{S_*}^T y||_2}{\sqrt{s}} - \kappa_a \sqrt{p} \\
&= \frac{||X_{S_*}^T X_{S_*} \alpha_{S_*} + X_{S_*}^T e||_2}{\sqrt{s}} - \kappa_a \sqrt{p} \\
&\stackrel{(1)}{\geqslant} \frac{||X_{S_*}^T X_{S_*} \alpha_{S_*}||}{\sqrt{s}} - \frac{||X_{S_*}^T e||_2}{\sqrt{s}} - \kappa_a \sqrt{p} \\
&= \frac{n||\frac{X_{S_*}^T X_{S_*}}{n} \alpha_{S_*}||}{\sqrt{s}} - \frac{||X_{S_*}^T e||_2}{\sqrt{s}} - \kappa_a \sqrt{p}
\end{aligned}
$$

*(1)* follows by the reverse-triangle inequality. We know that by construction $X/\sqrt{n}$ has isotropic, sub-gaussian, independent rows and thus satisfies RIP of order $s + 1$ (14). Thus, by Lemma 16,

$$
C \geqslant \frac{n(1 - \zeta_{s+1})||\alpha_{S_*}||_2}{\sqrt{s}} - \sqrt{n} \frac{||\frac{X_{S_*}^T}{\sqrt{n}} e||_2}{\sqrt{s}} - \kappa_a \sqrt{p}
$$

From Lemma 18 and since $\frac{X}{\sqrt{n}}$ satisfies and RIP of order $s + 1$ we observe that, $||\frac{X_{S_*}^T}{\sqrt{n}} e||_2 \leqslant \sqrt{1 + \zeta_{s+1}} ||e||_2 \leqslant \sqrt{(1 + \zeta_{s+1})n} ||e||_\infty$. Combining these two results, we have,

$$
C \geqslant \frac{n(1 - \zeta_{s+1})||\alpha_{S_*}||_2}{\sqrt{s}} - \frac{n\sqrt{(1 + \zeta_{s+1})}||e||_\infty}{\sqrt{s}} - \kappa_a \sqrt{p}
$$

Consider the events $B_1, B_2, ..., B_n$ s.t. $\Pr[B_i] = \Pr[|e_i| \leqslant B_b] \geqslant (1 - p_e), i \in [n]$ (for some probability $p_e \in [0, 1]$. Comparing the expression with Lemma 32 and noting that $e \sim \mathcal{N}(0, \sigma_\epsilon^2)$ we observe that

$B_e = \sigma_\epsilon \sqrt{2\log(\frac{2}{p_e})} = \kappa_\epsilon$. We bound the probability of event $B$ with the union bound *s.t.*, $\Pr[B] = \Pr[\max_{i \in [n]} |e_i| \leq B_e]$. Therefore, $\Pr[B^c] \leq \sum_{i \in [n]} \Pr[B_i^c] \leq n \cdot p_e$. Thus, $\Pr[B] \geq 1 - np_e$. Thus, w.p. $1 - np_e$ event $B$ occurs and thus,

$$C \geq \frac{n(1 - \zeta_{s+1})||\alpha_{S_*}||_2}{\sqrt{s}} - n\kappa_\epsilon \frac{\sqrt{(1 + \zeta_{s+1})}}{\sqrt{s}} - \kappa_a \sqrt{p} = C_*$$

Next, we identify the expression for $C'_*$. For any $j \notin S_*$ let us upper bound $|(\gamma_0^-)_j|$,

$$
\begin{aligned}
C' &= \max_{j \notin S_*} |(\gamma_0^-)_j| \\
&= \max_{j \notin S_*} |X_j^T y + \eta_{(\gamma_0^-)_j}| \\
&= \max_{j \notin S_*} |X_j^T X_{S_*} \alpha_{S_*} + X_j^T e + \eta_{(\gamma_0^-)_j}| \\
&\overset{(1)}{\leq} ||X_j^T X_{S_*} \alpha_{S_*}||_2 + ||X_j^T e||_2 + \max_{j \notin S_*} |\eta_{(\gamma_0^-)_j}| \\
&\leq n||\frac{X_j^T X_{S_*}}{n} \alpha_{S_*}||_2 + \sqrt{n}||\frac{X_j^T}{\sqrt{n}} e||_2 + \max_{j \notin S_*} |\eta_{(\gamma_0^-)_j}|
\end{aligned}
$$

*(1)* follows by the $L_2$-$L_\infty$ inequality. Considering event $B$ again, we have that $\Pr[B] = \Pr[\max_{i \in [n]} |e_i| \leq B_e]$ and $B$ occurs w.p. $1 - np_e$.

Further, consider events $C_k, k \notin S_* s.t., \Pr[C_k] = \Pr[|\eta_{(\gamma_0^-)_k}| \leq B_c] \geq (1 - p_c)$. To bound the maximum of the noise terms, we need all $C_k, k \in [p - s]$ events to hold. By Lemma 33, $\Pr[\max_{k \notin S_*} |\eta_{(\gamma_0^-)_k}| > B_c] = \Pr[\cup_{k \in S_*} C_k^c] \leq \sum_{k \in S_*} \Pr[C_k^c] = (p - s) \cdot p_c$. Therefore, $\Pr[\max_{k \notin S_*} |\eta_{(\gamma_0^-)_k}| \leq B_c] \geq 1 - (p - s) \cdot p_c$. By Lemma 32, $B_c = \sigma_0 \sqrt{2\log(\frac{2}{p_c})} = \sqrt{2\log(\frac{2}{p_c})} \frac{\kappa_p}{\epsilon_{step}} \sqrt{p} = \kappa_c \cdot \sqrt{p}$.

Combined with the RIP Lemmas 16, 18 for any $j \notin S_*$, we get the upper bound,

$$C' \leq n\zeta_{s+1}||\alpha_{S_*}||_2 + n\sqrt{1 + \zeta_{s+1}}\kappa_\epsilon + \kappa_c \sqrt{p} = C'_*$$

Note that by the union bound (Lemma 33), events A, B, C hold w.p. at least $1 - sp_a - 2np_e - (p - s)p_c$. Let $p_a \triangleq \frac{p_0}{4s}, p_e \triangleq \frac{p_0}{4n}, p_c \triangleq \frac{p_0}{4(p-s)}$.

With probability $1 - p_0$ the bounds, $C_*, C'_*$ hold. Let us identify the conditions for $C'_* < C_*$.

$$
\begin{aligned}
\frac{n||\alpha_{S_*}||_2}{\sqrt{s}} &< n\zeta_{s+1}||\alpha_{S_*}||_2 (1 + \frac{1}{\sqrt{s}}) \\
&+ n\sqrt{1 + \zeta_{s+1}}\kappa_\epsilon (1 + \frac{1}{\sqrt{s}}) + (\kappa_a + \kappa_c)\sqrt{p}
\end{aligned}
$$

Thus, put together we obtain the following condition on the RIC, $\zeta_{s+1}$,

$$
\begin{aligned}
\zeta_{s+1} &< \frac{n||\alpha_{S_*}||_2 - n\sqrt{1 + \zeta_{s+1}}\kappa_\epsilon (1 + \sqrt{s}) - (\kappa_a + \kappa_c)\sqrt{ps}}{n||\alpha_{S_*}||_2 (\sqrt{s} + 1)} \\
&< \frac{n||\alpha_{S_*}||_2 - \sqrt{2}(1 + \sqrt{s})n\kappa_\epsilon - (\kappa_a + \kappa_c)\sqrt{ps}}{n||\alpha_{S_*}||_2 (\sqrt{s} + 1)}
\end{aligned}
$$

where the last line follows from the fact that $\zeta_{s+1} < 1$. Suppose, we can bound $\sqrt{2}(1 + \sqrt{s})n\kappa_\epsilon + (\kappa_a + \kappa_c)\sqrt{p} \leq \frac{n||\alpha_{S_*}||_2}{2}$ that would give us $\zeta_{s+1} < \frac{n||\alpha_{S_*}||_2}{2n||\alpha_{S_*}||_2(\sqrt{s}+1)} = \frac{1}{2(\sqrt{s}+1)}$. It remains to characterize the condition for $\sqrt{2}(1 + \sqrt{s})n\kappa_\epsilon + (\kappa_a + \kappa_c)\sqrt{p} \leq \frac{n||\alpha_{S_*}||_2}{2}$ to hold. Thus, we only need,

$$n \geq \frac{2\sqrt{2}(1 + \sqrt{s})n\kappa_\epsilon}{||\alpha_{S_*}||_2} + \frac{2(\kappa_a + \kappa_c)\sqrt{p}}{||\alpha_{S_*}||_2}$$

$$\to n(1 - \frac{2\sqrt{2}(1+\sqrt{s})\kappa_\epsilon}{||\alpha_{S_*}||_2}) \geqslant \frac{2(\kappa_a + \kappa_c)\sqrt{p}}{||\alpha_{S_*}||_2}$$

$$\to n \geqslant \frac{2(\kappa_a + \kappa_c)}{||\alpha_{S_*}||_2 - 2\sqrt{2}(1+\sqrt{s})\kappa_\epsilon}\sqrt{p}$$

Note, $\kappa_c = \sqrt{2\log(\frac{8(p-s)}{p_0})}\frac{\kappa_p}{\epsilon_{step}}$ and so $n = \Omega(\frac{\sqrt{p\log(p)}}{||\alpha_{S_*}||_2})$. We also require that $||\alpha_{S_*}||_2 - 2\sqrt{2}(1+\sqrt{s})\kappa_\epsilon > 0$ which is generally applicable since $\kappa_\epsilon$ (dependent on $\sigma_\epsilon$) is generally smaller or comparable to 1.

**Step (l+1)**: By induction, $C'_* < C_*$ holds for all steps from 1 to $l$. We now prove the same for the $(l+1)^{th}$ step.

**Case A: Analysis for when chosen column $j \in S_\mathcal{A}^c$** Let us determine a lower bound for the correlation for when the column chosen belongs to the true basis,

$$\begin{aligned}
C &= \max_{j \in S_*} |(\gamma_l^-)_j| \\
&= \max_{j \in S_*} |(\gamma_0^- - \beta(\beta_{S_\mathcal{A}})^{-1}\gamma_{S_\mathcal{A}}| \\
&= \max_{j \in S_*} |(X_j^T y + \eta_{(\gamma_0^-)_j}) \\
&\quad - (X_j^T X_{S_\mathcal{A}} + \eta_{\beta_{j,S_\mathcal{A}}})(X_{S_\mathcal{A}}^T X_{S_\mathcal{A}} + \eta_{\beta_{S_\mathcal{A},S_\mathcal{A}}})^{-1} \\
&\quad \cdot (X_{S_\mathcal{A}}^T y + \eta_{\gamma_{S_\mathcal{A}}})|
\end{aligned}$$

We now separate the signal and noise terms in order to analyze them further. Consider the following expression to relate the true (*i.e.,* non-private) signal- $S_{true}^j$, the privatized version of the signal $S_{priv}^j$ for the $j^{th}$ and the various noise values (denoted by the $\eta$ terms) for the $j^{th}$ dimension where $j \in \{1,...,p\}$.

$$\begin{aligned}
S_{priv}^j &\triangleq (X_j^T y + \eta_{(\gamma_0^-)_j}) \\
&\quad - (X_j^T X_{S_\mathcal{A}} + \eta_{\beta_{j,S_\mathcal{A}}})(X_{S_\mathcal{A}}^T X_{S_\mathcal{A}} + \eta_{\beta_{S_\mathcal{A},S_\mathcal{A}}})^{-1} \\
&\quad \cdot (X_{S_\mathcal{A}}^T y + \eta_{\gamma_{S_\mathcal{A}}}) \\
&= \underbrace{X_j^T y - X_j^T X_{S_\mathcal{A}}(X_{S_\mathcal{A}}^T X_{S_\mathcal{A}} + \eta_{\beta_{S_\mathcal{A},S_\mathcal{A}}})^{-1}X_{S_\mathcal{A}}^T y}_{S_{noisy}^j} \\
&\quad - \underbrace{(X_j^T X_{S_\mathcal{A}})(X_{S_\mathcal{A}}^T X_{S_\mathcal{A}} + \eta_{\beta_{S_\mathcal{A},S_\mathcal{A}}})^{-1}(\eta_{\gamma_{S_\mathcal{A}}})}_{\eta_{31}^j} \\
&\quad - \underbrace{(\eta_{\beta_{j,S_\mathcal{A}}})(X_{S_\mathcal{A}}^T X_{S_\mathcal{A}} + \eta_{\beta_{S_\mathcal{A},S_\mathcal{A}}})^{-1}(X_{S_\mathcal{A}}^T y)}_{\eta_{32}^j} \\
&\quad - \underbrace{(\eta_{\beta_{j,S_\mathcal{A}}})(X_{S_\mathcal{A}}^T X_{S_\mathcal{A}} + \eta_{\beta_{S_\mathcal{A},S_\mathcal{A}}})^{-1}(\eta_{\gamma_{S_\mathcal{A}}})}_{} \\
&\quad + \underbrace{\eta_{(\gamma_0^-)_j}}_{\eta_2^j}
\end{aligned}$$

We apply the Woodbury Identity-Kailath Variant (Eqn.31) to each expression marked by an asterisk to obtain the following simplifications,

$$\begin{aligned}
S_{noisy}^j &\overset{*}{=} \underbrace{X_j^T y - X_j^T X_{S_\mathcal{A}}(X_{S_\mathcal{A}}^T X_{S_\mathcal{A}})^{-1}X_{S_\mathcal{A}}^T y}_{S_{true}^j} \\
&\quad - \underbrace{X_j^T X_{S_\mathcal{A}}(X_{S_\mathcal{A}}^T X_{S_\mathcal{A}})^{-1}M_1\eta_{\beta_{S_\mathcal{A},S_\mathcal{A}}}(X_{S_\mathcal{A}}^T X_{S_\mathcal{A}})^{-1}X_{S_\mathcal{A}}^T y}_{\eta_1^j}
\end{aligned}$$

$$\eta_{31}^j \overset{*}{=} X_j^T X_{S_{\mathcal{A}}} (X_{S_{\mathcal{A}}}^T X_{S_{\mathcal{A}}})^{-1} M_2 \eta_{\gamma_{S_{\mathcal{A}}}}$$

$$\eta_{32}^j \overset{*}{=} \eta_{\beta_{j,S_{\mathcal{A}}}} (X_{S_{\mathcal{A}}}^T X_{S_{\mathcal{A}}})^{-1} M_2 L_2$$

$$\eta_{33}^j \overset{*}{=} \eta_{\beta_{j,S_{\mathcal{A}}}} (X_{S_{\mathcal{A}}}^T X_{S_{\mathcal{A}}})^{-1} M_2 \eta_{\gamma_{S_{\mathcal{A}}}}$$

Here, $M_1 = [\mathbb{I} + \eta_{\beta_{S_{\mathcal{A}},S_{\mathcal{A}}}} (X_{S_{\mathcal{A}}}^T X_{S_{\mathcal{A}}})^{-1}]^{-1}$, $M_2 = \left( \mathbb{I} - [\mathbb{I} + \eta_{\beta_{S_{\mathcal{A}},S_{\mathcal{A}}}} (X_{S_{\mathcal{A}}}^T X_{S_{\mathcal{A}}})^{-1}]^{-1} \cdot \eta_{\beta_{S_{\mathcal{A}},S_{\mathcal{A}}}} (X_{S_{\mathcal{A}}}^T X_{S_{\mathcal{A}}})^{-1} \right)$ and $L_2 = X_{S_{\mathcal{A}}}^T y$.

We look at the correlation values for the true support columns and non-support columns. Let us suppose the lower bound and upper bound for $C, C'$ (defined below) are $C_*$ and $C'_*$ respectively. We wish to identify the expressions for $C_*, C'_*$ and compare their values $C'_* < C_*$ so that we always pick the column from the true basis (steps follows similarly to Step 1). For $j \in S_*$ we have,

$$C_j = |S_{true}^j + \eta_1^j + \eta_2^j + \eta_{31}^j + \eta_{32}^j + \eta_{33}^j|$$

$$\geqslant \frac{1}{\sqrt{s}} ||S_{true}||_2 - ||\eta_1||_\infty - ||\eta_2||_\infty$$

$$- ||\eta_{31}||_\infty - ||\eta_{32}||_\infty - ||\eta_{33}||_\infty$$

Similarly, when $j \notin S_*$ we have,

$$C'_j = |(S'_{true})^j + (\eta_1^j)' + (\eta_2^j)' + (\eta_{31}^j)' + (\eta_{32}^j)' + (\eta_{33}^j)'|$$

$$\leqslant ||(S'_{true})^j||_2$$

$$+ ||(\eta_1')^j||_2 + ||(\eta_2')^j||_2 + ||(\eta_{31}')^j||_2 + ||(\eta_{32}')^j||_2 + ||(\eta_{33}')^j||_2$$

where we leverage the $L_2$-$L_\infty$ inequality, the triangle inequality, and the fact that each term $S_{true}^j, \eta_1^j, \eta_2^j, \eta_3^j$ is 1-dimensional and thus their $L_2, L_\infty$ norms are the same.

We now bound the signal values,

$$||S_{true}||_2 = ||X_{S_{\mathcal{A}}^c}^T [\mathbb{I} - P_S] y||_2$$

$$= ||X_{S_{\mathcal{A}}^c}^T [\mathbb{I} - P_S] (X_{S_*} \alpha_{S_*} + \epsilon)||_2$$

$$\overset{(1)}{=} ||X_{S_{\mathcal{A}}^c}^T (X_{S_{\mathcal{A}}^c} \alpha_{S_{\mathcal{A}}^c} + \epsilon)||_2$$

$$= n || \frac{X_{S_{\mathcal{A}}^c}^T (X_{S_{\mathcal{A}}^c} \alpha_{S_{\mathcal{A}}^c} + \epsilon)}{n}$$

$$- \frac{X_{S_{\mathcal{A}}^c}^T X_{S_{\mathcal{A}}}}{n} (\frac{X_{S_{\mathcal{A}}}^T X_{S_{\mathcal{A}}}}{n})^{-1} \frac{X_{S_{\mathcal{A}}}^T (X_{S_{\mathcal{A}}^c} \alpha_{S_{\mathcal{A}}^c} + \epsilon)}{n} ||_2$$

$$\overset{(2)}{\geqslant} n \Big( || \frac{X_{S_{\mathcal{A}}^c}^T X_{S_{\mathcal{A}}^c} \alpha_{S_{\mathcal{A}}^c}}{n} ||_2 - || \frac{X_{S_{\mathcal{A}}^c}^T e}{n} ||_2$$

$$- || \frac{X_{S_{\mathcal{A}}^c}^T X_{S_{\mathcal{A}}}}{n} (\frac{X_{S_{\mathcal{A}}}^T X_{S_{\mathcal{A}}}}{n})^{-1} \frac{X_{S_{\mathcal{A}}}^T [X_{S_{\mathcal{A}}^c} \alpha_{S_{\mathcal{A}}^c} + \epsilon]}{n} ||_2 \Big)$$

$$\overset{(3)}{\geqslant} n(1 - \zeta_{s+1}) ||\alpha_{S_{\mathcal{A}}^c}||_2 - n\sqrt{1 + \zeta_{s+1}} \kappa_\epsilon - \frac{n\zeta_{s+1}}{1 - \zeta_{s+1}} ||L_1||_2$$

$$\overset{(4)}{\geqslant} n(1 - \zeta_{s+1}) ||\alpha_{S_{\mathcal{A}}^c}||_2 - n\sqrt{1 + \zeta_{s+1}} \kappa_\epsilon$$

$$- \frac{n\zeta_{s+1}}{(1 - \zeta_{s+1})} [\zeta_{s+1} ||\alpha_{S_{\mathcal{A}}^c}||_2 + \sqrt{1 + \zeta_{s+1}} \kappa_\epsilon]$$

$$\overset{(5)}{\geqslant} n(1 - \zeta_{s+1}) ||\alpha_{S_{\mathcal{A}}^c}||_2 - n\sqrt{1 + \nu\zeta_{s+1}} \kappa_\epsilon$$

$$- n\zeta_{s+1}(1 + \nu\zeta_{s+1}) [\zeta_{s+1} ||\alpha_{S_{\mathcal{A}}^c}||_2 + \sqrt{1 + \nu\zeta_{s+1}} \kappa_\epsilon]$$

We observe *(1)* since the projection matrix nullifies the impact of the columns already picked by OMP. *(2)* follows from the reverse triangle inequality and by definition of $y$. *(3)* follows from Lemma 18 and 36 for the

first and second terms and Lemma 17, eigen-value bounds for the third term. *(4)* follows from Lemma 21. *(5)* follows by Lemma 22.

**Case B: Analysis for when chosen column** $j \in S_*^c$ Let us determine an upper bound for the correlation when the column chosen does not belong to the true basis. These columns are also referred to as non-basis columns. For the non-basis columns, we consider a single column each time. By definition, the non-basis columns are separate and thus disjoint from the true basis set, and thus $X_{S_*^c}$ is independent of $X_{S_*}$. Consider any $j \in S_*^c$;

$$
\begin{aligned}
|S_{true}^{j'}| &= ||X_j^T[\mathbb{I} - P_S](X_{S_*}\alpha_{S_*} + \epsilon)||_2 \\
&\overset{(1)}{\leqslant} n||\frac{X_j^T X_{S_\mathcal{A}^c}\alpha_{S_\mathcal{A}^c}}{n}||_2 + n||\frac{X_j^T e}{n}||_2 \\
&\quad + n||\frac{X_j^T X_{S_\mathcal{A}}}{n}(\frac{X_{S_\mathcal{A}}^T X_{S_\mathcal{A}}}{n})^{-1}\frac{X_{S_\mathcal{A}}^T[X_{S_\mathcal{A}^c}\alpha_{S_\mathcal{A}^c} + \epsilon]}{n}||_2 \\
&\overset{(2)}{\leqslant} n\zeta_{s+1}||\alpha_{S_\mathcal{A}^c}||_2 + n\sqrt{1 + \zeta_{s+1}}\kappa_\epsilon \\
&\quad + \frac{n\zeta_{s+1}}{1 - \zeta_{s+1}}[\zeta_{s+1}||\alpha_{S_\mathcal{A}^c}|| + \sqrt{1 + \zeta_{s+1}}\kappa_\epsilon] \\
&\overset{(3)}{\leqslant} n\zeta_{s+1}||\alpha_{S_\mathcal{A}^c}||_2 + n\sqrt{1 + \nu\zeta_{s+1}}\kappa_\epsilon \\
&\quad + n\zeta_{s+1}(1 + \nu\zeta_{s+1})[\zeta_{s+1}||\alpha_{S_\mathcal{A}^c}|| + \sqrt{1 + \nu\zeta_{s+1}}\kappa_\epsilon]
\end{aligned}
$$

*(1)* follows from the triangle inequality and since a 1-dimensional value $a$ has $||a||_\infty = ||a||_2$. We have reduced columns in *(1)* for the measurement matrix due to the projection matrix $\mathbb{I} - P_S$. *(2)* follows steps identical to $||S_{true}||_2$ except for the first term. The first term uses the approximate orthogonality property (Lemma 17) due to the Restricted Isometry. By the union bound over all $j \notin S_*$ we can conclude that the $||S_{true}^{j'}||_\infty$ is upper bounded. *(3)* holds due to Lemma 22. We bound the remaining noise terms in the Appendix section D.1.

**A note about the probabilistic concentration bounds:** We re-use common concentration bounds for bounding the values of maximal noise contributions across matrices and vectors. Since keeping track of such values is non-trivial we note that, the number of dimensions $p$ and the stopping point $s$ of our algorithm are both finite. Thus, given that we have several highly probable bounds with the form $1 - k_1 e^{k_2}$ where both $k_1, k_2$ are positive numbers. Thus, the overall probability bound for our result will hold with a similar form as well mainly because our iterations are finite and small ($s$)

**Combination of Case A and Case B:** Therefore, now, we can begin combining the two bounds for this induction. Or more appropriately we compare the lower bound and upper bounds of the correlations for the columns picked from the true basis and the non-basis sets respectively. We have $C_*' < C_*$. By combining the maximum values for any columns we can compare $\max_{j \notin S_*} C_j' < \max_{j \in S_*} C_j$ and substituting these terms we obtain,

$$
\begin{aligned}
&\frac{1}{\sqrt{s}}||S_{true}||_2 \\
&\geqslant ||S_{true}'||_2 + \left(||\eta_1^j||_2 + ||(\eta_1^j)'||_2\right) \\
&\quad + \left(||\eta_2^j||_2 + ||(\eta_2^j)'||_2\right) + \left(||\eta_{31}^j||_2 + ||(\eta_{31}^j)'||_2\right) \\
&\quad + \left(||\eta_{32}^j||_2 + ||(\eta_{32}^j)'||_2\right) + \left(||\eta_{33}^j||_2 + ||(\eta_{33}^j)'||_2\right)
\end{aligned}
$$

We simplify by substituting the noise bounds derived in previous Lemmas 23, 27, 24, 25, 26 as well as the signal norms $||S_{true}||_2$ and $||S_{true}'||_2$.

$$
\begin{aligned}
n||\alpha_{S_\mathcal{A}}^c||_2 \geqslant\ & n(\sqrt{s}+1)\zeta_{s+1}||\alpha_{S_\mathcal{A}}^c||_2 \\
& + n(\sqrt{s}+1)\sqrt{1+\nu\zeta_{s+1}}\kappa_\epsilon \\
& + n(\sqrt{s}+1)\zeta_{s+1}^2(1+\nu\zeta_{s+1})||\alpha_{S_\mathcal{A}}^c||_2 \\
& + n(\sqrt{s}+1)\zeta_{s+1}(1+\nu\zeta_{s+1})^{3/2}\kappa_\epsilon \\
& + s^{5/2}\kappa_1(1+\nu\zeta_{s+1})^{5/2}(1+(\nu+1)\zeta_{s+1})||\alpha_{S_\mathcal{A}}^c||_2 \\
& + s^{5/2}\kappa_1(1+\nu\zeta_{s+1})^2(1+(\nu+1)\zeta_{s+1})\kappa_\epsilon \\
& + s^{3/2}\kappa_{31}(1+\nu\zeta_{s+1})(1+(\nu+1)\zeta_{s+1}) \\
& + 2\sqrt{p}s\kappa_{32}(1+\nu\zeta_{s+1})^{3/2}||\alpha_{S_\mathcal{A}}^c||_2 \\
& + 2\sqrt{p}s\kappa_{32}(1+\nu\zeta_{s+1})\kappa_\epsilon \\
& + 4\frac{\sqrt{p}s^2}{n}\kappa_{33}(1+\nu\zeta_{s+1}) \\
& + 2\frac{\sqrt{ps}}{n}\kappa_2
\end{aligned}
$$

where,

$$
\kappa_1 = \kappa_s\kappa_{M_2}\sqrt{2\log\left(\frac{2s^2}{p_b}\right)}
$$

$$
\kappa_{31} = \kappa_s\kappa_{M_2}\sqrt{2\log\left(\frac{2s}{p_b}\right)},
$$

$$
\kappa_{32} = \kappa_p\kappa_{M_2}\sqrt{2\log\left(\frac{2(p-s)}{p_b}\right)},
$$

$$
\kappa_{33} = \frac{2\kappa_{M_2}s\kappa_s\kappa_p}{\kappa_n}\sqrt{2\log\left(\frac{2s}{p_b}\right)\log\left(\frac{2(p-s)}{p_b}\right)},
$$

$$
\kappa_2 = \kappa_p\sqrt{2\log\left(\frac{2(p-s)}{p_b}\right)}
$$

and $p_b$ is the failure probability used in the Lemma 36.

To identify bounds on the various terms in our system model, we allocate the left-hand side budget of $n||\alpha_{S_\mathcal{A}}^c||_2/2$ appropriately. Therefore, we obtain, the bound on $n$,

$$
\begin{aligned}
\frac{n||\alpha_{S_\mathcal{A}}^c||_2}{2} &\geqslant 2\sqrt{p}s\kappa_{32}(1+\nu\zeta_{s+1})^{3/2}||\alpha_{S_\mathcal{A}}^c||_2 \\
\rightarrow n &\geqslant 4\sqrt{p}s\kappa_{32}(1+\nu\zeta_{s+1})^{3/2} = \mathcal{O}(\sqrt{p})
\end{aligned}
\tag{3}
$$

Similarly, we can obtain a bound on the RIC, $\zeta_{s+1}$, by allocating the budget of $n||\alpha_{S_\mathcal{A}}^c||_2/4$

$$
\begin{aligned}
\frac{n||\alpha_{S_\mathcal{A}}^c||_2}{4} &\geqslant n(\sqrt{s}+1)\zeta_{s+1}||\alpha_{S_\mathcal{A}}^c||_2 \\
\rightarrow \zeta_{s+1} &\leqslant \frac{1}{4(\sqrt{s}+1)}
\end{aligned}
\tag{4}
$$

Now, the $\kappa_\varepsilon$ terms can be combined such that,

$$
\kappa_\varepsilon\left(2\sqrt{p}s\kappa_{32}(1+\nu\zeta_{s+1}) + n(\sqrt{s}+1)(1+\nu\zeta_{s+1})\right)
$$

$$+ n(\sqrt{s} + 1)\zeta_{s+1}(1 + \nu\zeta_{s+1})^{3/2}$$
$$+ s^{5/2}\kappa_1(1 + \nu\zeta_{s+1})^2(1 + (\nu + 1)\zeta_{s+1})\Big)$$
$$\leqslant \kappa_\epsilon \Big(\frac{n}{2\sqrt{1 + \nu\zeta_{s+1}}} + n(\sqrt{s} + 1)(1 + \nu\zeta_{s+1})$$
$$+ \frac{n(1 + \nu\zeta_{s+1})^{3/2}}{8}$$
$$+ s^{5/2}\kappa_1(1 + \nu\zeta_{s+1})^2(1 + (\nu + 1)\zeta_{s+1})\Big)$$

Here, these inequalities arise from the first two bounds on $\sqrt{p}$ and the RIC $\zeta_{s+1}$ (Eqns. 3 and 4). Clearly, the dominating term should be $n(\sqrt{s} + 1)(1 + \nu\zeta_{s+1})$ as long as,

$$s^{5/2}\kappa_1(1 + \nu\zeta_{s+1})(1 + (\nu + 1)\zeta_{s+1}) \leqslant n(\sqrt{s} + 1)$$
$$\to n \geqslant \frac{s^{5/2}\kappa_1(1 + \nu\zeta_{s+1})(1 + (\nu + 1)\zeta_{s+1})}{\sqrt{s} + 1} \tag{5}$$

Thus, the $\kappa_\epsilon$ bound can be obtained by assuming that,

$$4\kappa_\epsilon n(\sqrt{s} + 1)(1 + \nu\zeta_{s+1}) \leqslant \frac{1}{16}n\|\alpha_{S_{\mathcal{A}}}^c\|_2$$
$$\to \kappa_\epsilon \leqslant \frac{\|\alpha_{S_{\mathcal{A}}}^c\|_2}{16(\sqrt{s} + 1)(1 + \nu\zeta_{s+1})} \tag{6}$$

From the remaining terms which are independent of both $p$ and $n$, we pick the dominant term to obtain another bound on $n$ in terms of $s$,

$$2s^{5/2}\kappa_1(1 + \nu\zeta_{s+1})^{5/2}(1 + (\nu + 1)\zeta_{s+1})\|\alpha_{S_{\mathcal{A}}}^c\|_2 \leqslant \frac{n\|\alpha_{S_{\mathcal{A}}}^c\|_2}{8}$$
$$\to n \geqslant 16s^{5/2}\kappa_1(1 + \nu\zeta_{s+1})^{5/2}(1 + (\nu + 1)\zeta_{s+1}) \tag{7}$$

We can see that the remaining terms are extremely small and can be easily bounded by the remaining budget.

## F  Proof of Main Theorem 8

Before proving the convergence result in Theorem 8, we will include some additional results required by our analysis. First, we calculate the upper bound on the $l_2$ differential privacy sensitivity of the maximum absolute correlation in Lemma 28.

**Lemma 28.** *Consider a design matrix $X \in \mathbb{R}^{n \times p}$ satisfying RIP of order $s + 1$ and with RIC $\zeta_{s+1}$. Now given the correlation/gradient $C_j$ (as computed in Line 4 of algorithm 4) for any $j \in [p]$, we show that the $L_2$ sensitivity of $C_j$ is upper bounded by the following values,*

$$\Delta_2(C_j) \leqslant 1 + 2X_M\kappa_\varepsilon + 2\sqrt{s}X_M^2\Big(\frac{\|(\alpha_*)_{S_{\mathcal{A}}^c}\|_\infty}{1 - \zeta_{s+1}} + \frac{\sqrt{1 + \zeta_{s+1}}\kappa_\varepsilon}{1 - \zeta_{s+1}}\Big) \triangleq B_C$$

*where $\alpha_*$ is the underlying ground-truth model, $\kappa_\varepsilon = \sqrt{2\log(2n/p_b)}\sigma_\varepsilon$ and $\sigma_\varepsilon$ is the additive system error's standard deviation. We also need to assume that $\zeta_{s+1} < \frac{1}{\sqrt{s}}$ and $n > \frac{6s^2C_1X_M^5 y_M\sqrt{2\log(2s/p_b)}}{\mu_s^2(1 - \zeta_{s+1})^2}$. Here, the result holds with probability $1 - 3p_b$.*

*Proof.* Consider arbitrary adjacent datasets $X, X' \in \mathbb{R}^{n \times p}$ such that they differ in only their $k^{th}$ record/row. At any iteration of Line 4 in Algorithm 4, the set $S_{\mathcal{A}}$ of basis identified until now has been fixed. Furthermore, we note that the private model received from the server before the current iteration is also fixed. Denote this

model by $\alpha + \eta_\alpha$ where $\alpha$ is based on the currently predicted feature set $S_\mathcal{A}$ and $\eta_\alpha$ denotes the privacy-related noise added to the model using the NOISY-SMPC algorithm (Algorithm 1). Note, here we refer to the model as fixed due to the differential privacy post-processing wherein the model has already been privatized, and thus, future processing will not affect its privacy. In comparison with standardized private methods, we could consider this equivalent to using the same private model while computing gradient sensitivity in DP-SGD under FL constraints. Thus, the $l_2$ sensitivity of the correlation of the $j^{th}$ column (denoted as $C_j$) can be computed as such,

$$
\begin{aligned}
\Delta_2(C_j) &= \max_{k \in [n]} \| X_j^T y - (X')_j^T y' + X_j^T X_{S_\mathcal{A}}(\alpha + \eta_\alpha) - (X')_j^T X'_{S_\mathcal{A}}(\alpha + \eta_\alpha) \|_2 \\
&= \max_{k \in [n]} \| X_j^T (X_{S_*}\alpha_* + \varepsilon - X_{S_\mathcal{A}}\alpha_{S_\mathcal{A}}) - ((X')_j^T(X'_{S_*}\alpha_* + \varepsilon - X'_{S_\mathcal{A}}\alpha_{S_\mathcal{A}})) \| + \| \Delta A \eta_\alpha \| \\
&= \max_{k \in [n]} \| (X_{jk} - X'_{jk})\varepsilon_k \| + \| (X_j^T X_{S_\mathcal{A}} - (X')_j^T X'_{S_\mathcal{A}})((\alpha_*)_{S_\mathcal{A}} - \alpha) \| \\
&\quad + \| (X_j^T X_{S_\mathcal{A}^c} - (X')_j^T X'_{S_\mathcal{A}^c})(\alpha_*)_{S_\mathcal{A}^c} \| + \| \Delta A \eta_\alpha \| \\
&\leqslant 2 X_M \kappa_\varepsilon + \left( \| \Delta A^c (\alpha_*)_{S_\mathcal{A}^c} \| + \| \Delta A((\alpha_*)_{S_\mathcal{A}} - \alpha) \| \right) + \| \Delta A \eta_\alpha \|
\end{aligned}
$$

Here, $A = X_j^T X_{S_\mathcal{A}}, \Delta A = -X_{jk} X_{S_\mathcal{A},k} + X'_{jk} X'_{S_\mathcal{A},k}, \Delta A^c = -X_{jk} X_{S_\mathcal{A}^c,k} + X'_{jk} X'_{S_\mathcal{A}^c,k}$ $\alpha = (X_{S_\mathcal{A}}^T X_{S_\mathcal{A}})^{-1} X_{S_\mathcal{A}}^T y$ and $\kappa_\varepsilon = \sqrt{2 \log(2n/p_b)}\sigma_\varepsilon$ where $p_b$ is a small positive probability. We also see that

$$
\begin{aligned}
\| \Delta A \alpha \|_2 &= \| X_{jk} X_{S_\mathcal{A},k} + X'_{jk} X'_{S_\mathcal{A},k})\alpha \| \\
&\leqslant X_M (\| X_{S_\mathcal{A},k}\alpha \|_2 + \| X'_{S_\mathcal{A},k}\alpha \|_2) \\
&\leqslant \sqrt{s} X_M (\| X_{S_\mathcal{A},k}\alpha \|_\infty + \| X'_{S_\mathcal{A},k}\alpha \|_\infty) \\
&\leqslant 2\sqrt{s} X_M^2 \|\alpha\|_\infty
\end{aligned}
$$

Similar results can be derived for other terms in the expression for the correlation bound. Thus, we simplify the correlation bound such that,

$$
\Delta_2(C_j) \leqslant 2 X_M \kappa_\varepsilon + 2\sqrt{s} X_M^2 (\| (\alpha_*)_{S_\mathcal{A}^c} \|_\infty + \| \eta_\alpha \|_\infty + \| (\alpha_*)_{S_\mathcal{A}} - \alpha) \|_\infty)
$$

Now, suppose without loss of generality, suppose $\alpha$ is developed on matrix $X$ and feature set $S_\mathcal{A}$ then we have,

$$
\begin{aligned}
\alpha &= (X_{S_\mathcal{A}}^T X_{S_\mathcal{A}})^{-1} X_{S_\mathcal{A}}^T y \\
&= (X_{S_\mathcal{A}}^T X_{S_\mathcal{A}})^{-1} X_{S_\mathcal{A}}^T (X_{S_\mathcal{A}}(\alpha_*)_{S_\mathcal{A}} + X_{S_\mathcal{A}^c}(\alpha_*)_{S_\mathcal{A}^c} + \varepsilon) \\
&= (\alpha_*)_{S_\mathcal{A}} + (X_{S_\mathcal{A}}^T X_{S_\mathcal{A}})^{-1} X_{S_\mathcal{A}}^T (X_{S_\mathcal{A}^c}(\alpha_*)_{S_\mathcal{A}^c} + \varepsilon) \\
\rightarrow \| \alpha - (\alpha_*)_{S_\mathcal{A}} \|_2 &\leqslant \frac{\zeta_{s+1} \| (\alpha_*)_{S_\mathcal{A}^c} \|_2}{1 - \zeta_{s+1}} + \frac{\sqrt{1 + \zeta_{s+1}} \kappa_\varepsilon}{1 - \zeta_{s+1}}
\end{aligned}
$$

Here, the final set of inequalities follows from the inequalities *(1)-(10)* below.

Furthermore, we consider the value of $\eta_\alpha$, which denotes the noise added to the model via the NOISY-SMPC mechanism using noisy correlations. We can compute this exact value using the Kailath (Woodbury Variant) (Eqn. 31), and the value is defined as follows,

$$
\eta_\alpha = (X_{S_\mathcal{A}}^T X_{S_\mathcal{A}})^{-1}\left( \eta_{\gamma_{S_\mathcal{A}}} - \eta_{\beta_{S_\mathcal{A}}}(\mathbb{I} + (X_{S_\mathcal{A}}^T X_{S_\mathcal{A}})^{-1}\eta_{\beta_{S_\mathcal{A}}})^{-1}(X_{S_\mathcal{A}}^T X_{S_\mathcal{A}})^{-1}(X_{S_\mathcal{A}}^T y + \eta_{\gamma_{S_\mathcal{A}}}) \right)
$$

We now note the following $l_2$ bounds,

$$
\max_{k \in [n]} \| \Delta A \|_2 \overset{(1)}{\leqslant} 2\sqrt{s} X_M^2
$$

$$
\max_{k \in [n]} \| X_j^T y - (X')_j^T y' \| \overset{(2)}{\leqslant} 2 X_M y_M
$$

$$\|\frac{X_{S_{\mathcal{A}}}^T y}{n}\| \overset{(3)}{\leqslant} \sqrt{s} X_M y_M$$

$$\|\alpha\| = \|(X_{S_{\mathcal{A}}}^T X_{S_{\mathcal{A}}})^{-1} X_{S_{\mathcal{A}}}^T y\| = \|(X_{S_{\mathcal{A}}}^T X_{S_{\mathcal{A}}})^{-1} X_{S_{\mathcal{A}}}^T (X_{S_*} \alpha_* + \varepsilon)\| \overset{(4)}{\leqslant} \frac{1 + \zeta_{s+1}}{1 - \zeta_{s+1}} \|\alpha_*\| + \frac{\sqrt{1 + \zeta_{s+1}}}{1 - \zeta_{s+1}} \kappa_\varepsilon$$

$$\triangleq \kappa_\alpha$$

$$\|(\frac{X_{S_{\mathcal{A}}}^T X_{S_{\mathcal{A}}}}{n})^{-1}\| \overset{(5)}{\leqslant} \frac{1}{1 - \zeta_{s+1}}$$

$$\|\eta_{\gamma_{S_{\mathcal{A}}}}\| \overset{(6)}{\leqslant} \frac{\sqrt{2s \log(2s/p_b)} X_M y_M}{\mu_s}$$

$$\|\eta_{\beta_{S_{\mathcal{A}}}}\| \overset{(7)}{\leqslant} \frac{s C_1 X_M^2}{\mu_s}$$

$$\|(\mathbb{I} + (X_{S_{\mathcal{A}}}^T X_{S_{\mathcal{A}}})^{-1} \eta_{\beta_{S_{\mathcal{A}}}})^{-1}\| \overset{(8)}{\leqslant} 1$$

$$\|\frac{X_{S_{\mathcal{A}}}^T X_{S_{\mathcal{A}}}}{n}\| \overset{(9)}{\leqslant} 1 + \zeta_{s+1}$$

$$\|\frac{X_P^T X_Q}{n}\| \overset{(10)}{\leqslant} \zeta_{s+1}$$

Here, inequalities *(1)-(3)* follow from the boundedness requirement on elements of $X, y$. Inequalities *(4)-(5)* follows from the RIP condition on $X/\sqrt{n}$ matrix. Specifically, in *(4)*, $\kappa_\varepsilon = \sqrt{2 \log(2n/p_b)} \sigma_\varepsilon$ generated from Lemma 36. Inequality *(6)* follows from Lemma 36 where elements in $\eta_{\gamma_{S_{\mathcal{A}}}}$ have distribution $\mathcal{N}(0, \frac{\sqrt{2s \log(2s/p_b)}}{\mu_s})$. Inequality *(7)* follows by noting the inequality of largest singular value inequality for a random matrix inequality from Rudelson & Vershynin (2008) where the elements in $\eta_{\beta_{S_{\mathcal{A}}}}$ are from distribution $\mathcal{N}(0, \frac{\sqrt{s}}{\mu_s})$. *(8)* follows from Lemma 19. *(9)* follows from the RIP property of $\boldsymbol{X}/\sqrt{n}$. Combining inequalities above, we can find the bound on $\|\eta_\alpha\|$,

$$\|\eta_\alpha\| \leqslant \frac{\sqrt{2s \log(2s/p_b)} X_M y_M}{n \mu_s (1 - \zeta_{s+1})} + \frac{s C_1 X_M^2}{\mu_s (1 - \zeta_{s+1})^2} \cdot (\frac{\sqrt{2s \log(2s/p_b)} X_M y_M}{n \mu_s} + \frac{\sqrt{s} X_M y_M}{n})$$

$$= \frac{\sqrt{2s \log(2s/p_b)} X_M y_M}{n \mu_s (1 - \zeta_{s+1})} + \frac{s^{3/2} C_1 X_M^3 y_M}{n \mu_s (1 - \zeta_{s+1})^2} (\frac{\sqrt{2 \log(2s/p_b)}}{\mu_s} + 1)$$

where this bound holds with probability $1 - 2p_b$ and $p_b$ is a small positive probability. We quickly note that given the $n$ (sample size) in the denominator, then we can say that $\|\Delta A \eta_\alpha\| \leqslant 1$ if $n > \frac{6s^2 C_1 X_M^5 y_M \sqrt{2 \log(2s/p_b)}}{\mu_s^2 (1 - \zeta_{s+1})^2}$. We now continue finding the upper bound of $\Delta_2(C_j)$ using the inequalities computed above,

$$\Delta_2(C_j) \leqslant 1 + 2 X_M \kappa_\varepsilon + 2\sqrt{s} X_M^2 (\|(\alpha_*)_{S_{\mathcal{A}}^c}\|_\infty + \frac{\zeta_{s+1} \|(\alpha_*)_{S_{\mathcal{A}}^c}\|_2}{1 - \zeta_{s+1}} + \frac{\sqrt{1 + \zeta_{s+1}} \kappa_\varepsilon}{1 - \zeta_{s+1}})$$

$$= 1 + 2 X_M \kappa_\varepsilon + 2\sqrt{s} X_M^2 (\frac{\|(\alpha_*)_{S_{\mathcal{A}}^c}\|_\infty}{1 - \zeta_{s+1}} + \frac{\sqrt{1 + \zeta_{s+1}} \kappa_\varepsilon}{1 - \zeta_{s+1}})$$

$$= 1 + 2 X_M \kappa_\varepsilon + 2\sqrt{s} X_M^2 \kappa_\alpha'$$

$$\triangleq B_C$$

where we assume that $\zeta_{s+1} < \frac{1}{\sqrt{s}}$ and $\kappa_\alpha' = (\frac{\|(\alpha_*)_{S_{\mathcal{A}}^c}\|_\infty}{1 - \zeta_{s+1}} + \frac{\sqrt{1 + \zeta_{s+1}} \kappa_\varepsilon}{1 - \zeta_{s+1}})$.

$\square$

## F.1 Performance of Algorithm 4

Here, similar to main theorem 7, we will demonstrate that Algorithm 4 can recover the correct basis w.h.p. Again, we will use induction to prove our result. We note from our proof sketch that the first induction step

for both the current theorem and main theorem 7 are the same (since the correlation and the gradient have the same value for linear regression model. Thus, we can immediately proceed to prove the induction step. Before proceeding, we restate the preliminary steps of the induction.

**Step 1:** Here, we show that in the first step of the algorithm involving picking the feature with the highest absolute correlation, we will pick the correct feature (*i.e.,* the one from the true basis set). Similar to Theorem 7, we find a condition on the sample size such that the lower bound of the maximum absolute correlation of the true feature set is greater than the upper bound of the maximum absolute correlation of the wrong feature set w.h.p. Thus, we need to find the conditions that ensure $C_* > C'_*$ where $C_*$ and $C'_*$ are lower and upper bounds of $C$ and $C'$ respectively. Here,

$$C = \max_{j \in S_*} |X_j^T y + \eta_{(\gamma_0^-)_j}|$$
$$C' = \max_{j \notin S_*} |X_j^T y + \eta_{(\gamma_0^-)_j}|$$

and the rest of the notation follows from Algorithms 4 and 3. Note that the proof follows exactly as Theorem 7's step 1 and the sample size condition remains the same.

**Step (l+1):** By induction, $C_* > C'_*$ holds for all steps from 1 to $l$. We will now show that a similar result holds for the $(l+1)^{th}$ step for Algorithm 4.

We first note the private correlation value is,

$$\begin{aligned}
C_j &= X_j^T y - X_j^T X_{S_\mathcal{A}} (X_{S_\mathcal{A}}^T X_{S_\mathcal{A}} + \eta_{\beta_{S_\mathcal{A}, S_\mathcal{A}}})^{-1} (X_{S_\mathcal{A}}^T y + \eta_{\gamma_{S_\mathcal{A}}}) + \eta_{C_j} \\
&= X_j^T y - X_j^T X_{S_\mathcal{A}} \left( (X_{S_\mathcal{A}}^T X_{S_\mathcal{A}})^{-1} - (X_{S_\mathcal{A}}^T X_{S_\mathcal{A}})^{-1} (\mathbb{I} + \eta_{\beta_{S_\mathcal{A}, S_\mathcal{A}}} (X_{S_\mathcal{A}}^T X_{S_\mathcal{A}})^{-1})^{-1} \eta_{\beta_{S_\mathcal{A}, S_\mathcal{A}}} (X_{S_\mathcal{A}}^T X_{S_\mathcal{A}})^{-1} \right) (X_{S_\mathcal{A}}^T y + \eta_{\gamma_{S_\mathcal{A}}}) \\
&\quad + \eta_{C_j} \\
&= X_j^T y - X_j^T X_{S_\mathcal{A}} (X_{S_\mathcal{A}}^T X_{S_\mathcal{A}})^{-1} X_{S_\mathcal{A}}^T y - X_j^T X_{S_\mathcal{A}} (X_{S_\mathcal{A}}^T X_{S_\mathcal{A}})^{-1} \eta_{\gamma_{S_\mathcal{A}}} + \eta_{C_j} \\
&\quad - X_j^T X_{S_\mathcal{A}} (X_{S_\mathcal{A}}^T X_{S_\mathcal{A}})^{-1} (\mathbb{I} + \eta_{\beta_{S_\mathcal{A}, S_\mathcal{A}}} (X_{S_\mathcal{A}}^T X_{S_\mathcal{A}})^{-1})^{-1} \eta_{\beta_{S_\mathcal{A}, S_\mathcal{A}}} (X_{S_\mathcal{A}}^T X_{S_\mathcal{A}})^{-1} (X_{S_\mathcal{A}}^T y + \eta_{\gamma_{S_\mathcal{A}}})
\end{aligned}$$

where $\eta_{C_j} \sim \mathcal{N}(0, \sigma_C^2)$, $\sigma_C = \frac{\sqrt{p} B_C}{\mu_p}$, $B_C = \kappa'_\alpha \sqrt{s} + 2 X_M y_M$ and $\eta_{\gamma_{S_\mathcal{A}}} \sim \mathcal{N}(0, \frac{s}{\mu_s^2} \mathbb{I}_s)$ and $\eta_{\beta_{S_\mathcal{A}, S_\mathcal{A}}} \sim \mathcal{N}(0, \frac{s}{\mu_s^2} \mathbb{I}_s)$. Note that the first term is the true correlation and can be re-written as $X_j^T (\mathbb{I} - P) X_{S_\mathcal{A}}^T y$ where $P = X_{S_\mathcal{A}} (X_{S_\mathcal{A}} X_{S_\mathcal{A}})^{-1} X_{S_\mathcal{A}}^T$ is the projection matrix.

**Case A: Analysis for when chosen column $j \in S_\mathcal{A}^c$ (true feature set except for features already chosen):**

First, we will find a lower bound for the maximum correlation/gradient when the features are chosen from the true set (except for the features already chosen),

$$\begin{aligned}
\|C_j\|_\infty &\geq \frac{1}{\sqrt{s}} \|C_j\|_2 \\
&\overset{(1)}{\geq} \frac{1}{\sqrt{s}} \Big( \|X_{S_\mathcal{A}^c}^T (\mathbb{I} - P) y\|_2 - \|X_{S_\mathcal{A}^c}^T X_{S_\mathcal{A}} (X_{S_\mathcal{A}}^T X_{S_\mathcal{A}})^{-1} \eta_{\gamma_{S_\mathcal{A}}}\|_2 - \|\eta_{C_j}\|_2 \\
&\quad - \|X_{S_\mathcal{A}^c}^T X_{S_\mathcal{A}} (X_{S_\mathcal{A}}^T X_{S_\mathcal{A}})^{-1} (\mathbb{I} + \eta_{\beta_{S_\mathcal{A}, S_\mathcal{A}}} (X_{S_\mathcal{A}}^T X_{S_\mathcal{A}})^{-1}) \eta_{\beta_{S_\mathcal{A}, S_\mathcal{A}}} (X_{S_\mathcal{A}}^T X_{S_\mathcal{A}})^{-1} (X_{S_\mathcal{A}}^T y + \eta_{\gamma_{S_\mathcal{A}}})\|_2 \Big) \\
&\overset{(2)}{\geq} \frac{1}{\sqrt{s}} \|X_{S_\mathcal{A}^c}^T (\mathbb{I} - P) y\|_2 - \frac{1}{\sqrt{s}} \|X_{S_\mathcal{A}^c}^T X_{S_\mathcal{A}} (X_{S_\mathcal{A}}^T X_{S_\mathcal{A}})^{-1} \eta_{\gamma_{S_\mathcal{A}}}\|_2 - \|\eta_{C_j}\|_\infty \\
&\quad - \frac{1}{\sqrt{s}} \|X_{S_\mathcal{A}^c}^T X_{S_\mathcal{A}} (X_{S_\mathcal{A}}^T X_{S_\mathcal{A}})^{-1} (\mathbb{I} + \eta_{\beta_{S_\mathcal{A}, S_\mathcal{A}}} (X_{S_\mathcal{A}}^T X_{S_\mathcal{A}})^{-1})^{-1} \eta_{\beta_{S_\mathcal{A}, S_\mathcal{A}}} (X_{S_\mathcal{A}}^T X_{S_\mathcal{A}})^{-1} (X_{S_\mathcal{A}}^T y + \eta_{\gamma_{S_\mathcal{A}}})\|_2 \\
&\overset{(3)}{\geq} \frac{n(1 - \zeta_{s+1}) \|\alpha_{S_\mathcal{A}^c}\|_2}{\sqrt{s}} - \frac{1}{\sqrt{s}} \|X_{S_\mathcal{A}^c}^T \varepsilon\|_2 - \frac{1}{\sqrt{s}} \|X_{S_\mathcal{A}^c}^T X_{S_\mathcal{A}} (X_{S_\mathcal{A}}^T X_{S_\mathcal{A}})^{-1} X_{S_\mathcal{A}}^T (X_{S_\mathcal{A}^c} \alpha_{S_\mathcal{A}^c} + \varepsilon)\|_2 \\
&\quad - \frac{\zeta_{s+1}}{\sqrt{s}(1 - \zeta_{s+1})} \|\eta_{\gamma_{S_\mathcal{A}}}\|_2 - \frac{\sqrt{p} B_C \kappa_p}{\mu_p} - \frac{\zeta_{s+1}}{\sqrt{s}(1 - \zeta_{s+1})^2} \|\eta_{\beta_{S_\mathcal{A}, S_\mathcal{A}}} (X_{S_\mathcal{A}}^T y + \eta_{\gamma_{S_\mathcal{A}}})\|_2
\end{aligned}$$

$$\stackrel{(4)}{\geqslant} \frac{n(1-\zeta_{s+1})\|\alpha_{S_{\mathcal{A}}^c}\|_2}{\sqrt{s}} - \frac{n\zeta_{s+1}^2\|\alpha_{S_{\mathcal{A}}^c}\|_2}{\sqrt{s}(1-\zeta_{s+1})} - \frac{\|X_{S_{\mathcal{A}}^c}^T\varepsilon\|_2}{\sqrt{s}} - \frac{\zeta_{s+1}\|X_{S_{\mathcal{A}}}^T\varepsilon\|_2}{\sqrt{s}(1-\zeta_{s+1})}$$

$$- \frac{\zeta_{s+1}}{\sqrt{s}(1-\zeta_{s+1})}\|\eta_{\gamma_{S_{\mathcal{A}}}}\|_2 - \frac{\sqrt{p}B_C\kappa_p}{\mu_p} - \frac{\zeta_{s+1}}{\sqrt{s}(1-\zeta_{s+1})^2}\|\eta_{\beta_{S_{\mathcal{A}},S_{\mathcal{A}}}}(X_{S_{\mathcal{A}}}^Ty + \eta_{\gamma_{S_{\mathcal{A}}}})\|_2$$

$$\stackrel{(5)}{\geqslant} \frac{n(1-\zeta_{s+1})\|\alpha_{S_{\mathcal{A}}^c}\|_2}{\sqrt{s}} - \frac{n\zeta_{s+1}^2\|\alpha_{S_{\mathcal{A}}^c}\|_2}{\sqrt{s}(1-\zeta_{s+1})} - \frac{1}{\sqrt{s}}n(\sqrt{1+\zeta_{s+1}})\kappa_\varepsilon\Big(1 + \frac{\zeta_{s+1}}{1-\zeta_{s+1}}\Big)$$

$$- \frac{2X_My_M\kappa_s\zeta_{s+1}}{\mu_s(1-\zeta_{s+1})} - \frac{\sqrt{p}B_C\kappa_p}{\mu_p} - \frac{\zeta_{s+1}C_1\sqrt{s}\kappa_s}{\mu_s\sqrt{s}(1-\zeta_{s+1})^2}\Big((1+\zeta_{s+1})\|\alpha_{S_*}\|_2 + \kappa_\varepsilon + \frac{2X_My_M\kappa_s}{\mu_s}\Big)$$

$$\stackrel{(6)}{=} \frac{n(1-\zeta_{s+1})\|\alpha_{S_{\mathcal{A}}^c}\|_2}{\sqrt{s}} - \frac{n\zeta_{s+1}^2\|\alpha_{S_{\mathcal{A}}^c}\|_2}{\sqrt{s}(1-\zeta_{s+1})} - \frac{n(\sqrt{1+\zeta_{s+1}})\kappa_\varepsilon}{\sqrt{s}(1-\zeta_{s+1})}$$

$$- \frac{2X_My_M\kappa_s\zeta_{s+1}}{\mu_s(1-\zeta_{s+1})} - \frac{\sqrt{p}B_C\kappa_p}{\mu_p} - \frac{\zeta_{s+1}C_1\kappa_s}{\mu_s(1-\zeta_{s+1})^2}\Big((1+\zeta_{s+1})\|\alpha_{S_*}\|_2 + \kappa_\varepsilon + \frac{2X_My_M\kappa_s}{\mu_s}\Big)$$

*(1)* follows by reverse triangle inequality. *(2)* holds due to the $l_2$-$l_\infty$ inequality. *(3)-(4)* follow from the RIP properties of $X/\sqrt{n}$ as described in the inequalities in the proof of Lemma 28 alongside the Lemma 36 where $\kappa_p' = \sqrt{2\log(\frac{2p}{p_b})}$ where $p_b$ is a small positive failure probability. *(5)* follows from Lemma 36 where $\kappa_s' = \sqrt{2\log(\frac{2s}{p_b})}$ and Theorem 2.4 (upper bound on largest singular value of random matrix) from Rudelson & Vershynin (2008).

**Case B: Analysis for when chosen column $j \notin S_*^c$:**

We now compute the upper bound on the correlations of the features not in the true set,

$$\|C_j\|_\infty \leqslant \|C_j\|_2$$
$$\leqslant \|X_j^T(\mathbb{I} - P)y\|_2 + \|X_j^TX_{S_{\mathcal{A}}}(X_{S_{\mathcal{A}}}^TX_{S_{\mathcal{A}}})^{-1}\eta_{\gamma_{S_{\mathcal{A}}}}\|_2 + \|\eta_{C_j}\|_2$$
$$- \|X_j^TX_{S_{\mathcal{A}}}(X_{S_{\mathcal{A}}}^TX_{S_{\mathcal{A}}})^{-1}(\mathbb{I} + \eta_{\beta_{S_{\mathcal{A}},S_{\mathcal{A}}}}(X_{S_{\mathcal{A}}}^TX_{S_{\mathcal{A}}})^{-1})\eta_{\beta_{S_{\mathcal{A}},S_{\mathcal{A}}}}(X_{S_{\mathcal{A}}}^TX_{S_{\mathcal{A}}})^{-1}(X_{S_{\mathcal{A}}}^Ty + \eta_{\gamma_{S_{\mathcal{A}}}})\|_2\Big)$$
$$\stackrel{(1)}{\leqslant} \|X_j^T(X_{S_{\mathcal{A}}^c}\alpha_{S_{\mathcal{A}}^c} + \varepsilon)\|_2 + \|X_j^TX_{S_{\mathcal{A}}}(X_{S_{\mathcal{A}}}^TX_{S_{\mathcal{A}}})^{-1}X_{S_{\mathcal{A}}}^T(X_{S_{\mathcal{A}}^c}\alpha_{S_{\mathcal{A}}^c} + \varepsilon)\| + \frac{2\sqrt{s}X_My_M\kappa_s\zeta_{s+1}}{\mu_s(1-\zeta_{s+1})} + \frac{\sqrt{p}B_C\kappa_p}{\mu_p}$$
$$+ \frac{\zeta_{s+1}C_1\sqrt{s}\kappa_s}{\mu_s(1-\zeta_{s+1})^2}\Big((1+\zeta_{s+1})\|\alpha_{S_*}\|_2 + \kappa_\varepsilon + \frac{2X_My_M\kappa_s}{\mu_s}\Big)$$
$$\stackrel{(2)}{\leqslant} \frac{n\zeta_{s+1}\|\alpha_{S_{\mathcal{A}}^c}\|_2 + n\sqrt{1+\zeta_{s+1}}\kappa_\varepsilon}{1-\zeta_{s+1}} + \frac{2\sqrt{s}X_My_M\kappa_s\zeta_{s+1}}{\mu_s(1-\zeta_{s+1})} + \frac{\sqrt{p}B_C\kappa_p}{\mu_p}$$
$$+ \frac{\zeta_{s+1}C_1\sqrt{s}\kappa_s}{\mu_s(1-\zeta_{s+1})^2}\Big((1+\zeta_{s+1})\|\alpha_{S_*}\|_2 + \kappa_\varepsilon + \frac{2X_My_M\kappa_s}{\mu_s}\Big)$$

*(1)-(2)* follow the same inequalities as the one developed in case A.

**Combining Case A and Case B:** We compare the upper bound of Case B with the lower bound of Case A to find the conditions on the system parameters and mainly the sample size.

$$n\|\alpha_{S_{\mathcal{A}}^c}\|_2 \geqslant n\zeta_{s+1}\|\alpha_{S_{\mathcal{A}}^c}\|_2\Big(1 + \frac{\sqrt{s}}{1-\zeta_{s+1}}\Big) + (\sqrt{s}+1)\Bigg(\frac{n(\sqrt{1+\zeta_{s+1}})\kappa_\varepsilon}{1-\zeta_{s+1}} + \frac{2\sqrt{s}X_My_M\kappa_s\zeta_{s+1}}{\mu_s(1-\zeta_{s+1})} + \frac{\sqrt{p}B_C\kappa_p}{\mu_p}$$
$$+ \frac{\zeta_{s+1}C_1\sqrt{s}\kappa_s}{\mu_s(1-\zeta_{s+1})^2}\Big((1+\zeta_{s+1})\|\alpha_{S_*}\|_2 + \kappa_\varepsilon + \frac{2X_My_M\kappa_s}{\mu_s}\Big)\Bigg)$$

We now compare specific terms so as to maintain the inequality above and set $\zeta_{s+1} < \frac{1}{\sqrt{s}+1}$

$$\frac{n\|\alpha_{S_{\mathcal{A}}^c}\|_2}{2} > \frac{(\sqrt{s}+1)\sqrt{p}B_c\kappa_p}{\mu_p}$$

$$\frac{n\|\alpha_{S_{\mathcal{A}}^c}\|_2}{6} > \kappa_\varepsilon(1+\nu\zeta_{s+1})^{3/2}\Big(n(\sqrt{s}+1) + \frac{C_1\sqrt{s}\kappa_s\sqrt{1+\zeta_{s+1}}}{\mu_s}\Big)$$

$$> (\sqrt{s}+1)\kappa_\varepsilon\Big(\frac{n\sqrt{1+\zeta_{s+1}}}{1-\zeta_{s+1}} + \frac{\zeta_{s+1}C_1\sqrt{s}\kappa_s}{\mu_s(1-\zeta_{s+1})^2}\Big)$$

$$\frac{n\|\alpha_{S_{\mathcal{A}}^c}\|_2}{6} > \frac{2\sqrt{s}X_My_M\kappa_s}{\mu_s(1-\zeta_{s+1})} > \frac{2\sqrt{s}(\sqrt{s}+1)X_My_M\kappa_s\zeta_{s+1}}{\mu_s(1-\zeta_{s+1})}$$

$$\frac{n\|\alpha_{S_{\mathcal{A}}^c}\|_2}{6} > \frac{C_1\sqrt{s}\kappa_s}{\mu_s(1-\zeta_{s+1})^2}\Big((1+\zeta_{s+1})\|\alpha_{S_*}\|_2 + \frac{2X_My_M\kappa_s}{\mu_s}\Big)$$

$$> \frac{\zeta_{s+1}(\sqrt{s}+1)C_1\sqrt{s}\kappa_s}{\mu_s(1-\zeta_{s+1})^2}\Big((1+\zeta_{s+1})\|\alpha_{S_*}\|_2 + \frac{2X_My_M\kappa_s}{\mu_s}\Big)$$

Simplifying these conditions, we obtain the following,

$$n > \frac{2(\sqrt{s}+1)\sqrt{p}B_c\kappa_p}{\mu_p\|\alpha_{S_{\mathcal{A}}^c}\|_2} = \frac{2(\sqrt{s}+1)(\sqrt{s}\kappa_\alpha' + 2X_My_M)\sqrt{p}\kappa_p}{\mu_p\|\alpha_{S_{\mathcal{A}}^c}\|_2}$$

$$\kappa_\varepsilon < \frac{\|\alpha_{S_{\mathcal{A}}^c}\|_2}{6(1+\nu\zeta_{s+1})^{3/2}\Big(\sqrt{s}+1+\frac{C_1\sqrt{s}\kappa_s\sqrt{1+\zeta_{s+1}}}{n\mu_s}\Big)}$$

Note that here, we do not need to work with the remaining terms of order $\sqrt{s}$ since we know $n$ and $\sqrt{p}$ are much larger than $\sqrt{s}$.

## G   Proofs of Risk and Estimation Error for the SPriFed-OMP algorithm

Before we provide the risk and estimation error analyses we first state and derive a commonly referred to result (Lemma 3 in Meinshausen & Yu (2009) and Lemma 3.1 in Wasserman & Roeder (2009)) regarding the 2-norm of the product $X_{S_*}^T\epsilon$ where $X_{S_*}$ refers to the features coinciding with the true model indices and $\epsilon$ is the model error as shown in Section 2.

**Lemma 29.** *The 2-norm of the product $X_{S_*}^T\epsilon$ is upper bounded by $\sqrt{ns}\sigma X_M(2\log(2s/p_b))^{1/4}$ with probability $1-p_b$ where $p_b$ is the base probability and the terms in the product are defined in Section 2.*

**Proof:**

$$\|X_{S_*}^T\epsilon\|_2 \overset{(1)}{\leqslant} \sqrt{n}\sqrt{\sum_{j\in S_*} Z_j^2}$$

$$\overset{(2)}{\leqslant} \sqrt{ns}\sqrt{\max_{j\in S_*} Z_j^2}$$

$$\overset{(3)}{\leqslant} \sqrt{2ns\log(2s/p_b))}\sigma_\epsilon X_M$$

*(1)* sets $Z_j = \frac{1}{\sqrt{n}}X_j^T\epsilon$.  *(2)* follows by definition of a maximum.  Now, $\mu(Z_j) = 0$ and $Var(Z_j) = \frac{1}{n}\sum_{i=1}^n X_{ij}^2\sigma_\epsilon^2 \leqslant X_M^2\sigma_\epsilon^2$ as the absolute value of coordinates $X_{ij}$ is bounded by $X_M$. Therefore, *(3)* follows by the Lemma 36 with probability $1-p_b$.

### G.1 Proof of Estimation Error Theorem 9

From Theorem 7 we know given that if satisfy the theorem's assumptions we can recover the true model basis with high probability. We will function under this high probability event and assume that the predicted basis $S_\mathcal{A}$ is equal to $S_*$ while computing the estimation error. Thus, by *SPriFed-OMP* our estimation error can be derived as below,

$$
\begin{aligned}
\Delta\alpha &:= ||\hat{\alpha} - \alpha_{S_*}||_2 \\
&= ||(\beta_{S_*})^{-1}\gamma_{S_*} - \alpha_{S_*}||_2 \\
&= ||(X_{S_*}^T X_{S_*} + \eta_\beta)^{-1}(X_{S_*}^T y + \eta_{\gamma_{S_*}}) - \alpha_{S_*}||_2 \\
&\overset{(1)}{=} ||(X_{S_*}^T X_{S_*})^{-1}(X_{S_*}^T y + \eta_{\gamma_{S_*}}) - \alpha_{S_*} \\
&\quad - (X_{S_*}^T X_{S_*})^{-1}(\mathbb{I} + \eta_\beta(X_{S_*}^T X_{S_*})^{-1})^{-1} \\
&\quad \eta_\beta(X_{S_*}^T X_{S_*})^{-1}(X_{S_*}^T y + \eta_{\gamma_{S_*}})||_2 \\
&\overset{(2)}{=} ||(X_{S_*}^T X_{S_*})^{-1}X_{S_*}^T \epsilon + (X_{S_*}^T X_{S_*})^{-1}\eta_{\gamma_{S_*}} \\
&\quad - (X_{S_*}^T X_{S_*})^{-1}(\mathbb{I} + \eta_\beta(X_{S_*}^T X_{S_*})^{-1})^{-1} \\
&\quad \cdot \eta_\beta(X_{S_*}^T X_{S_*})^{-1}(X_{S_*}^T y + \eta_{\gamma_{S_*}})||_2 \\
&\overset{(3)}{\leq} ||(\frac{X_{S_*}^T X_{S_*}}{n})^{-1}\frac{X_{S_*}^T \epsilon}{n}|| + ||(\frac{X_{S_*}^T X_{S_*}}{n})^{-1}\frac{\eta_{\gamma_{S_*}}}{n}||_2 \\
&\quad + ||(\frac{X_{S_*}^T X_{S_*}}{n})^{-1}(\mathbb{I} + \frac{\eta_\beta}{n}(\frac{X_{S_*}^T X_{S_*}}{n})^{-1})^{-1} \\
&\quad \cdot \frac{\eta_\beta}{n}(\frac{X_{S_*}^T X_{S_*}}{n})^{-1}(\frac{X_{S_*}^T y}{n} + \frac{\eta_{\gamma_{S_*}}}{n})||_2 \\
&\overset{(4)}{\leq} \frac{(1-\zeta_{s+1})\sqrt{2ns\log(2s/p_b)}\sigma_\epsilon X_M}{n} + \frac{\kappa_s s\sqrt{2\log(\frac{2s}{p_b})}}{n(1-\zeta_{s+1})} \\
&\quad + \frac{\kappa_M}{(1-\zeta_{s+1})}||\frac{\eta_\beta}{n}(\frac{X_{S_*}^T X_{S_*}}{n})^{-1}(\frac{X_{S_*}^T y}{n} + \frac{\eta_{\gamma_{S_*}}}{n})||_2 \\
&\overset{(5)}{\leq} \frac{(1-\zeta_{s+1})\sqrt{2s\log(2s/p_b)}\sigma_\epsilon X_M}{\sqrt{n}} + \frac{\kappa_s s\sqrt{2\log(\frac{2s}{p_b})}}{n(1-\zeta_{s+1})} \\
&\quad + \frac{\kappa_M}{(1-\zeta_{s+1})^2} \cdot \frac{s^{3/2}\kappa_s\sqrt{2\log(\frac{2s^2}{p_b})}}{n}||(\frac{X_{S_*}^T y}{n} + \frac{\eta_{\gamma_{S_*}}}{n})||_2 \\
&\overset{(6)}{\leq} \frac{(1-\zeta_{s+1})\sqrt{2s\log(2s/p_b)}\sigma_\epsilon X_M}{\sqrt{n}} + \frac{\kappa_s s\sqrt{2\log(\frac{2s}{p_b})}}{n(1-\zeta_{s+1})} \\
&\quad + \frac{s^{3/2}\kappa_s\kappa_M\sqrt{2\log(\frac{2s^2}{p_b})}}{n(1-\zeta_{s+1})^2}\Big((1+\zeta_{s+1})||\alpha_{S_*}||_2 \\
&\quad + \sqrt{1+\zeta_{s+1}}\kappa_\epsilon + \frac{\kappa_s\sqrt{s}\sqrt{2\log(\frac{2s}{p_b})}}{n}\Big) \\
&= \mathcal{O}\Big(\sqrt{\frac{s\log(s)}{n}}\Big)
\end{aligned}
$$

*(1)* follows by the Woodbury Identity (Eqn. 31). *(2)* follows since $(X_{S_*}^T X_{S_*})^{-1}X_{S_*}^T y = \alpha_{S_*} + X_{S_*}^T \epsilon$ as $y = X_{S_*}\alpha_{S_*} + \epsilon$. *(3)* follows by the triangle inequality and division and multiplication by $n$. *(4)* is based on bounds on the eigenvalues of $X_{S_*}^T X_{S_*}$, the Lemma 19 as well as by the Gaussian concentration bounds for the noise term $\eta_{\gamma_{S_*}}; \eta_{\gamma_{S_*}} \sim \mathcal{N}(0, \kappa_s^2 s\mathbb{I}_s)$ with probability $1 - p_b$. Furthermore, the first term's bound is

derived by the Lemma 29 and it holds with probability $1 - p_b$. *(5)* follows from bounds on eigenvalues of $X_{S_*}^T X_{S_*}$ and the upper bound of square matrices' maximum singular value (Theorem 2.4 from Rudelson & Vershynin (2008) and the by the Gaussian concentration bounds for the term $\eta_\beta; \eta_\beta \sim \mathcal{N}(0, \kappa_s^2 s^2 \mathbb{I}_{s^2})$. *(5)* also holds with probability $1 - p_b$. Assuming $\kappa_\epsilon$ is reasonably small, we notice that if $n \geqslant s^{3/2} ||\alpha_{S_*}||_2$ then the estimation error will be dominated by the second term. The expression holds with $1 - 4p_b$ as we use the concentration inequality four times and combine these events via the union bound.

### G.2 Proof of Risk Analysis Theorem 10

From Theorem 7 we again know given that if satisfy the theorem's assumptions we can recover the true model basis with high probability. We will function under this high probability event and assume that the predicted basis $S_{\mathcal{A}}$ is equal to $S_*$ while computing the estimation error. Thus, by *SPriFed-OMP* our estimation error can be derived as below,

$$\Delta R := R(\hat{\alpha}; X, y, n)$$

$$= \frac{1}{n} \sum_{i=1}^{n} \left( (x_i \hat{\alpha} - y_i)^2 - (x_i \alpha_{S_*} - y_i)^2 \right)$$

$$\overset{(1)}{=} \frac{1}{n} \left( ||X_{S_*} \hat{\alpha} - y||^2 - ||X_{S_*} \alpha_{S_*} - y||^2 \right)$$

$$\overset{(2)}{=} \frac{1}{n} \left( ||X_{S_*} \beta_{S_*})^{-1} \gamma_{S_*} - y||^2 - ||X_{S_*} \alpha_{S_*} - y||^2 \right)$$

$$\overset{(3)}{=} \frac{1}{n} \left( ||X_{S_*} (X_{S_*}^T X_{S_*} + \eta_\beta)^{-1} (X_{S_*}^T y + \eta_{\gamma_{S_*}}) - y||^2 \right.$$
$$\left. - ||X_{S_*} \alpha_{S_*} - y||^2 \right)$$

$$\overset{(4)}{=} \frac{1}{n} \left( ||X_{S_*} ((X_{S_*}^T X_{S_*})^{-1})(X_{S_*}^T y + \eta_{\gamma_{S_*}}) \right.$$
$$- X_{S_*} (X_{S_*}^T X_{S_*})^{-1} (\mathbb{I} + \eta_\beta (X_{S_*}^T X_{S_*})^{-1})^{-1}$$
$$\cdot \eta_\beta (X_{S_*}^T X_{S_*})^{-1} (X_{S_*}^T y + \eta_{\gamma_{S_*}}) - y||^2$$
$$\left. - ||X_{S_*} \alpha_{S_*} - y||^2 \right)$$

$$\overset{(5)}{=} \frac{1}{n} \left( ||X_{S_*} ((X_{S_*}^T X_{S_*})^{-1})(\eta_{\gamma_{S_*}} + X_{S_*}^T \epsilon) \right.$$
$$- X_{S_*} (X_{S_*}^T X_{S_*})^{-1} (\mathbb{I} + \eta_\beta (X_{S_*}^T X_{S_*})^{-1})^{-1}$$
$$\left. \eta_\beta (X_{S_*}^T X_{S_*})^{-1} (X_{S_*}^T y + \eta_{\gamma_{S_*}}) ||^2 \right)$$

$$\overset{(6)}{\leqslant} ||\frac{X_{S_*}}{\sqrt{n}} (\frac{X_{S_*}^T X_{S_*}}{n})^{-1} \frac{(\eta_{\gamma_{S_*}} + X_{S_*}^T \epsilon)}{n}$$
$$- \frac{X_{S_*}}{\sqrt{n}} (\frac{X_{S_*}^T X_{S_*}}{n})^{-1} (\mathbb{I} + \frac{\eta_\beta}{n} (\frac{X_{S_*}^T X_{S_*}}{n})^{-1})^{-1}$$
$$\frac{\eta_\beta}{n} (\frac{X_{S_*}^T X_{S_*}}{n})^{-1} (\frac{X_{S_*}^T y}{n} + \frac{\eta_{\gamma_{S_*}}}{n}) ||_2^2$$

$$\overset{(7)}{\leqslant} 2||\frac{X_{S_*}}{\sqrt{n}} (\frac{X_{S_*}^T X_{S_*}}{n})^{-1} \frac{\eta_{\gamma_{S_*}}}{n} ||^2$$
$$+ 2||\frac{X_{S_*}}{\sqrt{n}} (\frac{X_{S_*}^T X_{S_*}}{n})^{-1} \frac{X_{S_*}^T \epsilon}{n} ||^2$$
$$+ 2||\frac{X_{S_*}}{\sqrt{n}} (\frac{X_{S_*}^T X_{S_*}}{n})^{-1} (\mathbb{I} + \frac{\eta_\beta}{n}$$
$$\cdot (\frac{X_{S_*}^T X_{S_*}}{n})^{-1})^{-1} \frac{\eta_\beta}{n} (\frac{X_{S_*}^T X_{S_*}}{n})^{-1} (\frac{X_{S_*}^T y}{n} + \frac{\eta_{\gamma_{S_*}}}{n}) ||_2^2$$

$$\overset{(8)}{\leqslant} 2\Big(\frac{(1+\zeta_{s+1})\kappa_s s\sqrt{2\log(\frac{2s}{p_b})}}{n(1-\zeta_{s+1})}\Big)^2$$

$$+ 2\Big(\frac{(1+\zeta_{s+1})\sqrt{2ns\log(2s/p_b)}\sigma_\epsilon X_M}{n(1-\zeta_{s+1})}\Big)^2$$

$$+ 2\Big(\frac{s^{3/2}\kappa_s\kappa_M\sqrt{2\log(\frac{2s^2}{p_b})}}{n(1-\zeta_{s+1})^2}\Big((1+\zeta_{s+1})||\alpha_{S_*}||_2$$

$$+ \sqrt{1+\zeta_{s+1}}\kappa_\epsilon + \frac{\kappa_s\sqrt{s}\sqrt{2\log(\frac{2s}{p_b})}}{n}\Big)\Big)^2$$

$$= \mathcal{O}\Big(\frac{2s\log(s))}{n}\Big)$$

*(1)-(3)* follow from definitions of risk and the output of *SPriFed-OMP*. *(4)* follows by the Woodbury expression (Eqn. 31). *(5)* follows since $\alpha_{S_*} = (X_{S_*}^T X_{S_*})^{-1} X_{S_*}^T y$, $X_{S_*}(X_{S_*}^T X_{S_*})^{-1}X_{S_*}^T = y$ and $y = X_{S_*}\alpha_{S_*} + \epsilon$. *(6)* follows by including the $n$ inside the squared norm. *(7)* follows since $||a-b||^2 \leqslant 2(||a||^2 + ||b||^2)$ which follows from Cauchy-Schwartz inequality. *(7)* The terms in *(7)* however are similar to the intermdiate term *(3)* from Theorem 9 except for multiplication by $X_{S_*}/\sqrt{n}$ and the squared $l_2$ norm. Clearly by leveraging, the same inequalities and the RIP definition we can obtain *(8)*.

# H  Additional Results and Notation

## H.1  Simple Matrix Properties

For a general matrix $A \in \mathbb{R}^{n \times p}$, we state some simple properties that normally hold irrespective of its distributions. For a column index subset $S$, the pseudo-inverse of $A_S$ is given by $A^\dagger = (A_S^T A_S)^{-1} A_S^T$. Furthermore, we can easily identify the projection matrix for $A_S$, that maps $A_S$ onto its orthogonal components by, $P_S^\perp = I - P_S$, where $P_S = A_S A_S^\dagger$. A projection matrix $P_S$ has the properties $P_S^2 = P_S$ and $P_S^T = P_S$. Verifying these properties is straightforward by plugging in the value of $P_S$ in terms of $A_S$.

The Singular Value Decomposition of $A$, is written as $A = U\Sigma V^T$ where, $u \in \mathbb{R}^{n \times n}, \Sigma \in \mathbb{R}^{n \times p}$(diag. matrix)$, V \in \mathbb{R}^{p \times p}$. $U, V$ are unitary and $U^T U = UU^T = \mathbb{I}; V^T V = VV^T = \mathbb{I}$. Furthermore, the SVD of $A^T A$ is given by $V(\Sigma^T\Sigma)V^T = VDV^T$, where $D = (\Sigma^T\Sigma)$. Thus, we can see that $A^T AV = VD$ and therefore, the matrix $A^T A$ shares the same eigenvalue and singular values.

Denote the maximum and minimum singular values of a matrix $A$ by $\sigma_{max}(A)$ and $\sigma_{min}(A)$, respectively. Similarly, define the maximum and minimum eigenvalues of matrix $A$ by $\sigma_{\min}(A)$ and $\lambda_{min}(A)$ respectively. Thus, $\lambda_{\min}(X^T X) = \sigma_{max}(X^T X) = \sigma_{max}^2(X)$ and $\lambda_{min}(X^T X) = \sigma_{min}(X^T X) = \sigma_{min}^2(X)$. Finally, the $i^{th}$ singular value and eigenvalue of the matrix $A$ is given by $\sigma_i(A)$ and $\lambda_i(A)$ respectively.

**Lemma 30.** *[Spectral Norm eigen-value bounds] Consider the matrix $A \in \mathbb{R}^{n \times p}$ and vector $w \in \mathbb{R}^p$ then,*

$$\sigma_{\min}(A)||w||_2 \leqslant ||Aw||_2 \leqslant \sigma_{\max}(A)||w||_2$$

**Lemma 31.** *[Woodbury Identity: Kailath Variant] (page 153 from Bishop et al. (1995), Petersen et al. (2008)) For conformable matrices $A, B, C$ we have the following expression for the inverse of matrix sum,*

$$(A + BC)^{-1} = A^{-1} - A^{-1}B(\mathbb{I} + CA^{-1}B)^{-1}CA^{-1}$$

*Assuming, $B = \mathbb{I}, C = N$ we obtain $\rightarrow$*

$$(A + N)^{-1} = A^{-1} - A^{-1}(\mathbb{I} + NA^{-1})^{-1}NA^{-1}$$

*Assuming, $B = N, C = I$ we obtain $\rightarrow$*

$$(A + N)^{-1} = A^{-1} - A^{-1}N(\mathbb{I} + A^{-1}N)^{-1}A^{-1}$$

## H.2 Concentration bounds for Random Variables

We have previously seen in the differential privacy section, that we require that the matrix values or their functional outputs be bounded. Therefore, here we consider some of the useful properties of random variables that allows us to bound their contributions.

**Lemma 32.** *[Concentration Inequality (Boucheron et al., 2013)] For Gaussian random variable $x$ with distribution $\mathcal{N}(0, \sigma^2)$ we have,*

$$Pr[|x| \geqslant t] \leqslant 2e^{\frac{-t^2}{2\sigma^2}} \rightarrow Pr[|x| < t] > 1 - 2e^{\frac{-t^2}{2\sigma^2}}$$

**Lemma 33.** *[Union Bound] The union bound states for any events $E_1, E_2, ..., E_n$ we have,*

$$Pr(\cup_{i=1}^n E_i) \leqslant \sum_{i=1}^n Pr(E_i)$$

**Lemma 34.** *[Norm of matrix with subgaussian entries] (Theorem 4.3.5 from Vershynin (2018))*

*Consider matrix $A \in \mathbb{R}^{n \times p}$ where its entries $A_{i,j}$ are independent, mean zero, subgaussian random variables. Then, for any $t > 0$ we have,*

$$||A||_2 \leqslant CK(\sqrt{n} + \sqrt{p} + t)$$

*with probability at least $1 - 2e^{-t^2}$ and $K = \max_{i,j} ||A_{i,j}||_{\Psi_2}; i \in [n], j \in [p]$.*

**Lemma 35.** *[Lower Bound of Minimum Singular Value for a Random Matrix]: (Theorem 1.2 from Rudelson & Vershynin (2008)) Let $z_1, ..., z_p$ be independent, centered real random variables with variance at least 1 and sub-Gaussian moments bounded by $B$. Let $A \in \mathbb{R}^{p \times p}$ matrix whose rows are independent copies of the random vector $(z_1, ..., z_p)$. Then for every $\varepsilon \geqslant 0 \rightarrow$*

$$\Pr[\sigma_{\min}(A) \leqslant \frac{\varepsilon}{\sqrt{p}}] \leqslant C\varepsilon + c^p$$

*where $C > 0, c \in (0, 1)$ depend polynomially only on $B$.*

**Lemma 36.** *[Concentration of the maximum random variable] Suppose we have $k$ random variables $x_i; i \in [k]$ each with distribution $\mathcal{N}(0, \sigma^2)$. We provide an upper bound for the maximum of these $k$ random variables as following,*

$$\Pr[\max_{i=1}^k |x_i| \leqslant B] \geqslant 1 - p_b$$

*where $B = \sigma\sqrt{2\log(2/p_0)}$ and $p_b$ is the total failure probability such that $p_0 = \frac{p_b}{k}$.*
**Proof:** *Consider the events $E_i$ s.t., $|x_i| \leqslant B; i \in [k]$ and suppose $\Pr[|x_i| \geqslant B] \leqslant p_0$. With Lemma 32 applied to $x_i$, we have,*

$$\Pr[|x_i| \geqslant B] \leqslant 2e^{\frac{-B^2}{2\sigma^2}}$$

$$\Pr[|x_i| \geqslant \sigma\sqrt{2\log(\frac{2}{p_0})}] \leqslant 2\exp(-2\log(2/p_0)\sigma^2/2\sigma^2)$$

$$\leqslant 2\exp(-\log(2/p_0))$$

$$\leqslant p_0$$

*Then, with the union bound, we can consider the probability of the event $E_0$ s.t. $\max_{i=1}^k |x_i| \leqslant B$. Then, $\Pr[E_0^c] = \Pr[\cup_{i=1}^k |x_i| \geqslant B]$ (where at least one value is larger than the bound). Then,*

$$\Pr[E_0^c] = \Pr[\cup_{i=1}^k |x_i| \geqslant B]$$

$$\leqslant \sum_{i=1}^k \Pr[|x_i| \geqslant B]$$

$$= k \cdot p_0$$
$$\rightarrow \Pr[E_0] = 1 - \Pr[E_0^c] = 1 - k \cdot p_0$$

*The first inequality is due to Boole's inequality. In particular, if we wish to bound our failure probabilities to a fixed value $p_b$ then we can have $p_0 = \frac{p_b}{k}$ and thus our bound $B = \sigma \sqrt{2 \log(\frac{2k}{p_b})}$ is achieved.*

