# OpenReview forum: "SPriFed-OMP: A Differentially Private Federated Learning Algorithm for Sparse Basis Recovery"
_TMLR — Accepted by TMLR_

### Review · Reviewer_qoft · 2024-03-22

**Summary Of Contributions:**

The paper focused on privacy-preserving sparse basis recovery problem for high dimension data analysis under the federated learning setting.  Based on the Orthogonal Matching Pursuit algorithm, they propose two new private sparse basis recovery algorithms SPriFed-OMP and SPriFed-OMP-GRAD for the DP-FL problem. The effectiveness of the proposed algorithm is evaluated both theoretically and empirically.

**Audience:**

Yes

**Broader Impact Concerns:**

no concern

**Claims And Evidence:**

Yes

**Requested Changes:**

1. In the introduction, page 2, the paper mentioned that the empirical risk rate for DP-SGD with Lipschitz losses is $O(p/n)$, while it has benn proved that the optimal rate $O(\sqrt{p}/n+1/\sqrt{n})$ can be achieved even for excess population risk.

2. It would be better if the comparison of GPU time or other measures of computation cost could be provided in the experiments.

3. Is there any possible that the risk rate in Theorem 10 can be improved? I guess it might not be easy, the present results are fine with me.

4. The paper mentioned in Section 4 that Gaussian DP optimally composed over multiple iterations. Could the authors explain why it is better than RDP?

5. In NOISY-SMPC, all clients compute $q_i$ and submit is to the server. How about sampling some clients in each iteration (uniformly or using Poisson sampling) to update $q_i$ and submit their local updates to the server? It might decreases the communication cost. The guarantee of privacy could use sub-sampling property [1] to get a better bounds.

[1] Wang Y X, Balle B, Kasiviswanathan S P. Subsampled rényi differential privacy and analytical moments accountant. The 22nd International Conference on Artificial Intelligence and Statistics. PMLR, 2019: 1226-1235.

**Strengths And Weaknesses:**

Strengths

1. Two effective differentially private FL algorithms for sparse basis recovery are proposed, privacy guarantee and utility guarantee are provided  for the proposed algorithms.
2. Experiments show that the proposed algorithms outperform the baselines (DP-SGD, OMP et al.).

Weaknesses

1. The main concern is that the computation cost and the communication cost of the proposed algorithms seem heavy.   SPriFed-OMP and SPriFed-OMP-GRAD need to run Noisy-SMPC and PRIVATE-OLS algorithms many times, the effectiveness of the algorithms may be hard to guarantee when the dimensions are very high or the sample size is large. It looks like the efficiency of the algorithm is sacrificed in exchange for its effectiveness.

2. Theorem 10 provides the risk analysis of SPriFed-OMP, and it shows that the risk is in the order of $O(s/n)$. It seems that it is not good enough. Bassily et al.(2020) and Wang et al.(2022) proved that for Lipschitz non-smooth loss or non-Lipschitz smooth loss, DP-SGD can achieve the optimal rates $O(1/\sqrt{n} + \sqrt{p}/n)$. It would be better if the similar rate can be shown for the proposed algorithms.



[1] Raef Bassily, Vitaly Feldman, Crist´obal Guzm´an, and Kunal Talwar. Stability of stochastic gradient descent on nonsmooth convex losses. Advances in Neural Information Processing Systems, 33, 2020.

[2] Wang P, Lei Y, Ying Y, et al. Differentially private SGD with non-smooth losses. Applied and Computational Harmonic Analysis, 2022, 56: 306-336.

---

> ### Author Response · Authors · 2024-04-22
>
> > Comment 1: “Two effective differentially private FL algorithms for sparse basis recovery are proposed, privacy guarantee and utility guarantee are provided for the proposed algorithms.
> Experiments show that the proposed algorithms outperform the baselines (DP-SGD, OMP et al.).”
>
> **Response:** We thank the reviewer for the positive comments!
>
> > Comment 2: “The main concern is that the computation cost and the communication cost of the proposed algorithms seem heavy. SPriFed-OMP and SPriFed-OMP-GRAD need to run Noisy-SMPC and PRIVATE-OLS algorithms many times, the effectiveness of the algorithms may be hard to guarantee when the dimensions are very high or the sample size is large. It looks like the efficiency of the algorithm is sacrificed in exchange for its effectiveness.”
>
> **Response:** We thank the reviewer for their comment. We respectfully disagree that the computation cost and communication costs of our proposed algorithms are heavy. In fact, we argue below that the computation/communication costs are comparable to vanilla DP-SGD.
>
> First, we note that even vanilla DP-SGD incurs O(p) computation/communication costs per client in each iteration. Indeed, assuming each gradient is $p$-dimensional, even computing and communicating the gradient incur $\mathcal{O}(p)$ costs per client.
>
> Compared to vanilla DP-SGD, our algorithm has to perform the additional steps of secure aggregation using NoisySMPC or PRIVATE-OLS. However, there are efficient implementations of NoisySMPC without significantly increasing the computation/communication costs. For instance, in the SMPC scheme in [1], there is a public-private key-sharing between pairs of clients (facilitated by the server), which sets up common seeds for generating random masks per client pair. When performing SMPC, each client simply adds/subtracts the random masks with other clients to the data, and then sends it to the server. Since the masks generated are common for each pair of clients, the random masks are canceled at the server. Note that with this SMPC implementation, there is a one-time fixed cost between clients and the server to set up the common seeds, which incurs O(n) computation/communication cost per client. During each round, there is also O(n) computation cost per client for adding/subtracting the random masks (note that the same random masks can be used for all features; hence, this cost does not increase with $p$). Since we focus on $n \ll p$, both of these costs are small compared to the O(p) communication/computation cost of vanilla DP-SGD.
>
> Indeed, we can summarize the computation costs of our algorithm as follows. Suppose we have $n$ total clients, and all clients participate in each round. Each client in our algorithms computes the gradient (similar to existing methods like DP-SGD and DP-GCD)in each round. Assuming each gradient is $p$-dimensional, we have $\mathcal{O}(p)$ computations per client. Furthermore, in our method, we compute additional values (for the PRIVATE-OLS routine) that are each $s$-dimensional, adding $\mathcal{O}(s)$ computations per client. Noting $s \ll p$, the computation order remains unchanged.
>
> Similarly, for the two-way communication exchanges between the client and the server, we see that the client shares data artifacts with the server of order $p$ (for even existing DP-FL methods) and additional order-$s$ data artifacts for our method. Given $s \ll p$,  the size order of the data artifacts shared with the server remains unchanged. Thus, our per-client communication cost in each round is $\mathcal{O}(p)$.
>
> Putting these together, we note that the order of our communication and computation costs is $\mathcal{O}(p)$, exactly the same as existing DP-FL methods. Therefore, we argue that our algorithms (focusing on the $n \ll p$ regime) are also computation/communication-efficient.
>
> [1] Bonawitz, Keith, et al. "Practical secure aggregation for privacy-preserving machine learning." proceedings of the 2017 ACM SIGSAC Conference on Computer and Communications Security. 2017.
>
> > Comment 3: “Theorem 10 provides the risk analysis of SPriFed-OMP, and it shows that the risk is in the order of $\mathcal{O}(s/n)$. It seems that it is not good enough. Bassily et al.(2020) and Wang et al.(2022) proved that for Lipschitz non-smooth loss or non-Lipschitz smooth loss, DP-SGD can achieve the optimal rates $\mathcal{O}(\sqrt{p}/n + 1/\sqrt{n})$. It would be better if a similar rate can be shown for the proposed algorithms.”
>
> **Response:** Thank you for raising this concern. We would like to clarify that our proven risk order of $\mathcal{O}(s/n)$ is significantly better than $\mathcal{O}(\sqrt{p}/n)$ results in the above papers. Note that we assume that $s < n \approx \sqrt{p} \ll p$. Therefore, $\mathcal{O}(s/n)$ is much lower than $\mathcal{O}(\sqrt{p}/n)$. This improved risk for the sparse regime is a significant advantage of our proposed algorithms compared to DP-SGD.

---

> > ### Author Response · Authors · 2024-04-22
> >
> > > Comment 4: “In the introduction, page 2, the paper mentioned that the empirical risk rate for DP-SGD with Lipschitz losses is $\mathcal{O}(p/n)$, while it has been proved that the optimal rate
> > $\mathcal{O}(\sqrt{p}/n + 1/\sqrt{n})$  can be achieved even for excess population risk.”
> >
> > **Response:** Thank you for the remark. For clarification, we refer the reviewer to comment 3 above.
> >
> > > Comment 5: “It would be better if the comparison of GPU time or other measures of computation cost could be provided in the experiments.”
> >
> > **Response:** We thank the reviewer for their suggestion. Accordingly, we have added the following paragraph (highlighted) in Section 7 to the updated manuscript.
> >
> > *“To understand the general trend of computational time required by each algorithm, we consider the example case when $n=2000, s=20, p=20000$. Here, we provide the CPU time measured by the time library in Python. We note that SPriFed-OMP takes 20.17 seconds, SPriFed-OMP-Grad takes 26.38 seconds, DP-SGD takes 11.99 seconds, and DP-GCD takes 9.56 seconds. We note that the SPriFed-OMP and SPriFed-OMP-GRAD algorithms take longer due to their separate computations for the sparse model in each step. However, these computation times are still comparable to that of DP-SGD and DP-GCD. On the other hand and as reported earlier, the sparse-recovery performance of our proposed algorithms is much better."*
> >
> > > Comment 6: “Is there any possible that the risk rate in Theorem 10 can be improved? I guess it might not be easy, the present results are fine with me.”
> >
> > **Response:** As mentioned in Comment 3 above, even for non-private versions of  OMP,  their risk rate is at the same level as the one we show for our private algorithms, *SPriFed-OMP* and *SPriFed-OMP-GRAD*. Hence, we do not expect that it can be improved.

---

> > > ### Author Response · Authors · 2024-04-22
> > >
> > > > Comment 7: “The paper mentioned in Section 4 that Gaussian DP optimally composed over multiple iterations. Could the authors explain why it is better than RDP?”
> > >
> > > **Response:** Thank you for raising this concern. We choose GDP because it has been shown to produce lower $\epsilon$ values when composed over a finite (possibly small) number of steps $T$ [3]. This small-$T$ regime is important for our work because our proposed algorithms aim to find the correct basis with only a small number of iterations (proportional to the sparsity level $s$). A number of references have pointed out the efficiency of GDP in this regime. For instance, in  Table 1 from Article [1] (which we reproduce below), the authors empirically compare RDP, naive DP, advanced DP, and Gaussian DP. Here, the authors look at the net composition budget amassed by composing over $T=50$ iterations of $0.2$-$\epsilon$ DP.
> > >
> > > Method / $\delta$      | 0.1     | 0.01   | 0.001  | 0.0001 |
> > > | -------- | -------- | -------- | -------- | -------- |
> > > GDP		          | 3.1     | 5.06   | 6.47    | 7.62 |
> > > Naive 	 	          | 9.89   | 9.99   | 10        | 10 |
> > > Advanced	          | 5.25   | 6.51   | 7.47     | 8.28 |
> > > RDP		                  | 12.14 | 17.17 | 21.03   | 24.28 |
> > >
> > > We can see that overall GDP performs the best (in terms of the resulting $\epsilon$ values) across varying $\delta$. Furthermore, we see that RDP is much worse than other mechanisms (even naive DP) due to a large constant in its composition result (refer to Corollary 1 in [2]). For small values of $T$, this constant matters a lot more than the order with respect to $T$, which is why even though RDP composition attains an order of  $\sqrt{T}$ (for the resulting $\epsilon$ values) and naive DP composition attains an order of $T$, the actual $\epsilon$ value of RDP composition can still be much larger than naive DP.
> > >
> > > To address this comment, we provide the following clarification  in Section 4 (highlighted)  in the revised manuscript,
> > >
> > > *“Reasons behind choosing GDP for privacy analysis: We choose GDP because it has been shown to produce lower $\epsilon$ values when composed over a finite (possibly small) number of steps $T$ [3]. This small-$T$ regime is important for our work because our proposed algorithms aim to find the correct basis with only a small number of iterations (proportional to the sparsity level $s$). An empirical comparison of various composition mechanisms can be found, e.g., in Table [1] of Reference [1]. The composition of $50$ $0.2-\epsilon$ DP mechanisms is performed, and the resulting $\epsilon$ values are compared for various composition mechanisms, including Renyi DP, Advanced composition, naive DP, and Gaussian DP. One can observe that GDP performs the best (i.e., produces the smallest $\epsilon$ values) across varying values of $\delta$.”*
> > >
> > > [1] Liu, Yi, et al. "Identification, amplification and measurement: A bridge to Gaussian differential privacy." Advances in Neural Information Processing Systems 35 (2022): 11410-11422.
> > >
> > > [2] Mironov, Ilya. "Rényi differential privacy." 2017 IEEE 30th Computer Security Foundations Symposium (CSF). IEEE, 2017.
> > >
> > > [3] Dong, Jinshuo, Aaron Roth, and Weijie J. Su. "Gaussian differential privacy." Journal of the Royal Statistical Society Series B: Statistical Methodology 84.1 (2022): 3-37.

---

> > > > ### Author Response · Authors · 2024-04-22
> > > >
> > > > > Comment 8: “In NOISY-SMPC, all clients compute and submit is to the server. How about sampling some clients in each iteration (uniformly or using Poisson sampling) to update and submit their local updates to the server? It might decreases the communication cost. The guarantee of privacy could use sub-sampling property [1] to get a better bounds.”
> > > >
> > > > **Response:** Thank you for this valuable suggestion. Although we agree that sub-sampling can improve privacy guarantees and reduce communication costs, we envision certain technical difficulties in handling sub-sampling in our setting.
> > > >
> > > > First, note that we assume RIP in this work. If Poisson sampling is used, we would need RIP to hold for each subset of the clients. This assumption can be quite difficult to verify in practice.
> > > >
> > > > Second, we understand that sub-sampling will mostly help when the number of clients is already large so that a subset of the clients can provide sufficiently accurate signals (e.g., the feature correlation values in our proposed algorithms) for the progress of the algorithms. However, since we are already targeting the setting where $n$ is much smaller than $p$, we are concerned that the number of clients may be too small for sub-sampling to provide accurate signals.
> > > >
> > > > Due to these two reasons, we did not study sub-sampling in this paper. It is nonetheless an interesting direction for future work. Hence, we acknowledge our limitation and this future direction in the following paragraph in Section 7 (highlighted).
> > > >
> > > > *“Remark about Sub-sampling with DP-FL: We recognize that under a DP-FL setting, sub-sampling is beneficial in reducing per-round communication cost and amplifying privacy. However, the benefit of sub-sampling in the $n \ll p$ regime that we focus on in this paper has not been thoroughly studied, which could be an interesting direction for future work.”*

---

### Review · Reviewer_dfg4 · 2024-03-29

**Summary Of Contributions:**

This paper tackles the challenging problem of federated sparse basis recovery (i.e. the data is split among clients, the operations are computed by a central server) under privacy constraints (namely differential privacy) in the case where the number of model dimensions $p$ is much larger than the number of samples $n$. The authors intend to fill a gap in literature where either private solutions or federated solutions have been proposed up to now for sparse basis recovery; while most of these methods can be extended to a private-federated setting in a straightforward manner, it systematically hurts their accuracy. To mitigate this issue, the authors propose a new approach which combines both federated and private aspects simultaneously, called SPriFed-OMP. This algorithm elaborates on the federated adaptation of the well-known algorithm 'Orthogonal Matching Pursuit', while introducing privacy via Gaussian Differential Privacy. They propose an alternative version, SPriFed-OMP-GRAD, which revisits their first method from a gradient perspective. To justify their approach, the authors present a theoretical analysis of their algorithms (namely, privacy and utility results), which demonstrate that they can recover the true sparse basis with high probability in the case where $n=O(\sqrt{p})$. Finally, they conduct numerical experiments on both synthetic and realistic datasets, and compare their approach to concurrent works.

**Audience:**

Yes

**Broader Impact Concerns:**

None.

**Claims And Evidence:**

Yes

**Requested Changes:**

I recommend the authors to :
- add a more detailed justification on the use of Gaussian Differential Privacy
- add numerical experiments where the differential privacy parameter $\epsilon$ takes several values (on both synthetic and realistic datasets) to observe the dependence to the privacy constraint
- add a discussion on the modifications of their algorithms to handle heterogeneity issues and numerical experiments which show the robustness of their current setting to this question

**Strengths And Weaknesses:**

**Strengths**
- The paper is very well written; the computations are carefully detailed, a lot of intuition is given when concepts and algorithms are introduced, which makes the reading pleasant.
- This work is the first one to combine both federated and privacy constraints in the overparametrized regime of sparse basis recovery, which is of main interest for the community.
- This work is complete in the sense that it presents both theoretical and practical results, which strengthens the soundness of the approach.
- Section 5, which features the utility results, is particularly exhaustive and insightful.
- The numerical results are meaningful, as the authors propose several ablation studies.

**Weaknesses**
- The justification for choosing Gaussian Differential Privacy instead of other differential privacy approaches such as Rényi Differential Privacy (RDP) is partial: the authors invoke that the composition rule has a better behaviour, but it is hard to see why (for example, the composition in RDP is straightforward)
- No ablation study is conducted on the differential privacy parameters used in the numerics: in the synthetic datasets, the authors take $\epsilon=5.34$, while they consider $\epsilon=8.34$ for the realistic datasets, without further justification. These values are unfortunately too high to be used in practice. Hence, the utility/privacy tradeoff is not well highlighted in the numerics.
- This work does not tackle the question of heterogeneity among the datasets, which is nonetheless a major issue in FL. As far as I understand, the authors study settings where the whole data is homogeneous and is assumed to be separated between clients (see the numerics). In practice, practitioners observe heterogeneity across the local datasets, which hurt the aggregation in the naive FL setting. For instance, this setting has been studied in [1], where the authors propose to combine privacy with control variates.

[1] Noble et al. Differentially Private Federated Learning on Heterogeneous Data. 2022.

---

> ### Author Response · Authors · 2024-04-22
>
> > *Comment 1:* “The paper is very well written; the computations are carefully detailed, a lot of intuition is given when concepts and algorithms are introduced, which makes the reading pleasant. This work is the first one to combine both federated and privacy constraints in the overparametrized regime of sparse basis recovery, which is of main interest for the community. This work is complete in the sense that it presents both theoretical and practical results, which strengthens the soundness of the approach. Section 5, which features the utility results, is particularly exhaustive and insightful. The numerical results are meaningful, as the authors propose several ablation studies.”
>
> **Response:** We thank the reviewer for the positive comments!
>
> > *Comment 2:* “The justification for choosing Gaussian Differential Privacy instead of other differential privacy approaches such as Rényi Differential Privacy (RDP) is partial: the authors invoke that the composition rule has a better behaviour, but it is hard to see why (for example, the composition in RDP is straightforward). Requested Change: add a more detailed justification on the use of Gaussian Differential Privacy”
>
> **Response:** Thank you for raising this concern. We choose GDP because it has been shown to produce lower $\epsilon$ values when composed over a finite (possibly small) number of steps $T$ [3]. This small-$T$ regime is important for our work because our proposed algorithms aim to find the correct basis with only a small number of iterations (proportional to the sparsity level $s$). A number of references have pointed out the efficiency of GDP in this regime. For instance, in  Table 1 from Article [1] (which we reproduce below), the authors empirically compare RDP, naive DP, advanced DP, and Gaussian DP. Here, the authors look at the net composition budget amassed by composing over $T=50$ iterations of $0.2$-$\epsilon$ DP:
>
> Method / $\delta$      | 0.1     | 0.01   | 0.001  | 0.0001 |
> | -------- | -------- | -------- | -------- | -------- |
> GDP		          | 3.1     | 5.06   | 6.47    | 7.62 |
> Naive 	 	          | 9.89   | 9.99   | 10        | 10 |
> Advanced	          | 5.25   | 6.51   | 7.47     | 8.28 |
> RDP		                  | 12.14 | 17.17 | 21.03   | 24.28 |
>
> We can see that overall GDP performs the best (in terms of the resulting $\epsilon$ values) across varying $\delta$. Furthermore, we see that RDP is much worse than other mechanisms (even naive DP) due to a large constant in its composition result (refer to Corollary 1 in [2]). For small values of $T$, this constant matters a lot more than the order with respect to $T$, which is why even though RDP composition attains an order of  $\sqrt{T}$ for the resulting $\epsilon$ values and naive DP composition attains an order of $T$, the actual $\epsilon$ value of RDP composition can still be much larger than naive DP.
>
> To address this comment, we provide the following clarification  in Section 4 (highlighted)  in the revised manuscript,
>
> *“Reasons behind choosing GDP for privacy analysis: We choose GDP because it has been shown to produce lower $\epsilon$ values when composed over a finite (possibly small) number of steps $T$ [3]. This small-$T$ regime is important for our work because our proposed algorithms aim to find the correct basis with only a small number of iterations (proportional to the sparsity level $s$). An empirical comparison of various composition mechanisms can be found, e.g., in Table [1] of Reference [1]. The composition of $50$ $0.2-\epsilon$ DP mechanisms is performed, and the resulting $\epsilon$ values are compared for various composition mechanisms, including Renyi DP, Advanced composition, naive DP, and Gaussian DP. One can observe that GDP performs the best (i.e., produces the smallest $\epsilon$ values) across varying values of $\delta$.”*
>
> [1] Liu, Yi, et al. "Identification, amplification and measurement: A bridge to Gaussian differential privacy." Advances in Neural Information Processing Systems 35 (2022): 11410-11422.
>
> [2] Mironov, Ilya. "Rényi differential privacy." 2017 IEEE 30th Computer Security Foundations Symposium (CSF). IEEE, 2017.
>
> [3] Dong, Jinshuo, Aaron Roth, and Weijie J. Su. "Gaussian differential privacy." Journal of the Royal Statistical Society Series B: Statistical Methodology 84.1 (2022): 3-37.

---

> > ### Author Response · Authors · 2024-04-22
> >
> > > *Comment 3:* “No ablation study is conducted on the differential privacy parameters used in the numerics: in the synthetic datasets, the authors take $\epsilon = 5.34$,  while they consider
> > $\epsilon = 8.34$ for the realistic datasets, without further justification. These values are, unfortunately too high to be used in practice. Hence, the utility/privacy tradeoff is not well highlighted in the numerics. Requested Change: add numerical experiments where the differential privacy parameter takes several values (on both synthetic and realistic datasets) to observe the dependence to the privacy constraint”
> >
> > **Response:** We thank the reviewer for raising this concern. To address this concern, we provide the following experimental updates in Section 7 of the article (highlighted). We also provide the same note here (except for the plots),
> >
> > *“Ablation study of privacy-utility trade-off: To understand the impact of the privacy parameters on the algorithm’s performance, we next study the test error and the number of bases recovered for both SPriFed-OMP and SPriFed-OMP-Grad, as we vary the net epsilon value. We first consider the synthetic datasets of Section 7.2 for the case of $n = 2000, s=5$, and $p=10000$. Both algorithms are run for $3$ random seeds. The value of $\mu_p$ is varied over $[0.1, 0.2, 0.3, 0.4, 0.5, 0.8, 1]$, which in turn affects the final $\epsilon$ value. Note that $\mu_s = 0.09$ is constant since it does not greatly affect the net $\epsilon$ value. In Fig. 5(a) and (b), we plot both algorithms' test MSE and the basis recovery capabilities vs $\epsilon$. We can observe that, as $\epsilon$ increases, the performance of both algorithms improves (i.e., the test MSE decreases, and the number of correctly recovered bases increases). Both algorithms generally reach constant performance for any $\ epsilon$ larger than $5$ (which is around the value chosen in our results reported in Section 7.2).*
> >
> > *We then perform a similar study of privacy-utility tradeoff on the realistic chop dataset. We consider the following setup where $\mu_p$ is again varied over $[0.1, 0.2, 0.3, 0.4, 0.5, 0.8, 1]$ to adjust the net privacy budget. We run both algorithms over 7 random seeds and set the clipping bound for elements of $X, y$ to be $0.5$. $\mu_s = 0.15$. Since we do not know the underlying basis for the chop dataset, we simply report the Test MSE error in Fig. 5(c). We can see that as $\epsilon$ increases, the error generally tends to lower. However, we observe that the error dips the most from $\epsilon \in [6, 12]$. Thus, we pick $\epsilon \approx 8$, which gives a reasonable privacy-utility trade-off for a $\epsilon < 10$.”*
> >
> > > *Comment 4:* “This work does not tackle the question of heterogeneity among the datasets, which is nonetheless a major issue in FL. As far as I understand, the authors study settings where the whole data is homogeneous and is assumed to be separated between clients (see the numerics). In practice, practitioners observe heterogeneity across the local datasets, which hurt the aggregation in the naive FL setting. For instance, this setting has been studied in [1], where the authors propose to combine privacy with control variates. Requested Change: add a discussion on the modifications of their algorithms to handle heterogeneity issues and numerical experiments which show the robustness of their current setting to this question”
> >
> > **Response:** Thank you for providing the reference and suggesting a discussion regarding heterogeneity. We believe our results do not depend on whether the clients are homogeneous or heterogeneous. To see this, note that our algorithm aggregates the data together over all clients per round using NoisySMPC. As a result, the server receives the same aggregated values (e.g., feature correlation),  irrespective of whether individual data is homogeneous or heterogeneous. We did not provide new numerical results for the same reason. We hope this clarifies the reviewer’s concerns.
> >
> > In Section 2 of the paper, we provide the following footnote about heterogeneity under the RIP assumption (highlighted);
> > *“Note that we do not specifically assume that the clients are homogeneous. Even if the clients are heterogeneous, as long as the RIP assumption holds, our analytical results in the paper will hold.”*

---

### Review · Reviewer_FeBV · 2024-04-09

**Summary Of Contributions:**

In this work the authors propose a new method for private sparse recovery in the context of federated learning based on OMP. The proposed method is able to recovery the sparse basis with $O(\sqrt{p})$ samples. Empirical results show that the proposed method is able to significantly outperform prior baselines.

**Audience:**

Yes

**Claims And Evidence:**

No

**Requested Changes:**

1. Improving utility bounds in Theorem 7-10 to make it clear how privacy affects the utility.
2. Adding pareto privacy utility frontier is useful.

**Strengths And Weaknesses:**

Strengths:
- The paper is well written, with good presentation of background and the methodology.
- The method comes with theoretical guarantees.

Weaknesses:
- It looks like in Algorithm 3, NOISY-SMPC on $X_k$ and $X_{l^*}$ is used multiple times but with different noise scale. I'm wondering whether the privacy budget for the proposed method is overestimated.
- Assumption 2 which requires bounded $X$ seems quite restrictive. Could the authors provide some justifications why this is a realistic setting?
- The accuracy analysis is hard to parse. Theorem 7 and 8 do not have privacy parameter, making it hard to understand how privacy affect the recovery accuracy. Similarly in Theorem 9 and 10, while $O(slog(s))$ error is good, I don't see the privacy parameter in the bound. Is it implicitly omitted?
- In the current experiments the privacy parameter is fixed. Could the authors show the pareto privacy-utility tradeoff?

---

> ### Author Response · Authors · 2024-04-22
>
> > Comment 1: “The paper is well written, with good presentation of background and the methodology. The method comes with theoretical guarantees.”
>
> **Response:** Thank you for the positive comments!
>
> > Comment 2: It looks like in Algorithm 3, NOISY-SMPC on $X_k$ and $X_l^{*}$ is used multiple times but with different noise scale. I'm wondering whether the privacy budget for the proposed method is overestimated.
>
> **Response:** We thank the reviewer for their question. As the reviewer correctly pointed out, we have intentionally performed re-privatization in each step of Algorithm 3. , first for the overall feature space and then for the feature with the maximum gradient/correlation. Although this might add some additional privacy costs, as we argue below, there is a significant advantage to doing such an additional step of re-privatization.
> First, For $X_k$, since $k$ ranges over all $p$ features, our privacy noise needs to be scaled up to a variance of order $\mathcal{O}(p)$. Furthermore, we need a larger privacy budget for steps with $p$ features due to this step’s larger $l_2$ sensitivity. Intuitively, the larger noise in this step will affect the eventual utility of our algorithm.
> In contrast, when we only deal with the selected features ($X_l^{*}$ case), we only have at most $s$ features. Thus, the noise added to these features can be much smaller,  with a variance of order only $\mathcal{O}(s)$. In other words, with the smaller noise, the features retain more of their utility when we compute the estimated alpha. On the other hand, thanks to the smaller $l_2$ sensitivity, such a smaller noise does not increase the privacy budget significantly.
> Third, to see how the privacy budgets of the above two steps add together, suppose that we pick the following $\mu-GDP$ parameters, $\mu_p = 0.5$ for the steps with $p$ features and $\mu_s = 0.09$ for steps with $s$ features. Now, we leverage the $\mu-GDP$ composition where net GDP-$\mu = \sqrt{\mu_1^2 + \mu_2^2 +...+\mu_T^2}$ (over $T$ steps). Suppose we have $T = 25$ additional steps with the $\mu_s$ budget. Even then, we will still only have a net additional budget of $\sqrt{T} \mu_s = \sqrt{25} \cdot 0.09 \approx 0.45$, which only adds a value smaller than the total $\mu_p$ budget of a single $p$ step. However, in return for a small increase in privacy costs, we obtain much more accurate estimates for the resulting values (e.g., predicted model $\alpha$) computed over the selected important features, allowing us to present a risk analysis independent of the total feature dimension $p$. As a result,  both Theorems 9 and 10 have high utility due to the selective re-privatization of the most important basis features.
>
> We hope the above explanation clarifies why re-privatization is helpful and why our privacy budget is not overestimated.
>
> > Comment 3: Assumption 2, which requires bounded $X$, seems quite restrictive. Could the authors provide some justifications for why this is a realistic setting?
>
> **Response:** We thank the reviewer for raising this concern. We believe it is reasonable to assume that $X$ has bounded elements for the following reasons. We note that, usually, before training, we standardize our data. Post-standardization, we can observe most data lies between 1 to 3 standard deviations of the mean (here 0) with high probability. Thus, we already have some sense of implicit boundedness in the dataset. We could also combine the standardization above with explicit clipping. We could either do a min-max standardization in the first place or regular standardization and simply clip the data and label values at a reasonable threshold to obtain $X$ with bounded elements. Again, due to the high probability claim, most data clipped will lie in the outlier regime and can be safely removed. Finally, we could assume the clipping parameter as a hyperparameter to obtain better utility-privacy trade-offs. Note that the clipping bound is regularly considered a hyperparameter, as seen in the popular DP-SGD paper [1].
>
> [1] Abadi, Martin, et al. "Deep learning with differential privacy." Proceedings of the 2016 ACM SIGSAC conference on computer and communications security. 2016.

---

> > ### Author Response · Authors · 2024-04-22
> >
> > > Comment 4: The accuracy analysis is hard to parse. Theorem 7 and 8 do not have privacy parameter, making it hard to understand how privacy affect the recovery accuracy. Similarly in Theorem 9 and 10, while error is good, I don't see the privacy parameter in the bound. Is it implicitly omitted?  Requested Change: Improving utility bounds in Theorem 7-10 to make it clear how privacy affects the utility.
> >
> > **Response:** Thank you for the suggestion. We note that in theorems 7 and 8, we have the privacy parameters in the form of $\mu_p$ and $\mu_s$; both are related to the privacy guarantees in Theorem 6. In Theorems 9 and 10, we still see the privacy parameter $\mu_s$, which is again related to the privacy guarantees in Theorem 6. We do not see $\mu_p$ here since the results in Theorems 9 and 10 are based on the outputs of the PRIVATE-OLS routine, where we already have access to the predicted set of bases. Thus, noise added to the model is only based on the $\mu_s$ parameter reserved for data artifacts with$s$ dimensions.
> >
> > Given the reviewer’s comment, we have made the following changes to our paper,
> > - After Theorems 7 and 8, we added that *“Even though we do not explicitly mention privacy requirements in the Theorems 7 and 8, the results here directly depend on the privacy parameters such as $\mu_s$ and $\mu_p$, which will be eventually connected to the privacy guarantee in the Theorem 6.”*
> > - After Theorems 9 and 10, we add “*Results in Theorems 9 and 10 only depend on the $\mu_s$ parameter and not $\mu_p$ privacy parameter. The reason is that the privatized model in these results is outputted based on the PRIVATE-OLS routine, which only adds noise with a privacy parameter of $\mu_s$ (as the predicted bases are already known).”*
> >
> > > Comment 5: In the current experiments the privacy parameter is fixed. Could the authors show the pareto privacy-utility tradeoff? Requested Change: Adding Pareto privacy utility frontier is useful.
> >
> > **Response:** Thank you for this suggestion. To address this concern, we provide the following experimental updates in Section 7 of the article (highlighted). We also provide the same note here (except for the plots which are available in Figure 5 of the updated manuscript),
> >
> > *“Ablation study of privacy-utility trade-off: To understand the impact of the privacy parameters on the algorithm’s performance, we next study the test error and the number of bases recovered for both SPriFed-OMP and SPriFed-OMP-Grad, as we vary the net epsilon value. We first consider the synthetic datasets of Section 7.2 for the case of $n = 2000, s=5$, and $p=10000$. Both algorithms are run for $3$ random seeds. The value of $\mu_p$ is varied over $[0.1, 0.2, 0.3, 0.4, 0.5, 0.8, 1]$, which in turn affects the final $\epsilon$ value. Note that $\mu_s = 0.09$ is constant since it does not greatly affect the net $\epsilon$ value. In Fig. 5(a) and (b), we plot both algorithms' test MSE and the basis recovery capabilities vs $\epsilon$. We can observe that, as $\epsilon$ increases, the performance of both algorithms improves (i.e., the test MSE decreases, and the number of correctly recovered bases increases). Both algorithms generally reach constant performance for any $\ epsilon$ larger than $5$ (which is around the value chosen in our results reported in Section 7.2).*
> >
> > *We then perform a similar study of privacy-utility tradeoff on the realistic chop dataset. We consider the following setup where $\mu_p$ is again varied over $[0.1, 0.2, 0.3, 0.4, 0.5, 0.8, 1]$ to adjust the net privacy budget. We run both algorithms over 7 random seeds and set the clipping bound for elements of $X, y$ to be $0.5$. $\mu_s = 0.15$. Since we do not know the underlying basis for the chop dataset, we simply report the Test MSE error in Fig. 5(c). We can see that as $\epsilon$ increases, the error generally tends to lower. However, we observe that the error dips the most from $\epsilon \in [6, 12]$. Thus, we pick $\epsilon \approx 8$, which gives a reasonable privacy-utility trade-off for a $\epsilon < 10$.”*

---

### Author Response · Authors · 2024-04-22

We would like to thank the editor and the reviewers for their valuable time and for providing detailed comments about our work. The suggestions have helped to improve the manuscript significantly. We have addressed each reviewer's comment point-by-point, only electing to combine the exact same comments together for readability. Changes made in the manuscript are highlighted in magenta.

Below, we summarize the significant changes to the manuscript,
- We added a remark explaining the choice of Gaussian DP over Renyi DP. More details are presented in the responses below as well.
- We have included additional experiments demonstrating the utility-privacy tradeoffs.
- We have added remarks to discuss sub-sampling and heterogeneity for DP-FL in the context of the proposed methods.
- We have included results on the run times for each method.

The response below includes several more details addressing each comment, and we believe that the changes made to the manuscript align with the requested modifications. Please let us know if there are additional feedback and suggestions that can enhance the manuscript. Thank you for your time!

---

### Decision · Action_Editor_fsyF · 2024-05-27

**Recommendation:** Accept with minor revision

**Comment:**

All of the reviewers are positive about accepting the paper, and the authors have already addressed most of the critique and comments in the rebuttal process. However, I think some room for improvement still remains, which I think the authors can address with another revision.

Please consider the following points when preparing the final version:

- Please have still a look at the comment of reviewer FeBV : You could make the utility bounds more understandable by writing them in terms of the privacy parameters $\varepsilon$ and $\delta$ instead of the noise scale parameter $\sigma$. Please make it more explicit what does $\sigma_{\varepsilon}$ mean.

- Related to the discussion with reviewer dfg4 on the privacy analysis and to the related modifications : As the privacy guarantees are essentially those of a composition of Gaussian mechanisms, it is obvious that GDP will give the best possible $(\varepsilon,\delta)$-DP guarantees for the method. One can consider a non-adaptive composition of Gaussian mechanisms also as a vector-valued Gaussian mechanism, and then it is obvious that the GDP-bound gives a tight bound whereas RDP does not, since GDP gives tight bounds for the Gaussian mechanism (see e.g. Balle and Wang, 2018). And with given sensitivities for individual mechanisms, in case of adaptive compositions, the situation is similar, thanks to the results, e.g., by Dong et al. (2019) which you also cite. You could consider rephrasing that part and replacing/augmenting the numerical comparisons with short reasoning.

- In the privacy analysis of Alg. 3 (Theorem 6), is the DP cost of steps 4-5 included ?

- Add names of the datasets either to figures or captions.

- Please carefully check the writing. Few example: p.1: "Forward-Backward Algorithmand Least Angles Regression, \emph{those} leverage sparsity..." -> "Forward-Backward Algorithmand Least Angles Regression, which leverage sparsity..."
Figure 1 y-label: "arifact" - "artifact". There seem to be other similar examples.

- I find the notation a bit confusing: $\mathbf{X}$ seems to denote both the set of features, i.e., $\mathbf{X} = \{ x_1, \ldots, x_n \}$, where $x_i \in \mathbb{R}^n$ for all $i \in [n]$, but also the matrix $\mathbf{X} \in \mathbb{R}^{n \times p}$, where the feature vectors are row vectors. In Alg. 2 to 5, you have $p$ features and $n$ labels. Please go through the notation carefully.

**Audience:**

This paper proposes a general DP ML method for the sparse recovery which does have practical applications and as pointed out by reviewer qoft, the topic of sparse recovery has received attention from the ICLR community.

**Claims And Evidence:**

This paper considers private federated sparse basis recovery. The work bridges a gap in the literature as existing methods are either private or federated solutions for sparse basis recovery. The proposed method is based on using the Gaussian mechanism in the data-sensitive parts which makes it suitable for summation protocols and to off-loading the noise addition to the clients. As a result, the DP analysis is essentially the analysis of Gaussian mechanisms composotions. While the method mostly seems to be a modification / randomization of an existing method, there are non-trivial parts related to the sensitivity analysis and to adjusting the parameters of different noise-adding parts. Also, the paper proposes a theoretical utility analysis and additionally a gradient-based alternative.

The numerical experiments on both synthetic and real-world datasets verify the effectiveness of this method (the first distributed algorithm desgined for this problem) when compared to existing applicable baseline methods (DP-SGD and DP-CGD combined with LASSO) that are also hyperparameter tuned without accounting for the tuning privacy cost. Also, the theoretical bounds indicate the convergence of the method.